# Genetically encoded assembly recorder temporally resolves cellular history

Yuqing Yan[1,2,3,9], Jiaxi Lu[1,2,9], Zhe Li[4,8,9], Zuohan Zhao[1,2], Timothy F. Shay[5], Shunzhi Wang[4], Yaping Lei[5], Yimei Wang[1,2], Wei Chen[6], Patrick Parker[6], Hongru Yang[1,2], Aileen Qi[1], Yongzhi Sun[1,2], Dwight E. Bergles[3,6], David Baker[4] & Dingchang Lin[1,2,3,7 ✉]

Cells constantly change their molecular state in response to internal and external cues[1]. Mapping cellular activity in tissues with spatiotemporal precision is essential for understanding organ physiology, pathology and regenerative processes. Current cell-sensing modalities primarily rely on either end point analysis that takes static snapshots[2] or real-time sensing that monitors a small subset of cells[3,4]. Here we introduce granularly expanding memory for intracellular narrative integration (GEMINI), an in cellulo recording platform that leverages a computationally designed protein assembly as an intracellular memory device to record the history of individual cells. GEMINI grows predictably within live cells, capturing cellular events as tree-ring-like fluorescent patterns for imaging-based retrospective readout. Absolute chronological information of activity histories is attainable with hour-level accuracy. GEMINI effectively maps differential NF-κB-mediated transcriptional changes, resolving fast dynamics of 15 min and providing quantifiable signal amplitudes. In a xenograft model, GEMINI records inflammation-induced signalling dynamics across tissue, revealing spatial heterogeneity linked to vascular density. When expressed in the mouse brain, GEMINI minimally impacts neuronal functions and can resolve both transcriptional changes and activity patterns of neurons. Together, GEMINI provides a robust and generalizable means for spatiotemporal mapping of cell dynamics underlying physiological and pathological processes in both culture and intact tissues.

It has been a longstanding goal in biology to map cell dynamics spatiotemporally across a large cell population within intact tissues, which is crucial to understand how cell signaling is coordinated within an organ. One strategy is to write cellular histories within individual cells for retrospective readout. To this end, signal integrators have been explored to accumulate reporters in the cytoplasm during a user-defined period[5–10]. Although promising, these approaches fail to resolve signalling dynamics and individual events.

This limitation could be overcome by introducing intracellular memory devices to physically write signals. Efforts have been made to develop such devices using nucleic acids, where cellular events are recorded as edits and retrieved via sequencing[11–22]. The relatively slow editing kinetics limits their capability to capture events with rapid dynamics[23]. Moreover, sequencing methods used in signal retrieval typically require structural disruption of cells or tissues, therefore losing spatial information. Although many recorders can write the temporal order of events[13,14,19–22], absolute chronological information and cell dynamics encoding are still unavailable.

Protein assemblies have emerged as promising complementary memory devices. One-dimensional (1D) assemblies have received the most attention due to their ticker-tape-like linear recording[24,25]. During elongation, cellular histories are recorded as linear segmented fluorescent patterns, and signals are retrievable later by confocal imaging. Leveraging distinct linear assemblies, Lin et al. have demonstrated recordings that provide absolute temporal information with hour-level resolution[24], and Linghu et al. have resolved the order of events and showed the potential for in vivo implementation[25]. Nevertheless, their practical applications have encountered several obstacles, many of which are intrinsic to linear assemblies. Long filamentous assemblies can mechanically perturb the plasma membrane and interrupt cell division and migration, and their arbitrary spatial orientations complicate imaging-based signal readout. Despite the clear need for new recording scaffolds, natural candidates are scarce: among over 200,000 structurally characterized proteins deposited in the Protein Data Bank, only approximately 40 have been reported to assemble into extendable lattices in live cells[26], and even fewer are suitable for developing recorders.

Here we crafted an in cellulo recording platform termed GEMINI that addresses the above challenges. To overcome the limited candidates, we combined computational design and experimental screening to create new intracellular protein assemblies, greatly expanding the existing toolbox for recorder scaffolds. These granular assemblies are smaller than cells in all dimensions, thus minimizing mechanical

[1]Department of Materials Science and Engineering, Whiting School of Engineering, Johns Hopkins University, Baltimore, MD, USA. [2]Institute for NanoBiotechnology, Whiting School of Engineering, Johns Hopkins University, Baltimore, MD, USA. [3]Kavli Neuroscience Discovery Institute, Johns Hopkins University, Baltimore, MD, USA. [4]Institute for Protein Design, University of Washington, Seattle, WA, USA. [5]Division of Biology and Biological Engineering, California Institute of Technology, Pasadena, CA, USA. [6]Solomon H. Snyder Department of Neuroscience, School of Medicine, Johns Hopkins University, Baltimore, MD, USA. [7]Center for Cell Dynamics, School of Medicine, Johns Hopkins University, Baltimore, MD, USA. [8]Present address: Department of Biomedical Engineering, Southern University of Science and Technology, Shenzhen, China. [9]These authors contributed equally: Yuqing Yan, Jiaxi Lu, Zhe Li. ✉e-mail: dclin@jhu.edu

perturbation and membrane deformation. They grow isotropically in 3D, allowing accurate readout independent of their orientations, paving the way for large-scale automated signal readout by imaging. GEMINI functions like molecular 'tree rings': as the assembly expands through continued subunit addition, it lays down successive fluorescent layers that encode timing and amplitude of cellular events. The concentric rings at the cross-section can then be read out retrospectively by imaging to reconstruct cellular history.

## Construction of GEMINI

A complete GEMINI system has three components: (1) blank subunits that enable steady and predictable assembly; (2) reporter subunits that transduce cellular events into recordable signals; and (3) timestamp subunits that map GEMINI growth for temporal decoding (Fig. 1a). During recordings, GEMINI writes the real-time cytoplasmic level of reporter subunits as concentric fluorescent patterns. The intensity of signal bands in GEMINI varies with the amplitude of the recorded activities. The incorporation of multiple timestamps allows derivation of the growth profile of individual particles, thus decoding the absolute chronological information of signals.

We first created in cellulo assemblies by computational design and multi-level screening (Extended Data Fig. 1a). A hierarchical assembly strategy was adopted, in which two-component protein cages were docked into 3D lattices[27] (Fig. 1b) and three distinct lattices were generated (Extended Data Fig. 1b). Rosetta-predicted mutations were introduced at critical residues in the lattice interfaces to expand the library. Assembly was assessed by mixing purified and concentrated subunit solutions in vitro. Of more than 30 variants screened in vitro, five were found to nucleate at 0.5 M NaCl or lower and therefore evaluated further in cellulo (Extended Data Fig. 1c).

The variants were then expressed in human embryonic kidney (HEK) 293T cells via transient transfection. The two subunits of each variant were linked by self-cleaving P2A for equimolar expression (Supplementary Fig. 1a). For visualization, a subunit tagged with fluorescent protein was co-expressed in a low fraction (approximately 5%) to generate fluorescent assemblies. All variants showed clear intracellular precipitants within 72 h of transfection (Fig. 1c and Extended Data Fig. 1d), demonstrating the efficacy of the design and screening strategies. These assemblies exhibited faceted morphologies, indicating ordered microscopic structures.

In cellulo recording favours scaffolds with early nucleation and, ideally, only one assembly per cell. Lattice #3-v2 and #3-v3 were excluded as they formed multiple assemblies per cell. Among the remaining candidates, lattice #1-v2 exhibited an early and synchronous nucleation, and was therefore chosen. Of note, it also nucleated efficiently in multiple mammalian cell lines, highlighting its broad applicability (Supplementary Fig. 1b). Although the other variants were not explored further, they still hold promise for recording in specific contexts after optimization.

## The growth behaviours of GEMINI scaffolds

Clonal HEK lines encoding both the blank and the timestamp components of GEMINI were generated, in which GEMINI expression was controlled by doxycycline (DOX) inducible expression (Tet-ON)[28] (Fig. 1d and Supplementary Fig. 2). The timestamp subunit was constructed by fusing HaloTag (HT) to the N terminus of the A-chain[29], which exhibited the lowest effect on assembly, as evidenced by its low equilibrium concentration in the cytoplasm (Supplementary Fig. 3a,b). The same terminus was later used to construct the reporter subunit. This low equilibrium concentration of subunits is critical for achieving sharp transitions of timestamps and distinguishing closely spaced events (Supplementary Fig. 3c,d). Upon DOX addition, GEMINI began nucleating in approximately 4 h and plateaued at approximately 10 h within this cell line (Fig. 1e and Supplementary Video 1), where most cells grew one

in their cytoplasm (Supplementary Fig. 4). The rapid and synchronous nucleation enables concurrent recording from a large cell population.

Information storage requires structurally stable GEMINI with minimal subunit exchange, which was assessed by successively labelling particles with distinct HT ligands (HTLs) and monitoring changes in the band sharpness and position over time. To achieve this, a fluorophore-free HTL (dark-HTL) was first added to bind to the initial HT at induction, creating a dim core in all particles. Sequential staining began at 6 h after induction using membrane-permeable Janelia Fluor (JF) conjugated HTLs in the order of $JF_{669}$, $JF_{608}$ and $JF_{552}$ at 6-h intervals[30,31]. After the last switch, GEMINI grew for an additional 6–50 h before fixation and imaging. The sharpness (Fig. 1f) and position (Fig. 1g) of the colour transition bands remained consistent across groups, indicating minimal exchange between subunits in GEMINI and those unbound. The dimensions of GEMINI continued to increase over time, indicating continuous lattice extension (Fig. 1g).

To temporally decode signals, GEMINI growth needs to be modelled accurately by timestamps. Although linear recorders are advantageous for decoding[24], we reasoned that nonlinear growth of 3D scaffolds could achieve comparable, if not better, temporal resolution.

We first monitored growth over 48 h. The midplane projection of each particle was determined via timelapse imaging, and growth profiles were resolved using a custom single-particle tracking algorithm (Supplementary Video 2). As expected, 3D growth resulted in a nonlinear increase in GEMINI radius. We then established a model to describe the growth, in which a constant-rate subunit addition onto GEMINI was assumed (Supplementary Fig. 5). An equation was derived that linked the radius ($r$) of particles with time ($t$): $r = (Kt + A)^{1/3}$, where $K$ and $A$ are constants. The model was examined by fitting the mean linearized growth profile (defined as volume index) from population average of representative particles, showing an $R^2$ of 0.995 (Fig. 1h), whereas the fitting of growth profiles from individual particles yielded a mean $R^2$ of 0.982 (Fig. 1i). Minor deviations from linearity were observed near the end, probably due to reduced accuracy in capturing the true midplane as particles grew larger, and a possible increase in assembly energy resulting from defect accumulation. Fluctuations in individual growth profiles were attributed to tracking inaccuracies and the out-of-plane motion of particles during imaging, rather than intrinsic structural instability. These results demonstrate the high accuracy of the model in describing GEMINI growth.

Scalable and automated signal readout is crucial, necessitating fluorescent pattern retrieval regardless of recorder orientation. Granular recorders, with their isotropic growth, are well-suited for this goal. GEMINI maintains octahedral throughout various growth stages, indicating uniform addition of subunits onto {111} facets (Extended Data Fig. 1e–j). To assess recording at various directions, we created GEMINI with 11 thin bands by rapidly switching dyes (Fig. 1j). For a typical particle, fluorescence profiles measured along six distinct orientations were well aligned, demonstrating recording consistency in individual GEMINI. By contrast, linear recorders encoded information in a single orientation, and a slight tilt from the focal plane deteriorated imaging quality (Extended Data Fig. 2a). In tissue-mimicking 3D culture, many linear recorders were oriented out of plane, posing challenges on scalable readout (Extended Data Fig. 2e). Although volumetric imaging can map out-of-plane recorders with minimal effect on decoding precision, it substantially lowers throughput (Extended Data Fig. 2f–h). Collectively, these results demonstrate that GEMINI has desired growth behaviours for a scalable intracellular recorder.

## GEMINI minimally impacts cell processes

An ideal recorder should negligibly perturb essential cell processes. We first assessed the effect of GEMINI on survival of HEK cells using a live/dead assay (Supplementary Fig. 6). After growing GEMINI for 4 days, no apparent increase in cell death was observed, indicating

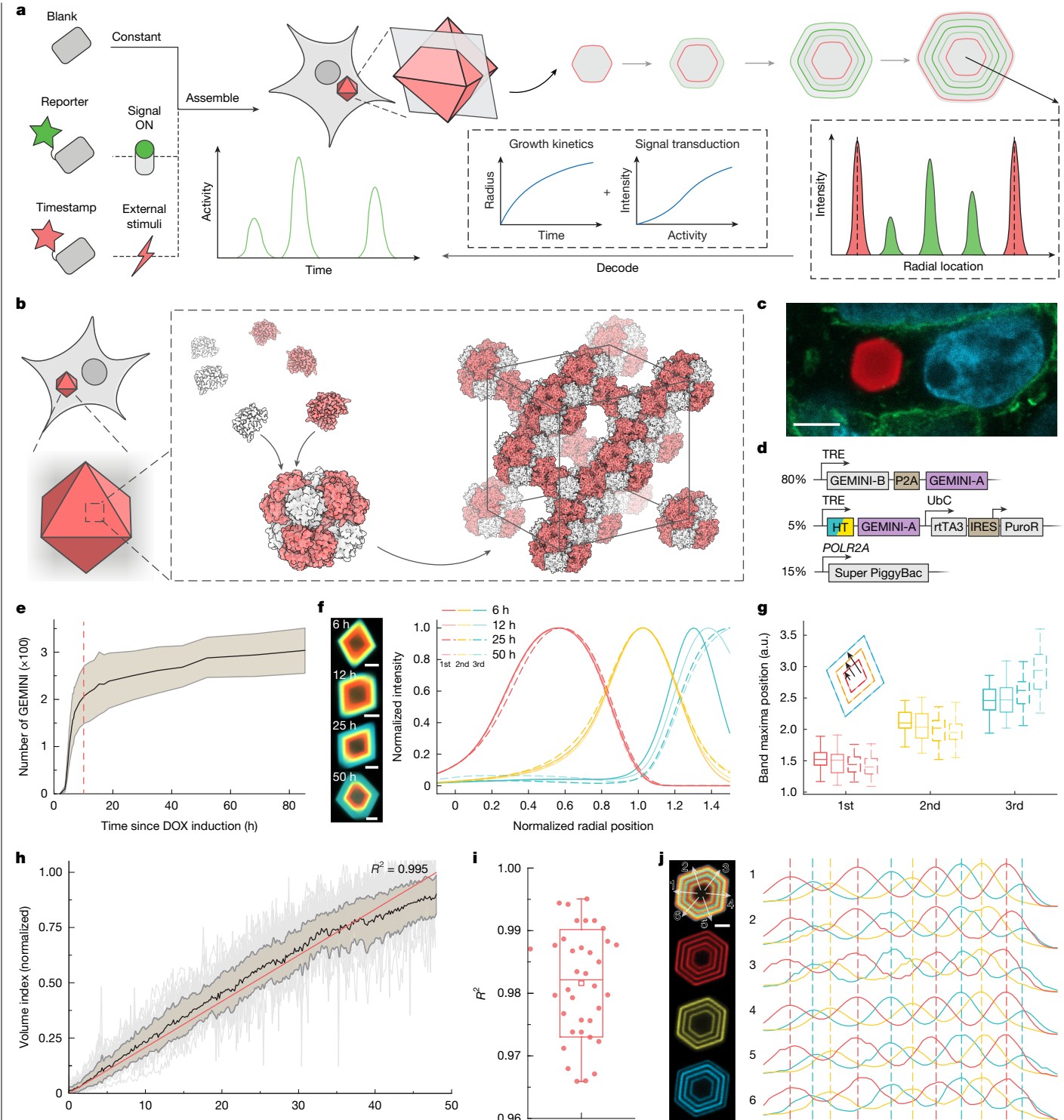

**Fig. 1 | Concept and characterization of GEMINI. a**, Schematic of the GEMINI recording principle, in which blank, timestamp and reporter components assemble within live cells, forming fluorescence patterns that encode the cell activity history. **b**, Hierarchical assembly of GEMINI in live cells. Two subunits first assemble into protein cages, which further stack into an extendable lattice. **c**, Image showing GEMINI particle growth in HEK293T cells. GEMINI particles are in red, nuclei are in cyan and the cell membrane is in green. Scale bar, 5 μm. **d**, DNA constructs for the development of the clonal HEK cell line expressing GEMINI. **e**, Number of GEMINI particles in the clonal culture over time after DOX induction. The black line denotes the mean, and the shaded area indicates ±s.d. The red dashed line indicates the time when GEMINI nucleation reaches a plateau. **f**, Images (left) and mean normalized fluorescence profiles

(right) of GEMINI particles incubated in live cells for various durations before fixation, following identical dye-switch protocols ($n = 54, 50, 52$ and $62$ particles for 6, 12, 25 and 50 h, respectively). Scale bars, 2 μm. **g**, Band positions of GEMINI particles in **f**. a.u., arbitrary units. **h**, Growth profiles of GEMINI particles in HEK cells ($n = 35$ particles). The black line denotes the mean, the shaded area indicates the s.d. and fitting is shown in red. **i**, $R^2$ values from the fits of the growth profiles of individual particles ($n = 35$ particles). **j**, Image of a GEMINI particle with 11 distinct thin bands (left) and the alignment of fluorescence profiles taken from different directions (right). Scale bar, 2 μm. For the boxplots, the box bounds denote the 25th and 75th percentiles, the whiskers indicate the minimum and maximum, the squares represent the mean and the centre lines show the median (**g,i**).

its low cytotoxicity (Supplementary Fig. 6c–e). Next, we profiled the morphological features of subcellular structures via the Cell Painting assay[32]. Although most compartments exhibited minimal structural changes, a decrease in the nuclear area and a subtle increase in the mitochondrial area were observed at 48 h after expression (Supplementary Fig. 7a–e), which could be due to potential physical interactions and changes in confluency. However, these changes were not observed in U2OS cells (Supplementary Fig. 7f–j), suggesting that distinct cell types may respond differently to GEMINI.

Moreover, we examined cell proliferation by tracking cell density over time. Comparable proliferation rates were observed between cells with and without GEMINI nucleation (Extended Data Fig. 3a,b) and there was no apparent disruption of mitosis and cytokinesis following nucleation. After division, GEMINI entered one of the daughter cells (Extended Data Fig. 3c). By contrast, cytoplasmic nucleation of linear recorders such as iPAK4 disrupts proliferation by preventing cytokinesis (Extended Data Fig. 3e,f).

## Temporal resolution of GEMINI recording

We next investigated whether GEMINI can temporally map cellular events. To assess decoding accuracy and resolution, we first introduced an artificial signal by adding an HTL dye that mimicked the appearance of biomolecules in cytoplasm (Fig. 2a), in which the addition time represents the ground truth of signal onset[24]. Before recording, cells were pre-incubated in a dark-HTL, generating a dim core in GEMINI. The first switch (yellow) occurred 8 h after induction, serving as the first timestamp ($t = 0$). The signal dye (violet) was added at $t = 2$–8 h to different groups, followed by a final switch at $t = 11$ h as the second timestamp (blue). Cells were incubated for another 11 h before fixation, and over 200 particles per group were batch imaged and analysed.

For all resolved patterns, signal onsets were consistently bordered between the two timestamps, demonstrating the ability of GEMINI to capture event order (Fig. 2b). Fluorescence profiles from the groups were normalized and compared, in which the signal onsets appeared at expected locations (Fig. 2c), indicating preservation of temporal information. Using our growth-prediction model, decoded onsets from individual particles agree closely with the ground truths (Fig. 2d), with the timing standard deviation (s.d.) within the sub-hour range. We further analysed time error between decoded onsets and ground truths, in which 75.9% of the single-particle decoding results fell within ±1 h of the ground truths, 98.2% within ±2 h and 100% within ±3 h, showing precise temporal decoding at the single-particle level (Supplementary Fig. 8). It is noteworthy that we only deployed two timestamps here. Even higher temporal resolution can be expected with more timestamps.

As the radius of GEMINI grows nonlinearly, we expect lower temporal resolution if signals are recorded at later stages. To assess this, we initiated recording 48 h after expression and recorded for another 36 h (Extended Data Fig. 4a). Despite a much larger core and slower radial growth (Extended Data Fig. 4b), a timing s.d. of 2–4 h was still achieved (Extended Data Fig. 4d), and most particles (92.9%) still exhibited a time error of no more than 6 h (Extended Data Fig. 4f), showing the capability of GEMINI to record at an extended window.

## GEMINI resolves physiological signals

We then assessed whether physiological events could be captured via activity-dependent expression of the reporter subunit. This approach directly reflects signalling cascade output, exhibits a large dynamic range, and is modular and easily adaptable to report diverse pathways[33,34].

As a proof of concept, we recorded transcriptional dynamics mediated by NF-κB signalling, exploiting a synthetic promoter that combines tandem repeats of the NF-κB response element with a minimal promoter ($P_{min}$) to drive reporter-subunit expression[11] (Supplementary Fig. 9a). We first examined orthogonality between GEMINI and NF-κB signalling, where the phosphorylation of IκBα, a key element in the NF-κB signalling cascade, was quantified (Supplementary Fig. 10). Comparable responses to tumour necrosis factor (TNF) were found with and without GEMINI. A modest increase in basal IκBα level and its phosphorylation state was observed in GEMINI-expressing cells, which may indicate a low-level cellular stress associated with particle formation. Regarding the reporter-subunit expression, the basal level in cytoplasm is low, which is increased by approximately sevenfold after 12 h of incubation in TNF (Supplementary Fig. 11).

We then developed a clonal HEK line encoding complete GEMINI machinery for DOX-induced recording of NF-κB signals (NF-κB-GEMINI line; Fig. 2e). Upon DOX induction, dark-HTL was added to obtain a dim core, followed by switching to an HTL dye (yellow) 6 h after induction as the first timestamp ($t = 0$). TNF was then added at $t = 3$, 6 or 9 h to separate groups (green), together with another HTL dye (violet) as an accompanying timestamp. At $t = 12$ h, all groups were switched to the third HTL dye (blue) as the last timestamp, and cells were incubated for another 8 h before fixation (Fig. 2f). In GEMINI images (Fig. 2g) and the mean normalized fluorescence profiles (Supplementary Fig. 12a) from the groups, the NF-κB onset appeared at the anticipated locations, slightly later than the accompanying timestamp, presumably due to transcription and translation delays. We then temporally decoded both NF-κB activation and its accompanying timestamp. The mean decoded times for timestamps aligned with ground truths, indicating the minimal effect of NF-κB activation on GEMINI growth and decoding accuracy. The decoded times for NF-κB activation were $4.82 ± 1.35$, $7.72 ± 1.59$ and $10.71 ± 1.18$ h for $t = 3$, 6 and 9 h, respectively, showing a consistent approximately 2-h delay to TNF induction. Of note, the temporal distribution of NF-κB signals was broader than that of the second timestamps, which may reflect the heterogeneity in cellular response to TNF.

We reasoned that signal deactivation is also resolvable, as particles served as a reservoir for reporter-subunit uptake, thus accelerating their removal from the cytoplasm. To test this, we designed a similar recording course but having TNF constitutively applied while emulating the deactivation by its removal (Fig. 2i). In this test, the first (yellow) and last (blue) timestamps marked $t = 0$ and 10 h, respectively, and TNF was removed from different groups at $t = 4$, 6 and 8 h (green), with an accompanying timestamp (violet) marking TNF removal (Fig. 2j). Accurate timing of the deactivation onset (NF-κB peaks) was achieved (Fig. 2k and Supplementary Fig. 12b), similarly showing an approximately 2-h delay to TNF removal ($6.04 ± 1.15$, $7.75 ± 2.06$ and $9.84 ± 2.28$ h for $t = 4$-h, 6-h and 8-h groups, respectively) due to extended reporter expression with undegraded mRNA.

## GEMINI resolves dynamics and amplitudes

The ability of GEMINI to resolve both activation and deactivation presents an opportunity to map signalling dynamics that involve repetitive cycles. To examine this, we simulated NF-κB dynamics to three TNF exposures over 40 h (Fig. 3a), applying timestamps with each addition and removal. All events were captured by individual particles (Fig. 3b), accurately mapping the ON/OFF times (Fig. 3c).

We then determined the fastest resolvable dynamics by measuring the separation of two NF-κB peaks with various time intervals. After expressing GEMINI for 8 h, cells were incubated successively in two TNF exposures for 8 h each, with decreasing intervals from 12 to 1 h (Fig. 3d). At intervals of 6 h or longer, the two bands exhibited minimal overlap. Although merging as the interval decreased, two distinct bands were still evident at 1-h interval.

Although capturing the hour-level dynamics is prominent, some signalling events exhibit faster dynamics and necessitate better resolution. We hypothesized that this could be achieved by accelerating the turnover of cytoplasmic reporter subunit and its mRNA (Supplementary Fig. 13a) that reset the system more rapidly for capturing subsequent events.

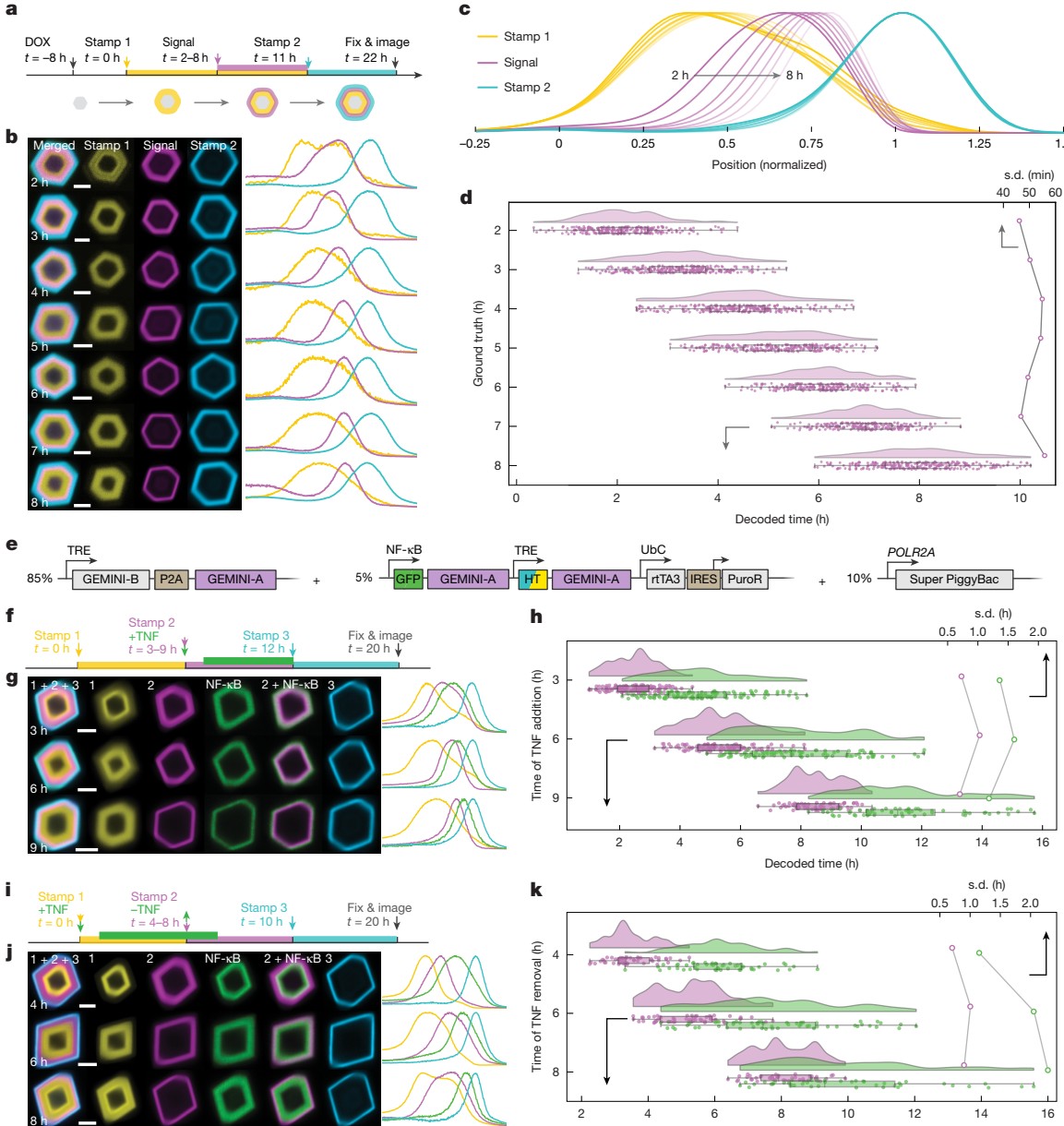

**Fig. 2 | In cellulo recording using GEMINI. a**, Experimental procedure for testing the precision of the temporal decoding. The onset of signal-mimetic dye-addition events (violet) is decoded using two timestamps: yellow ($t = 0$ h) and blue ($t = 11$ h). **b**, Images (left) and fluorescence profiles (right) of GEMINI particles with signals introduced at $t = 2–8$ h. Scale bars, 2 μm. **c**, Mean normalized fluorescence profiles of the timestamp and signal channels. **d**, Decoded onsets (bottom $x$ axis, $n = 258, 275, 279, 261, 225, 219$ and 275 particles for $t = 2, 3, 4, 5, 6, 7$ and 8 h, respectively) and the s.d. (top $x$ axis) for each group. **e**, DNA constructs for the development of the NF-κB-GEMINI lines. **f**, Experimental procedure for recording NF-κB upregulation in response to TNF (green). A dye is added with TNF as stamp #2 (violet). Both NF-κB activation and stamp #2 are decoded using stamp #1 (yellow, $t = 0$ h) and stamp #3 (blue, $t = 12$ h). **g**, Images (left) and fluorescence profiles (right) of GEMINI

particles recording NF-κB activation at $t = 3, 6$ and 9 h. Scale bars, 2 μm. **h**, Decoded times (bottom $x$ axis) and s.d. (top $x$ axis) for NF-κB activation (green) and stamp #2 (violet; $n = 106, 79$ and 47 particles for $t = 3, 6$ and 9 h, respectively). **i**, Experimental procedure for recording NF-κB deactivation after TNF removal (green). A dye is added as stamp #2 (violet) when TNF is removed. NF-κB deactivation and stamp #2 are decoded using stamp #1 (yellow, $t = 0$ h) and stamp #3 (blue, $t = 10$ h). **j**, Images (left) and fluorescence profiles (right) of GEMINI particles recording NF-κB deactivation at $t = 4, 6$ and 8 h. Scale bars, 2 μm. **k**, Decoded times (bottom $x$ axis) and s.d. (top $x$ axis) for NF-κB deactivation (green) and stamp #2 (violet) from individual GEMINI particles ($n = 50, 59$ and 45 particles for $t = 4, 6$ and 8 h). For the boxplots, the box bounds denote the 25th and 75th percentiles, the whiskers indicate the minimum and maximum and the centre lines show the median (**d**,**h**,**k**).

Here we incorporated the P1N4 domain to destabilize both the reporter and its mRNA[35]. A corresponding clonal HEK293T line (NF-κB-GEMINI-Boost) was established (Supplementary Fig. 13b) that resolved distinct signalling events separated by as short as 15 min (Fig. 3e).

We next investigated whether the signal amplitude could be quantified by band intensity. A dose–response test was performed at TNF doses ranging from $10^{-5}$ to 60 ng ml$^{-1}$, with a TNF-free group included

for comparison (Fig. 3f). The NF-κB peaks, normalized by HTL-staining intensity, measured signal amplitude. GEMINI reported a dose of $10^{-4}$ ng ml$^{-1}$, two orders of magnitude lower than the detection limit of the cytoplasmic transcriptional reporter (Supplementary Fig. 11b). GEMINI recording exhibited a dynamic range of approximately 60-fold, over eight times the cytoplasmic benchmark. The improved detection limit and dynamic range can be attributed to concentrating

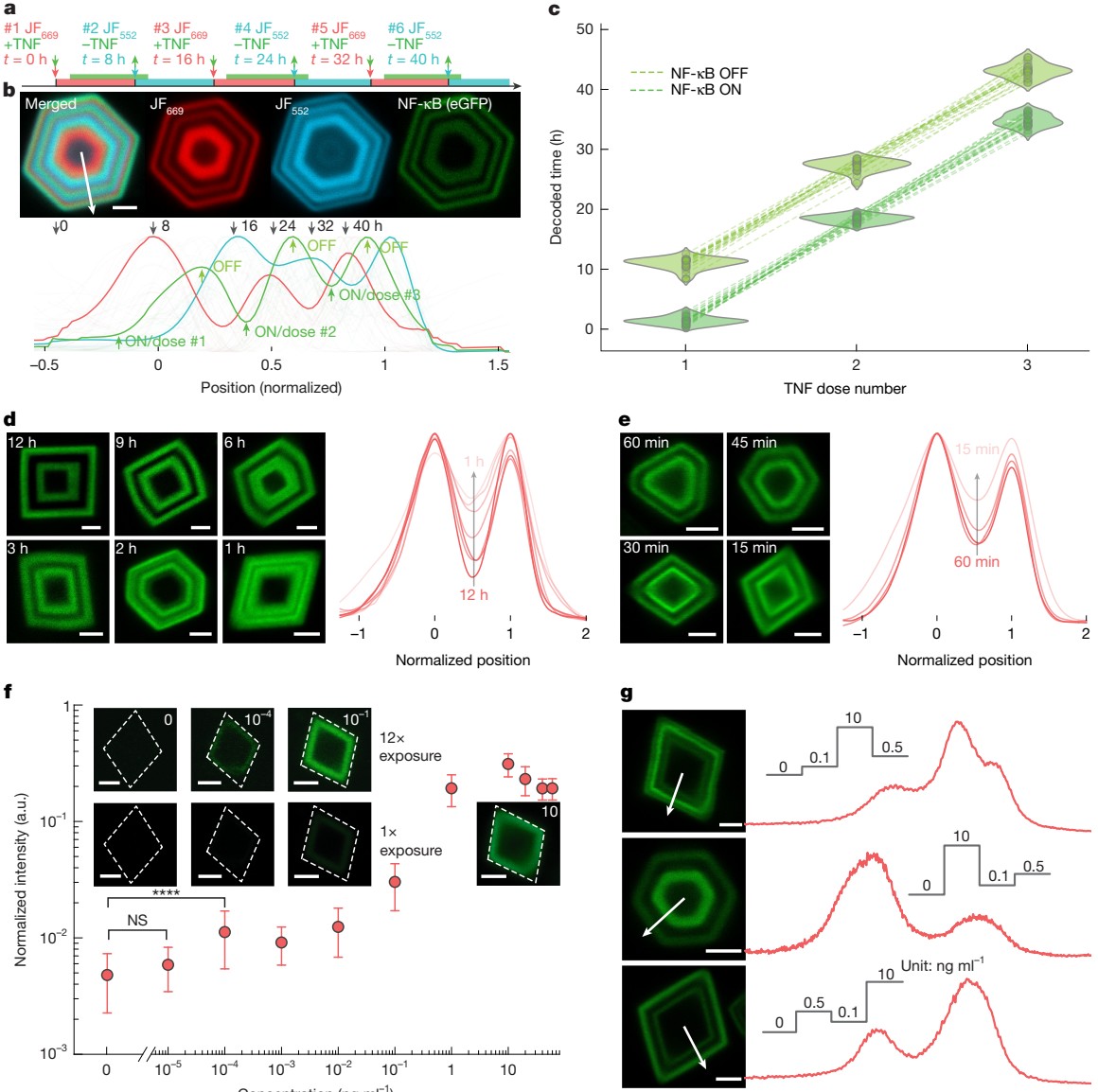

**Fig. 3 | Recording multiple events and dynamics in live cells. a**, Experimental procedure for multi-event recording. Three ON–OFF cycles were introduced, with a stamp applied at each TNF addition and removal. **b**, Image (top) and the mean normalized fluorescence profiles (bottom) of the timestamp and signal channels. Scale bar, 2 μm. **c**, Decoded times for ON and OFF events during each TNF dose ($n = 21$ particles). **d**, GEMINI particles encoding dual NF-κB activation signals with decreasing intervals without destabilization (left; 12 to 1 h) and the mean normalized fluorescence profiles (right; $n = 15, 17, 13, 16, 8$ and 8 particles for 12, 9, 6, 3, 2 and 1 h, respectively). **e**, GEMINI particles encoding boosted dual NF-κB activation signals with decreasing intervals with destabilization (left; 60 to 15 min) and the mean normalized fluorescence profiles (right; $n = 18, 17, 19$

and 20 particles for 60, 45, 30 and 15 min, respectively). Scale bars, 2 μm. **f**, Dose-dependent intensity of signals recorded by GEMINI ($n = 28, 21, 40, 22, 41, 32, 48, 44, 35, 31$ and 37 particles for $0, 10^{-5}, 10^{-4}, 10^{-3}, 10^{-2}, 10^{-1}, 1, 10, 20, 40$ and 60 ng ml$^{-1}$ groups, respectively). The circles denote the mean and the whiskers indicate s.d. Significance was determined using a two-sided Welch's $t$-test: not significant (NS) $P > 0.5$ and ****$P < 0.0001$. The insets show images of single GEMINI particles capturing signal amplitudes at various doses. The images in the top row are the same particles as the bottom, but with a 12× exposure. **g**, GEMINI particles (left) recording varied signal amplitudes and the corresponding fluorescence profiles (right). TNF doses of 0, 0.1, 0.5 and 10 ng ml$^{-1}$ were applied in distinct sequences. Scale bars, 2 μm.

effect of GEMINI that condenses and amplifies cytoplasmic signals. We then recorded trains of stimuli with differential amplitudes of low (0.1 ng ml$^{-1}$), medium (0.5 ng ml$^{-1}$) and high (10 ng ml$^{-1}$) doses at various sequences. Cells were incubated in each concentration for 8 h (Fig. 3g). In representative particles, distinct fluorescence plateaus were observed in the expected order, showing the ability of GEMINI to resolve varying signal amplitudes.

## GEMINI maps inflammatory signals in vivo

To evaluate whether GEMINI could record physiologically relevant events in vivo, we first established a xenograft model to monitor

inflammation-induced NF-κB activation in immunodeficient mice[11], achieved by subcutaneously implanting NF-κB-GEMINI cells. Inflammation was induced through intraperitoneal injection of lipopolysaccharide (LPS), a well-established method for triggering acute inflammatory responses[36]. Upon stimulation, host-derived cytokines, including TNF, circulate and diffuse into the xenograft, activating NF-κB signalling in the GEMINI-expressing cells.

Ten days after implantation, when the xenograft had developed into a palpable tumour, DOX was administered to initiate GEMINI expression (Fig. 4a). LPS was then administered at a dose of 3 mg kg$^{-1}$ 6 days after DOX induction (designated as day 0), and tumours were obtained on day 4 for GEMINI (HTL dye) and vasculature (anti-CD31) labelling.

Widespread GFP signal, indicative of NF-κB activation, was observed throughout the tumour tissue (Fig. 4b and Extended Data Fig. 5). Of note, GFP intensity exhibited substantial spatial variation, reflecting heterogeneity in the cellular response.

We next tested whether GEMINI could resolve inflammation levels in vivo. A range of LPS concentrations (0.03–3 mg kg$^{-1}$) was administered 6 days after induction, with a control group receiving equivolume saline. GFP intensity (normalized to HTL staining) from individual particles revealed a clear dose-dependent response (Fig. 4c), confirming the ability of GEMINI to encode inflammation levels.

To resolve absolute timing, we developed an in vivo timestamping strategy. Although the in vitro dye-switch approach is not directly applicable, we leveraged the high bioavailability of several HTLs that enables timestamping via retro-orbital injection[31]. Owing to the rapid clearance of HTLs from circulation, each administration produced a transient pulse that labelled the instant surface of a particle with a narrow band (Extended Data Fig. 6). When applying two timestamps 48 h apart, distinct bands were observed in GEMINI in vivo (Fig. 4d,e).

We then examined the ability of GEMINI to retrospectively decode the timing of inflammation. After initiating GEMINI expression, mice were divided into three groups, receiving LPS on days 0, 1 or 2, respectively. Timestamps were applied on days 0 and 2 for temporal decoding (Fig. 4f). As expected, later LPS administration corresponded to an outer position of GFP bands relative to the timestamps (Fig. 4g). Temporal decoding of individual particles showed that signals peaked approximately 1 day after LPS injection (Fig. 4h). Compared with in vitro measurements, the timing showed larger variation within each group, which may reflect heterogeneous cytokine diffusion within tumours, differential cellular responses to inflammatory cues and lower precision of in vivo timestamping.

Differences in local vascularization could influence cytokine accessibility and therefore the timing of NF-κB activation. Indeed, histological examination revealed substantial variation in vascular density across regions (Fig. 4i,j). To test this, we decoded NF-κB signals from regions with high and low vascular density, defined by the volume occupancy of more than 8% and less than 2%, respectively. A noticeable delay in NF-κB signalling was observed for the poorly vascularized region (Fig. 4k), in which the mean peak time was approximately 1.8 days after LPS injection compared with 0.7 days for the well-vascularized region (Fig. 4l).

## Implement GEMINI in the mouse brain

To assess the performance of GEMINI in native tissues, we created a single adeno-associated viral vector (AAV) encoding both blank and timestamp subunits (btAAV) and delivered this to mice via intracranial injection (Extended Data Fig. 7a). GEMINI nucleated efficiently across brain regions (Extended Data Fig. 7b), with particles appearing as early as day 5 and growing continuously (Extended Data Fig. 7c). Similar to in vitro growth, most neurons nucleated a single particle (Extended Data Fig. 7d). Segmentation enabled accurate registration of each particle to its corresponding neuron in the tissue (Extended Data Fig. 7e–h).

To evaluate biocompatibility, we assessed neuronal and inflammatory responses following GEMINI expression. Histological analysis of hippocampal CA1 from days 7 to 14 after injection revealed no significant changes in neuronal density or astrocyte reactivity despite increased GEMINI density (Extended Data Fig. 8a,b). By day 14, high-density GEMINI particles were consistently observed in regions receiving btAAV, such as the cortex and hippocampus (Fig. 5a). Histological analyses revealed no significant differences from sham-injected controls in neuronal density or astrocyte reactivity (Fig. 5b and Extended Data Fig. 8e–g). Furthermore, we analysed morphological features of GEMINI-expressing neurons, focusing on soma and nucleus (Extended Data Fig. 9a–g). Although soma sphericity and nucleus volume remained unchanged, there was a slight increase in soma volume and a modest decrease in nucleus sphericity (Extended Data Fig. 9d–g). These subtle changes may reflect the occupation of cytoplasmic space and nucleus deformation by some large particles.

To assess possible functional impact, we performed in vivo two-photon calcium imaging in *Thy1*-jRGECO1a transgenic mice injected with btAAV in the primary visual cortex (Extended Data Fig. 9h–l). GEMINI$^+$ neurons exhibited calcium transient patterns and $\Delta F/F$ of calcium responses comparable with GEMINI$^-$ neurons, suggesting minimal effect on neuronal firing.

We then evaluated the effect of GEMINI on animal behaviours. btAAV was injected bilaterally to regions involved in motor control and memory, including the primary motor cortex, dorsal hippocampus and ventral hippocampus, and behavioural tests were performed on day 14. In open-field tests, no differences in total travelling distance, maximal speed or average speed were observed between GEMINI$^+$ and sham groups (Fig. 5c–f). In the Y-maze test, mice from both groups spent comparable duration in the novel arms, indicating that GEMINI minimally impacts short-term memory (Fig. 5g,h).

To further evaluate fine motor coordination, we unilaterally injected btAAV into the primary motor cortex. A positive control group expressing diphtheria toxin subunit A (dtA) and a negative control group receiving saline were included (Extended Data Fig. 10a). Horizontal-ladder-rung walking tests were performed on day 14 (Extended Data Fig. 10b,c). GEMINI-expressing mice performed comparably with saline controls, whereas dtA-expressing mice exhibited longer passing time and impaired coordination (Extended Data Fig. 10d,e). Post-mortem analysis revealed preserved brain symmetry in GEMINI mice, whereas dtA mice showed mild shrinkage in the injected hemisphere (Extended Data Fig. 10f). Histological analysis further confirmed minimal neuronal loss associated with GEMINI expression (Extended Data Fig. 10g,h).

## GEMINI maps neuronal history in animals

We first examined whether systemic timestamping is effective in the mouse brain. The blood–brain barrier-crossing HTL dyes, including JF$_{669}$ and JF$_{552}$, were used in this study[37,38]. The dyes, when injected 24 h apart (Fig. 5i), afforded two sharp peaks in the expected order, confirming successful timestamping (Fig. 5j,k).

We then recorded transcriptional histories in the mouse brain. We first tracked DOX-mediated transcription using the Tet-ON system, where a second AAV encoding constitutively expressed rtTA3G and TRE-promoter-driven reporter subunit was co-injected (Supplementary Fig. 14a). At 7 days after injection, the first timestamp was applied (JF$_{669}$, $t = 0$), followed by intraperitoneal administration of DOX at 0, 12 and 24 h. The second stamp was introduced at 24 h using JF$_{552}$ (Fig. 5l). Band positions coincided with the timing of DOX induction, indicating that temporal information was captured (Fig. 5m and Supplementary Fig. 14c). The signals exhibited consistent delay in peak timing, as DOX persisted in the tissue for an extended period[39]. Differences among the groups were resolvable (Fig. 5n), despite a much broader distribution than recordings in culture and xenograft model.

We then asked whether GEMINI could temporally resolve activity-dependent immediate early gene (such as *Fos*) transcription[40]. To test this, we constructed a signal transduction circuit jointly controlled by neural activity and DOX (Supplementary Fig. 14b). Recording was initiated by DOX and neurons were activated at various times by pentylenetetrazole, a GABA$_A$ receptor antagonist that induces seizures across brain regions[41]. Two timestamps were applied with a 24-h interval and seizure was induced at either 0 or 24 h (Fig. 5o). The individual particles (Fig. 5p) and the mean normalized fluorescence profiles (Supplementary Fig. 14d) showed expected band positions corresponding to the timing of induction. Statistical analysis revealed significance

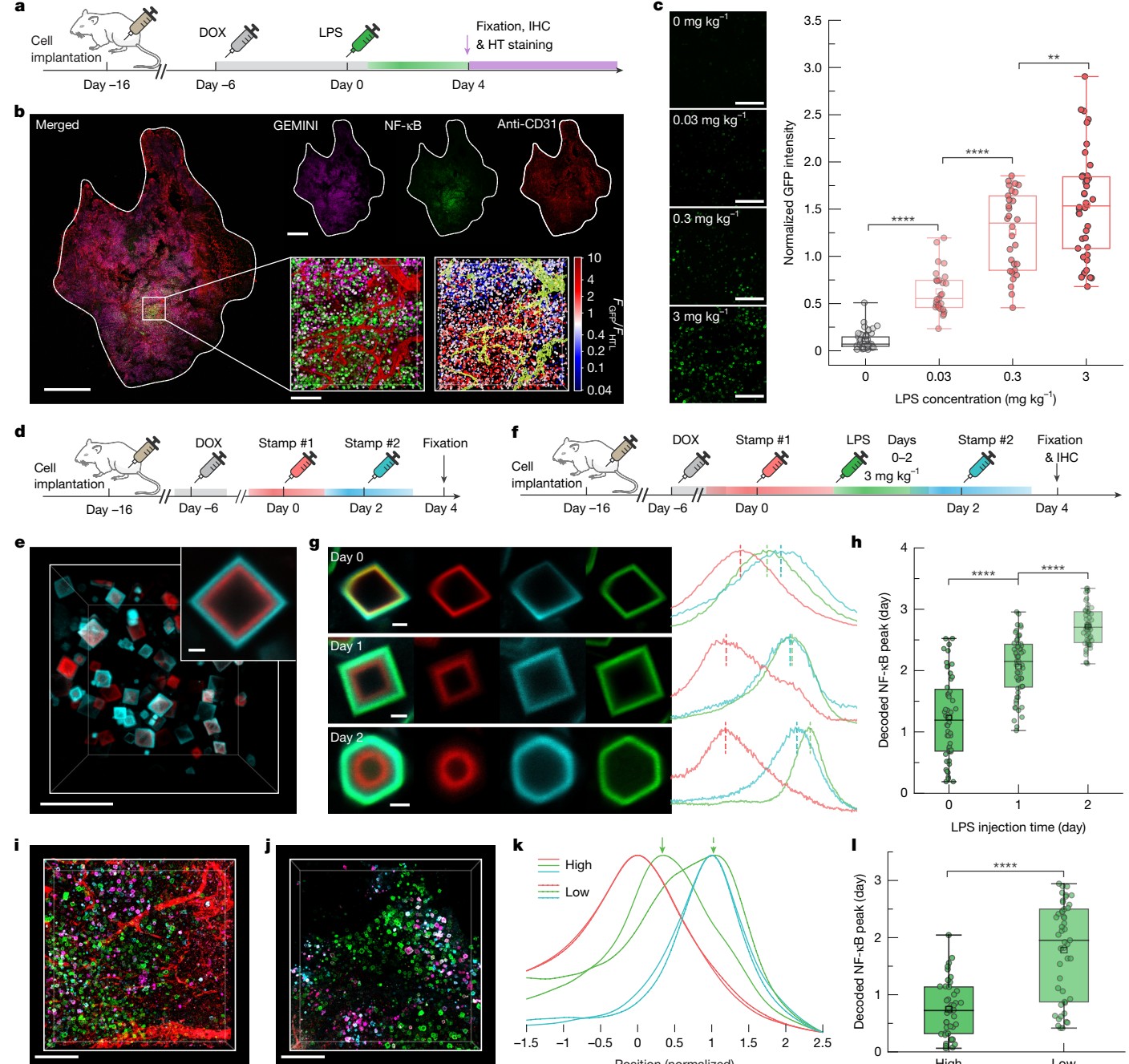

**Fig. 4 | In vivo recording of inflammation. a**, Experimental procedure for in vivo recording of LPS-induced inflammation. IHC, immunohistochemistry. **b**, Section of a HEK293T xenograft showing GEMINI recording of NF-κB activation following systemic injection of 3 mg kg⁻¹ LPS. GEMINI is in violet, NF-κB in green and vasculature (anti-CD31) in red. The insets show a magnified 3D image (left) and the corresponding heatmap displaying the normalized GFP intensity ($F_{GFP}/F_{HTL}$) of individual particles (right). Green in the heatmap represents vasculature segmentation. Scale bars, 1 mm and 100 μm (insets). **c**, Images of the GEMINI recording NF-κB signal at various LPS doses (left) and the dose-dependent signal intensity (right; $n$ = 61, 27, 31 and 42 particles for 0, 0.03, 0.3 and 3 mg kg⁻¹ LPS, respectively). Scale bars, 100 μm. **d**, Experimental procedure for in vivo timestamping via systemic injection of HTL dyes. **e**, Image of GEMINI particles in xenografts labelled with two timestamps; the inset shows the magnified image of a GEMINI particle. Scale bars, 20 μm and 2 μm (inset). **f**, Experimental procedure for recording the NF-κB signal with LPS administered at various times. Schematic of the syringes in panels **a**,**d**,**f** adapted from ref. 24,

Springer Nature America. **g**, Images (left) and fluorescence profiles (right) of GEMINI particles recording NF-κB activation in mice injected with LPS on days 0, 1 or 2. Scale bars, 2 μm. **h**, Decoded timing of NF-κB activation peaks in groups receiving LPS on days 0–2 ($n$ = 52, 59 and 55 particles for $t$ = 0, 1 and 2 days, respectively). **i**,**j**, 3D images of xenograft tissue with high (**i**) and low (**j**) vascular density. LPS was administered on day 0. Scale bars, 100 μm. **k**, Mean normalized fluorescence profiles of NF-κB signals recorded in GEMINI particles located in regions with high (solid) and low (dashed) vasculature density. **l**, Decoded peak timing of NF-κB activation for GEMINI particles from high-vascularization and low-vascularization regions ($n$ = 42 and 47 particles for high and low, respectively). For the boxplots, the box bounds denote the 25th and 75th percentiles, the whiskers indicate the minimum and maximum, the squares represent the mean and the centre lines show the median (**c**,**h**,**l**). Significance was determined using one-way analysis of variance (ANOVA) with Tukey's test (**c**,**h**) and two-sided Welch's $t$-test (**l**): **$P$ < 0.01 and ****$P$ < 0.0001.

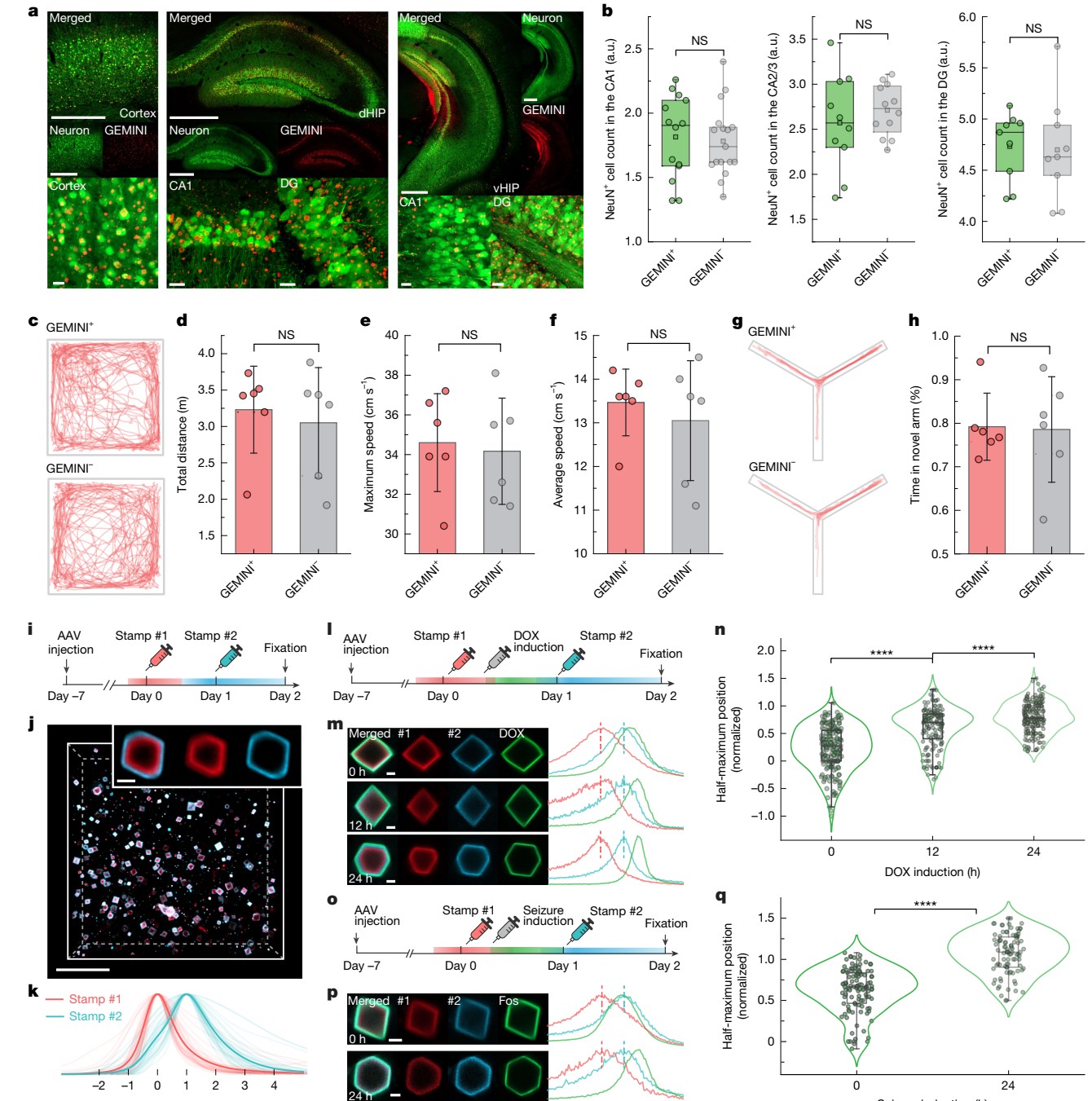

**Fig. 5 | Implementing GEMINI to the mouse brain. a**, Images showing high-level GEMINI expression in the cortex (left), dorsal hippocampus (dHIP; middle) and ventral hippocampus (vHIP; right). Scale bars, 500 and 20 μm for low and high magnification, respectively. DG, dentate gyrus. **b**, Comparison of the neuronal density (NeuN+ cells) between GEMINI+ or GEMINI− groups (CA1: n = 14 and 17; CA2/3: n = 10 and 12; DG: n = 9 and 9 images for GEMINI+ and GEMINI− groups, respectively, from 5 mice per group). **c**, Traces of GEMINI+ or GEMINI− mice in an open-field arena. **d**–**f**, Comparison of travelling distance (**d**), maximal speed (**e**) and average speed (**f**) between GEMINI+ or GEMINI− groups. **g**, Traces of GEMINI+ or GEMINI− mice in a Y-maze. **h**, Comparison of time spent in the novel arm between the GEMINI+ or GEMINI− groups. **i**, Experimental procedure of in vivo timestamping in the brain. **j**, Images of GEMINI in the mouse brain with two timestamps applied. The inset shows a magnified image of a GEMINI. Scale bars, 50 and 2 μm for low and high magnification, respectively. **k**, Mean normalized fluorescence profiles of stamps #1 and #2 (n = 36 particles). **l**, Experimental procedure of in vivo recording of DOX-induced expression. **m**, Images (left) and fluorescence profiles (right) of GEMINI that encode

DOX-mediated expression signal induced at 0 (top), 12 (middle) and 24 h (bottom). Scale bars, 2 μm. **n**, Normalized radial positions of half-maximum intensity in GEMINI that encode signals induced at 0, 12 and 24 h (n = 179, 150 and 187 particles for 0, 12 and 24 h, respectively). **o**, Experimental procedure of in vivo recording of seizure-induced *Fos* activation. Schematic of the syringes in panels **i**,**l**,**o** adapted from ref. 24, Springer Nature America. **p**, Images (left) and the fluorescence profiles (right) of GEMINI that encode the *Fos* activation signal induced at 0 (top) and 24 h (bottom). Scale bars, 2 μm. The coloured vertical dashed lines in **m** and **p** indicate the peak positions of the corresponding bands. **q**, Normalized radial positions of half-maximum intensity in GEMINI that encode signals induced at 0 and 24 h (n = 111 and 80 particles for 0 and 24 h, respectively). For the boxplots, the box bounds denote the 25th and 75th percentiles, the whiskers indicate the minimum and maximum, the squares represent the mean and the centre lines show the median (**b**,**n**,**q**). Six mice per group were used, and the bars denote the mean and the whiskers show s.d. (**d**–**f**,**h**). Significance was determined using two-sided Welch's t-test (**b**,**d**–**f**,**h**,**q**) and Kruskal–Wallis analysis with Dunn's test (**n**): NS P > 0.05 and ****P < 0.0001 (**b**,**d**–**f**,**h**,**n**,**q**).

between the two groups (Fig. 5q). It is noteworthy that these analyses utilize single-particle readouts to quantify population-level trends. A true atlas-like map of neuronal dynamics at the single-cell level is still not attainable at the current stage, largely due to variations in decoding. We further examined whether the seizure induction could be recorded by driving reporter-subunit expression directly using the *Fos* promoter. Genes for direct and DOX-mediated signal reporting were co-expressed for comparison and seizure induction was accompanied by a timestamp (Supplementary Fig. 15a,b). Seizure-induced activation was clearly resolvable in both methods, whereas notable background signals were found before seizure induction in the direct-reporting channel, probably due to background neural activity (Supplementary Fig. 15c). Together, these studies demonstrate that GEMINI can assess diverse cellular activities in native tissue.

## Discussion

In cellulo recording using engineered protein assemblies provides new opportunities for resolving cell dynamics. Here we expanded the toolbox of intracellular assemblies through computational design. The obtained scaffold is relatively compact, with its subunits (GEMINI-A/B: 19.0/13.4 kDa) comparable in size or even smaller than several cellular-history reporters[6,7].

GEMINI exhibits distinct advantages to linear recorders: it introduces less mechanical perturbation and allows orientation-independent read-out. By contrast, linear recorders often stretch cells, potentially activating mechanosignalling[42] and complicating in vivo implementation. Moreover, their uniaxial elongation imposes challenges in retrieving signals from tilted recorders. The in vivo compatibility and scalable signal retrieval of GEMINI offer an attainable path to organ-wide mapping.

To map physiologically relevant signals, we utilized transcriptional dynamics as a proxy for their corresponding cascades. This modality has been broadly used in fluorescent and luciferase transcriptional reporters, readily transferrable to GEMINI[33,34]. A potential limitation involves the noticeable delay of signals with respect to the actual events. Nevertheless, delays were quantifiable and thus could be systemically subtracted. GEMINI also permits the employment of other reporting mechanisms that act on proteins directly to eliminate delays[43].

Compared with other sensing modalities, GEMINI can be implemented with instrumentation available in most biological laboratories. Thus, it is a powerful alternative to existing reporter assays while preserving additional spatiotemporal details. Moreover, GEMINI could complement real-time fluorescent biosensors for time-resolved cellular tracking[3], especially when hour-level resolution is sufficient or real-time imaging is not accessible.

GEMINI can provide valuable insight into intricate cell dynamics when deployed in vivo, where timestamping is crucial but challenging, especially in the brain as the blood–brain barrier excludes many potential timestamping molecules[44]. Our systemic timestamping method represents a notable advance. Building on this, we showed the potential of GEMINI for tissue-wide recording of signalling history, revealing pronounced cellular heterogeneity in NF-κB activation in response to systemic inflammation, and highlighted heterogeneous vascularization as an important contributing factor. This modality, combining engineered cells and their transplantation, holds promise for studying oncogenic signalling and evaluating therapeutic responses. The temporal resolution in the brain, however, was lower than that in culture or xenografts, owing to factors such as inherent heterogeneity of neurons, less-accurate timestamping, variability in GEMINI expression and less-accurate decoding model. Disentangling these factors and resolving spatiotemporal maps of single-cell activities without populational analysis requires further efforts on homogenous expression and precise signal decoding. Continued optimization would help to transform GEMINI into a platform broadly applicable to spatiotemporally resolve cellular histories in animals, enabling deeper insight into the mechanisms underlying health and disease.

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

## Methods

### Protein assembly design

The design of protein assemblies followed the method we have recently reported[27]. In brief, validated protein cages were docked to target the $F4_132$ and $I432$ space groups. Dockings were performed in PyMOL by alignment and translations. For the docking of the $F4_132$ space group, two tetrahedral cages (TET-1 and TET-2) were aligned with every $C2$ symmetry axis along the $x$, $y$ and $z$ coordinate axes and centre of mass at the origin of the coordinate. While keeping TET-1 fixed at the origin, TET-2 was rotated along the $z$ axis for $90°$ and translated along the $[1, 1, -1]$ direction to dock with TET-1. The hexamer between the two cages with $D3$ dihedral symmetry was extracted and aligned with its $C3$ axis along the $z$ axis and its $C2$ axis along the $x$ axis and with its centre of mass at the origin. The $I432$ space group was docked between two octahedral cages (OCT-1 and OCT-2), where OCT-2 was translated only without any rotation. In both cases, the D3 assembly units contained the crystal contact and were used for the Rosetta sequence design. In the RosettaScripts framework, input monomers were symmetrized into D3 dihedrals, and then we sampled the interface distances by translation along the $z$ axis (dihedral axis) within $±1.5$ Å of the docked conformation without a rotational degree of freedom. For each translation, the dihedral interface was designed with rigid protein backbones by packing a rotamer with layer design restrictions at interacting residues. Then, designs were filtered by ddG < 0, solvent-accessible surface area > 200, clash check ≤ 2 and unsatisfied hydrogen bonds ≤ 2 before being visually inspected for hydrophobic packings. The schematics of protein structures were rendered using Protein Imager[45].

### Protein expression and purification

Synthetic genes were purchased from Integrated DNA Technologies (IDT) as plasmids in the pET29b vector with a hexahistidine affinity tag. Plasmids were cloned into BL21* (DE3) *Escherichia coli*-competent cells (Invitrogen). Single colonies from agar plate with 100 mg l$^{-1}$ kanamycin were inoculated in 50 ml of Studier autoinduction media, and the expression continued at 37 °C for over 24 h. The cells were obtained by centrifugation at 4,000$g$ for 10 min, and resuspended in 35 ml lysis buffer of 300 mM NaCl, 25 mM Tris pH 8.0 and 1 mM PMSF. After lysis by sonication and centrifugation at 14,000$g$ for 45 min, the supernatant was purified by Ni$^{2+}$ immobilized metal affinity chromatography with Ni-NTA Superflow resins (Qiagen). Resins with bound cell lysate were washed with 10 ml (bed volume of 1 ml) of washing buffer (300 mM NaCl, 25 mM Tris pH 8.0 and 60 mM imidazole) and eluted with 5 ml of elution buffer (300 mM NaCl, 25 mM Tris pH 8.0 and 300 mM imidazole). Eluted proteins were further purified by size-exclusion chromatography in 150 mM NaCl and 25 mM Tris pH 8.0 on a Superose 6 Increase 10/300 gel filtration column (Cytiva), before concentrated by 10 K concentrators (Amicon).

### Library generation of mutated protein assembly

For structurally validated protein assemblies, their computational design models and X-ray models were further used for more extensive cycles of Rosetta sequence design, where diverse, mutated sequences were generated from fixed interface backbones. The new design models were visually inspected for their differences in the side-chain interactions from the validated designs. Alternative sequences near the core of the interfaces were selected either individually or as groups to build mutation libraries for in vitro screening of GEMINI scaffolds.

### In vitro protein assembly screening

Protein assembly subunits with Rosetta-predicted mutations were screened by batch crystallization experiments in vitro, aiming at assembly conditions with ionic strength lower than 500 mM NaCl. Equimolar ratios of protein subunits were mixed at 5–50 μM concentration in 150 mM, 300 mM and 500 mM NaCl and 25 mM Tris pH 8.0. Crystallization results were inspected by optical microscopy imaging for up to 2 weeks.

### Cloning and molecular biology

GEMINI constructs were cloned into a PiggyBac plasmid backbone (Addgene #104454) for in vitro study and clonal cell line development, and into an AAV plasmid backbone (Addgene #100854) for AAV packaging and in vivo study. Plasmids were constructed using standard Gibson assembly. In brief, the vector was linearized by double restriction enzyme digestion and purified by the GeneJET gel extraction kit (Thermo Fisher Scientific). DNA fragments were obtained by PCR amplification and then combined with the linearized backbones by Gibson ligation. Genes for the protein assembly variants were obtained by DNA synthesis (Twist Bioscience). The *Fos* promoter was cloned from Addgene #47907; the NF-κB reporter promoter was cloned from Addgene #82024; rtTA3G was cloned from Addgene #120309; HT was cloned from #177882; GFP was cloned from Addgene #177881; and mCherry was cloned from Addgene #81041. A 6×GGS linker was used to connect the HT or fluorescent proteins with the assembly subunits. All plasmids were verified by whole-plasmid sequencing or Sanger sequencing around the cloned regions. The key plasmids used in this work are listed in Supplementary Table 1 and are available on Addgene. The amino acid sequences of protein variants that we developed to assemble in cellulo are listed in Supplementary Table 2.

### Cell culture and transfection

HEK293T and other mammalian cells (gifts from T. Inoue and K. Konstantopoulos, originally acquired from the American Type Culture Collection) were grown and passed following standard protocols as previously described[24]. Cells at low passage number (less than 15 passages) were seeded at a confluence of 30% onto 35–100-mm dishes coated with gelatin (STEMCELL Technologies) or 14-mm glass-bottom dishes (D35-14-1.5-N, CellVis) coated with 0.1 mg ml$^{-1}$ poly-D-lysine (P6407, Sigma-Aldrich). Dulbecco's modified Eagle medium (DMEM; Corning) supplemented with 10% FBS (VWR) and 1% penicillin–streptomycin (Millipore Sigma) was used as the culture medium. All cells were maintained at 37 °C in a humidified atmosphere of 5% CO$_2$. When cells reached 50–70% confluence, genes were delivered using 25 K linear polyethylenimine (Polysciences) following the protocol previously reported[46] or Lipofectamine 3000 transfection kit (Thermo). The cells were then further incubated at 37 °C and 5% CO$_2$ for approximately 8–24 h before experiments. For expressing in cellulo assemblies (including GEMINI and iPAK4) through transient transfection, plasmids encode untagged subunits and the fluorescent proteins:HT-tagged subunits were co-transfected at a molar ratio of approximately 10:1.

### Clonal cell line development and in vitro tests

The clonal HEK cell lines with stable integration of the gene of interest were generated using the PiggyBac transposon system[47]. In brief, HEK293T cells were co-transfected with the PiggyBac transposon plasmids carrying the gene of interest and a puromycin-resistant gene, and a Super PiggyBac transposase expression vector (NovoPro). Polyethylenimine (1 mg ml$^{-1}$) was complexed with the plasmids for transfection. At 48 h after transfection, cells were selected using 2 μg ml$^{-1}$ puromycin (A1113803, Gibco). Selection was maintained for 1–2 weeks to ensure stable integration. Clonal cell lines were then established by limiting dilution, where single cells were plated into 96-well plates and allowed to expand. Clones were screened to confirm the presence and expression of the transgene. Positive clones were further functionally validated, and the selected colony was expanded, maintained under selection conditions and cryopreserved for future use. Without special note, 2 μg ml$^{-1}$ DOX (D-500-1, Goldbio) was utilized to induce the GEMINI expression in the clonal cell lines, and 10 ng ml$^{-1}$ TNF (Z03333-10, GenScript) was utilized to activate the NF-κB signalling in culture.

## AAV production and purification

AAVs were produced and purified following the method previously reported[46]. In brief, HEK293T cells were transfected with the AAV cargo plasmid of interest, helper plasmid and the AAV-PHP.eB rep-cap plasmid (Addgene #103005). Cells were incubated for 48–72 h after transfection before collecting. Viral particles were isolated from the cell pellet and supernatant via freeze–thaw lysis or SAN digestion and clarified by centrifugation. Purification of AAV was carried out using iodixanol gradient ultracentrifugation to ensure the purity of viral particles. Viral titres were determined through quantitative PCR. The AAVs were concentrated into a titre of $1 \times 10^{14}$ viral genomes ml$^{-1}$ before use. The final AAV preparations were stored at 4 °C for short-term use (less than 1 month) or −80 °C for long-term storage.

## Confocal imaging

Confocal imaging was performed using Zeiss LSM780, LSM800 or LSM980 microscopes. Lambda scan mode (LSM780 and LSM980) was used to image GEMINI particles with multi-colour labelling. The excitation laser wavelengths for various fluorescent proteins or dyes were: 405 nm (Hoechst 33342), 488 nm (GFP, YFP and JF$_{525}$), 561 nm (mCherry, JF$_{552}$ and JF$_{608}$) and 633 nm (JF$_{669}$). Single crystals were acquired with ×63 oil immersion objectives. In each Lambda scan, a 32-channel QUASAR detector was utilized to simultaneously acquire a hyperspectral stack of images with distinct collection wavelengths in the range of 410–695 nm. The multispectral images were then unmixed with the built-in linear unmixing algorithm in ZEISS Zen Blue and/or Zen Black software. Images of individual fluorescent labels were acquired in the same instrumental configuration as the references for linear unmixing. The quality of spectral unmixing was assessed by examining the residual signals after unmixing.

## Timelapse imaging

Cells expressing the target constructs were grown on glass-bottom culture dishes (CellVis) and imaged under a Zeiss Axio Observer wide-field epifluorescence microscope at 37 °C and 5% $CO_2$. Focus across frames was automatically calibrated by a Definite Focus 3 module.

## Vertebrate animals

Adult (4–8 weeks) female athymic nude mice J:NU (007850, Jackson Laboratory) were used in the transplantation of HEK293T xenografts. Adult (5–10 weeks) CD-1 (022, Charles River Laboratories) or C57BL/6 (000664, Jackson Laboratory) mice were used throughout the other study. The mice were housed at $22 \pm 1$ °C with humidity ranging from 30% to 70% and on a 12-h light–dark cycle. All animal experiments were conducted in strict accordance with the guidelines and regulations set forth by the Johns Hopkins University Animal Care and Use Committee. Experimental protocols were reviewed and approved by the Animal Care and Use Committee under protocol numbers MO21E409 and MO24E325.

## Stereotaxic surgeries and in vivo GEMINI expression

Before surgery, mice were anaesthetized with 4% isoflurane in air (induction) and maintained at 1–2% isoflurane in air throughout the procedure. Carprofen (5 mg kg$^{-1}$, subcutaneous) was administered as preoperative analgesia. During surgery, the animal body temperature was monitored and maintained at 37 °C using a heating pad. Once anaesthetized, mice were secured in a stereotaxic frame (Stoelting) and bilateral openings (1 mm in diameter) were made on the skull at the corresponding coordinates. A Nanoliter2020 Injector (WPI) was utilized to inject AAVs at a rate of 100 nl min$^{-1}$. Without special note, 100 nl AAV was injected into each position, and a dose of $1 \times 10^{12}$ viral genomes per mouse. After each injection, the microcapillary needle was held in place for 3 min to allow for the diffusion of AAVs into the tissue before moving to the next position. When the injection was completed,

the scalp was sealed with Vetbond (3 M), and mice were continuously monitored until recovery. The following coordinates were used to target specific brain regions (relative to bregma): primary motor cortex: anteroposterior +1.0 and +1.4, mediolateral ±1.5 and dorsoventral −0.3 to −1.0 (0.1 increments); dorsal hippocampus: anteroposterior −1.8 and −2.2, mediolateral ±1.5 and dorsoventral −1.2 to −1.8 (0.1 increments); and ventral hippocampus: anteroposterior −2.8 and −3.2, mediolateral ±2.8 and dorsoventral −2.8 to −4.0 (0.1 increments). Without special note, the in vivo expression of GEMINI was induced by intraperitoneal injection of 40 mg kg$^{-1}$ DOX.

## In vivo timestamping via retro-orbital dye injection

In vivo timestamps were applied via retro-orbital injection of blood–brain barrier-permeable HTL dyes. In brief, 100 nmol of a HTL dye (JF$_{669}$ or JF$_{552}$, gift from the Lavis lab, Howard Hughes Medical Institute) was first dissolved in 100 µl DMSO with 10% Pluronic F127. Of sterile saline, 400 µl was further added to yield the HTL solution with a final concentration of 200 µM. Before injection, the mouse was anaestheized with isoflurane (3% induction and 1.5% during injection). The level of anaesthesia was assessed by toe pinch. The mouse was then positioned, and the eyelid of the target eye was gently retracted to expose the globe. A syringe with 31-G needle containing 50 µl (xenograft) or 100 µl (mouse brain) of the HTL dye solution was carefully inserted into the retro-orbital space with the bevel down, ensuring that the needle tip was within the venous sinus. The dye solution was then injected steadily and slowly to avoid damaging the eye and the surrounding tissues. After injection, the mouse was allowed to recover from anaesthesia in the home cage, with close monitoring for any signs of discomfort or complications.

## HEK293T xenograft model

For the in vivo NF-κB-GEMINI xenograft model, the NF-κB-GEMINI clonal cells were resuspended by phosphate-buffered saline (PBS) and mixed with Matrigel (356231, Corning) in a 1:2 volume ratio. Cells ($5 \times 10^6$) were planted subcutaneously in the flank regions of J:NU athymic nude mice. After implantation, tumour size was monitored and all the mice were randomly assigned into different groups. Tumour size was ensured no larger than 2 cm in all dimensions, as approved by Johns Hopkins University Animal Care and Use Committee. Typical tumours in this study were smaller than 1 cm in all dimensions. In vivo crystal growth was initiated by 40 mg kg$^{-1}$ DOX administrated via intraperitoneal injection every 2 days starting from 10 days after implantation. LPS (437627, Sigma) was administered via intraperitoneal injection at designated time points. A dose of 3 mg kg$^{-1}$ was utilized without special note.

## Open-field tests

The open-field test was conducted in a 38 × 38 cm arena with 38-cm high walls. Mice were acclimated to the testing room for 1 h before the test. When the test began, each mouse was placed in the centre of the arena and allowed to explore for 7 min. Animal travelling within the arena was recorded by video. The arena was cleaned with 70% ethanol three times between sessions to eliminate scent cues. Data were analysed offline with in-house Python codes, where the total travelling distance and average travelling speed were analysed as indicators of the motor function. All procedures were in accordance with institutional and national animal care guidelines.

## Y-maze tests

The Y-maze test was conducted using a Y-shaped maze with three arms (each 35 cm long and 5 cm wide, with 10-cm high walls). Mice were acclimated to the testing room for 1 h before the test. Each mouse was placed at the end of the initial arm and allowed to explore the maze for 10 min with one arm (the novel arm) blocked. The mouse was returned to the home cage for 20 min before re-entering the maze for 5 min, when all arms were accessible. The second entrance was recorded by video. The maze was cleaned with 70% ethanol three times between sessions to

eliminate scent cues. Data were analysed offline with in-house Python codes. The duration the mouse spent in the novel arm versus the alternative arm was analysed. All procedures adhered to institutional and national guidelines for animal care and use.

## Seizure model
Pentylenetetrazole (18682, Cayman) was utilized to induce seizure in mice. Pentylenetetrazole was first dissolved in sterile saline at 9 mg ml$^{-1}$, which was delivered to mice via intraperitoneal injection at a dose of 50 mg kg$^{-1}$. The behaviour of mice was carefully monitored for 1 h after seizure induction. The level of seizure was assessed by the Racine scale[48], in which only mice that showed a seizure level of 5 or higher were utilized in further study.

## Immunohistochemistry
To prepare the brain samples for immunohistochemistry, mice were first euthanized with $CO_2$ flow of 50% of cage volume per minute for 10 min, and transcardially perfused with 40 ml of ice-cold PBS, followed by 40 ml of 4% paraformaldehyde in PBS. After decapitation, the brain was carefully extracted and post-fixed in 4% paraformaldehyde overnight at 4 °C. The brain was then transferred to a 30% sucrose solution in PBS for cryoprotection and stored until fully equilibrated (sank to the bottom of the tube). Once the brain was fully equilibrated, they were embedded in optimal cutting temperature (OCT) compound (Tissue-Tek) and frozen on dry ice. Coronal sections, 50 μm thick, were prepared using a cryo-sectioning machine (Leica 1850 CM). For immunohistochemistry, brain sections were first washed three times with PBS to remove any residual OCT compound. The sections were then blocked in PBS containing 0.3% Triton X-100 and 5% normal donkey serum (1:500; ab7475, Abcam) for 1 h at room temperature and washed three times with PBS. The sections were then incubated with primary antibodies anti-NeuN (1:500; GTX132974, GeneTex) and anti-glial fibrillary acidic protein (anti-GFAP; 1:500; #3670, Cell Signaling) overnight at 4 °C with gentle shaking. The sections were then washed three times with PBS and incubated for 2 h at room temperature in the dark with corresponding secondary antibodies including donkey anti-mouse IgG (H + L) Alexa Fluor 488 (1:500; A-21202, Invitrogen) and donkey anti-rabbit IgG (H + L) Alexa Fluor 546 (1:500; A10040, Invitrogen). After additional washes with PBS three times, the brain sections were stained with Hoechst 33342 (H3570, Invitrogen) for 30 min to mark all cell nuclei. The sections were mounted on glass slides with #1.5 coverslips using Aqua-Poly/Mount (18606, Polysciences) mounting media. The samples were stored in the dark at room temperature for 4–8 h to allow complete permeation of the media before imaging.

For immunohistochemistry staining of xenografts, tumours were sliced into 150 μm thickness instead. Anti-CD31 antibody (1:500; 550274, BD Biosciences) was used as primary antibody and donkey anti-rat AF647 antibody (1:500; A48272, Invitrogen) was used as secondary antibody.

## Image processing and data analysis
Python, MATLAB, ImageJ, Vision4D (arivis) and napari were used for image processing and/or visualization. Fluorescence profiles were extracted along the radial directions of GEMINI particles, perpendicular to one of the edges. The profiles were averaged over a certain width along the radial axis to reduce the noise. A low-pass filter (Butterworth) might be applied to further denoise the profiles. Baseline subtraction might be applied to remove the background intensity outside of the band regions. The profiles before and after filtering were compared to ensure no distortion to the curves. The onset of a fluorescence band was determined by the intercept of the tangent line at 50% peak intensity with the baseline. The locations of the peaks were determined by the intensity maxima.

To track GEMINI growth, we first used adaptive thresholding and Sobel filters to enhance GEMINI boundaries. After distance transform, GEMINI centres were located by a maxima-finding algorithm, whereas radius estimation based on intensity cut-offs and background thresholds helps to delineate particle regions. In the tracking phase, we used a linear programming-based approach to connect crystals across video frames. The algorithm detected particles frame by frame in reverse order, starting from the last frame, and formed tracks for each crystal based on their position, size and intensity. Using features such as centroid coordinates and area, the algorithm calculated distances between particles in consecutive frames and minimized a cost function through linear sum assignment to optimally match particles. Area and intensity differences were factored into the cost to ensure robust tracking. The tracking results were then examined, and those showing errors in tracking were excluded from further analysis. Overall, the method achieved accurate and consistent tracking for most particles. As the algorithm measured the area of the 2D GEMINI projection $A$ (in pixels), we quantified the volume of the GEMINI particles by a volume index defined as $A^{3/2}$. The index was normalized for comparing the growth rate of different particles.

## Statistics and reproducibility
Sample sizes have been specified in the figure legends. No statistical methods were used to predetermine sample size. Differences with $P < 0.05$ were considered statistically significant (*$P < 0.05$, **$P < 0.01$, ***$P < 0.001$, ****$P < 0.0001$ and non-significant). For the statistical comparison of two groups, two-sided Welch's $t$-tests were applied. Without special note, three cultures per group were included in in vitro experiments, and three mice per group were included in animal experiments. Animals were randomly assigned to experimental groups. All micrographs presented in the main text, extended data and supplementary figures are representative images from at least three independent cultures or two independent animals.

## Schematics
The schematics in this work were prepared by the corresponding author (D.L.) using Adobe Illustrator or Blender. Copyright is held by D.L. Portions of these drawings may have previously appeared in other publications by the authors[24,49].

## Materials availability
The plasmids constructed and used in this work have been deposited to Addgene (plasmid IDs: 228881–228886, 228888 and 228889).

## Reporting summary
Further information on research design is available in the Nature Portfolio Reporting Summary linked to this article.

## Data availability
Source datasets for the in vivo experiments are provide with the paper. Complete datasets for this article, including those for main figures, extended data figures and supplementary figures, are available at GitHub (https://github.com/DCLinLab/GEMINI).

## Code availability
Custom data analysis code developed and used for this project is available at GitHub (https://github.com/DCLinLab/GEMINI).

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

**Acknowledgements** We thank L. D. Lavis and his team for sharing HTL dyes; and T. Inoue and K. Konstantopoulos for sharing mammalian cell lines used in Supplementary Fig. 1. This work is supported by the US National Institutes of Health (NIH) National Institute of General Medical Sciences grant R35GM147274 (to D.L., Y.Y., J.L. and Y.W.), Department of Defense Air Force Office of Scientific Research Young Investigator Program grant FA9550-23-1-0174 (to D.L., Y.W. and H.Y.), NIH BRAIN Initiative grant R21EY035955 (to D.L. and Y.S.), David and Lucile Packard Foundation (to D.L.), Kavli Neuroscience Discovery Institute Distinguished Graduate Fellowship (to Y.Y.), Johns Hopkins Discovery Award (to D.L. and D.E.B.), the Howard Hughes Medical Institute (to D.B.), the Audacious Project at the Institute for Protein Design (to Z.L., S.W. and D.B.) and the Beckman Institute CLOVER Center at Caltech (to T.F.S.).

**Author contributions** D.L. conceived the project. D.L., Y.Y. and J.L. designed the experiments. Z.L., S.W. and D.B. designed the protein assemblies. Z.L. and S.W. performed in vitro screening of the assemblies. Y.Y. performed in cellulo screening of the assemblies. Y.Y. and J.L. designed and cloned the plasmids, performed the confocal and timelapse imaging, and developed the clonal cell lines. Y.Y., J.L., Y.W., A.Q., H.Y. and Y.S. performed other characterizations in cultured cells and acquired the imaging data. T.F.S., Y.L. and Y.Y. produced and purified AAVs. Y.Y. and D.L. performed the in vivo experiments in the mouse brain and acquired the data. J.L. performed the in vivo xenograft study and acquired the data. Y.Y., W.C., J.L. and D.E.B. collected and analysed the immunohistochemistry data. P.P., W.C. and Y.Y. performed the in vivo calcium imaging and analysis. Y.Y., D.L., Z.Z. and J.L. analysed the data and prepared the figures. D.L. and Y.Y. wrote the paper. All authors participated in the revision of the manuscript.

**Competing interests** D.L., Y.Y. and J.L. have filed a US patent application on GEMINI for in cellulo recordings. The other authors declare no competing interests.

**Additional information**
**Correspondence and requests for materials** should be addressed to Dingchang Lin.

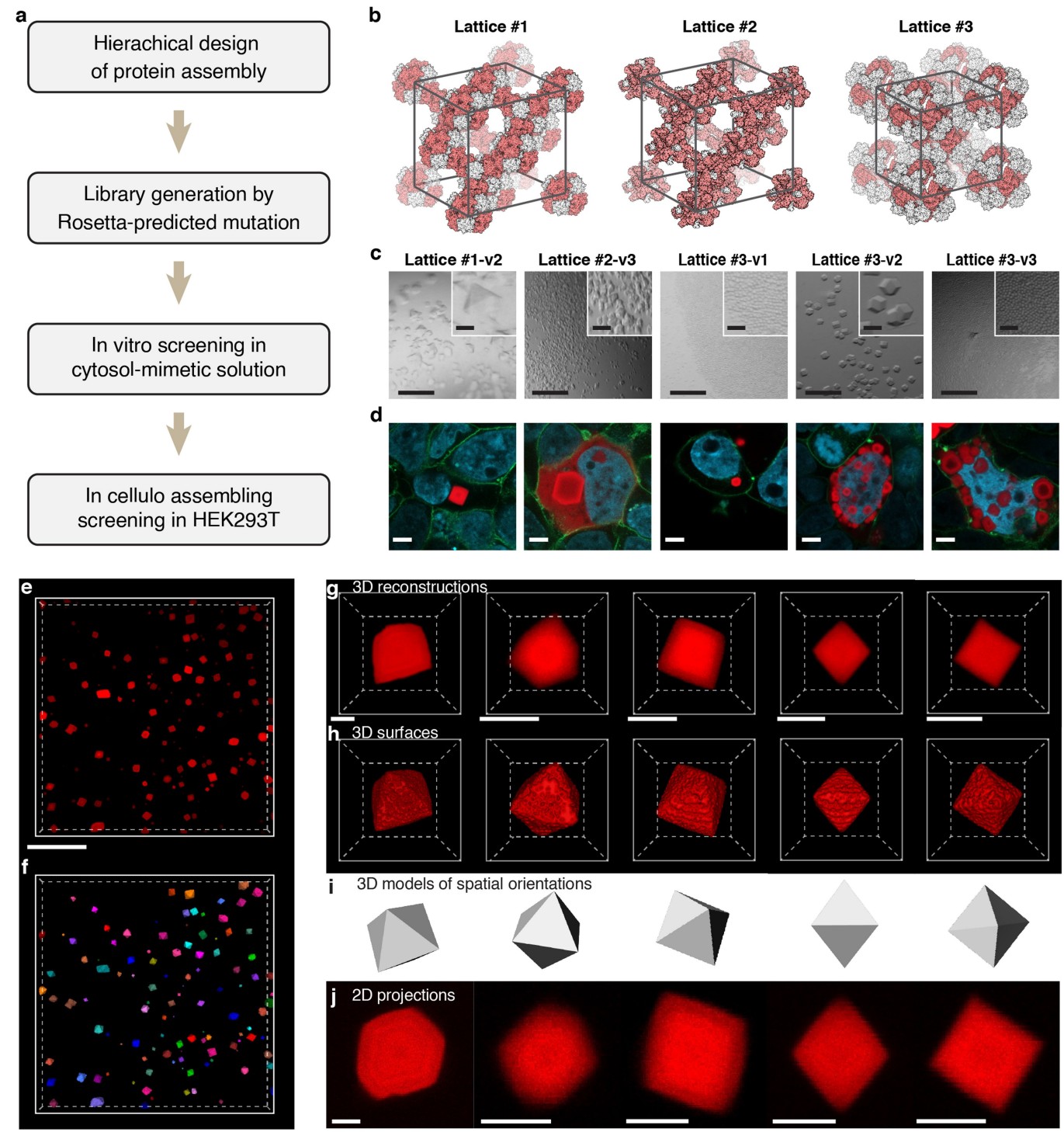

**Extended Data Fig. 1 | Development of new intracellular protein assemblies and GEMINI morphology. a**, Experimental procedure to design and select for intracellular protein assemblies suitable as the GEMINI scaffold. **b**, Lattice structures of three variants involved in this study. **c**, Images of in vitro assembly of the variants in an ion strength comparable to the cytoplasm (100-500 mM NaCl). Scale bars: 200 μm; insets: 40 μm. **d**, Images showing the assembly of variants that pass the in vitro screening, where all the variants can assemble in live HEK293T cells. Scale bars: 5 μm. **e,f**, Low-magnification 3D image of GEMINI particles grown in live HEK293T cells (**e**) and the single-particle segmentation (**f**). **g-j**, 3D-reconstruction (**g**), 3D surfaces (**h**), 3D models (**i**), and 2D projections (**j**) of individual GEMINI particles that exhibit different sizes and spatial orientations. The particles exhibit consistent octahedral shapes. Scale bars: **e**, 50 μm; **g,h,j**, 2 μm.

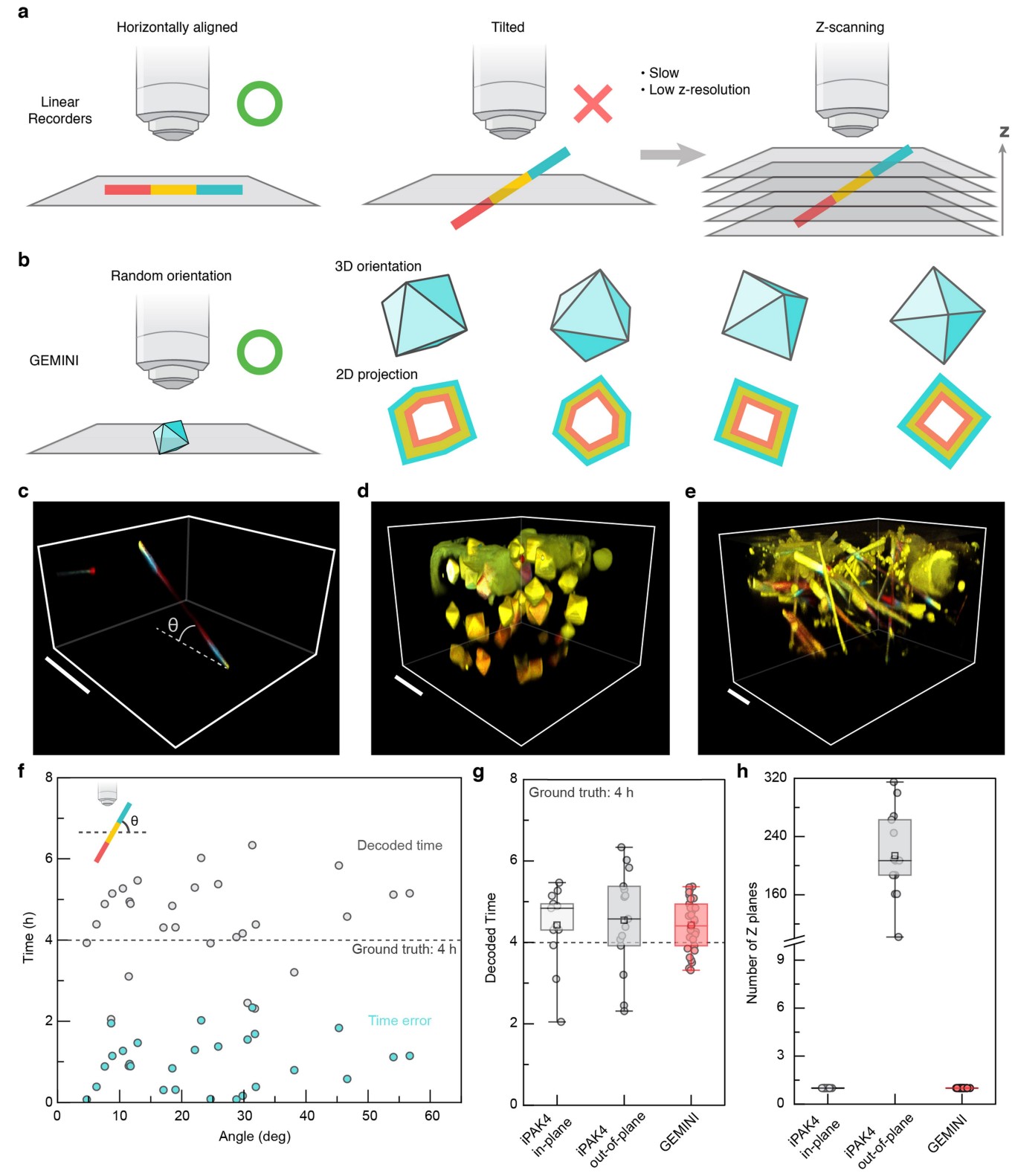

**Extended Data Fig. 2 | Comparison of linear recorders and GEMINI in signal retrieval. a**, Schematic of imaging linear recorders by fluorescence microscopy. When the recorder happens to align horizontally, they can be imaged at high resolution. However, for most recorders that are tilted out-of-plane, volumetric imaging is needed to retrieve the recorded patterns, which is slow and exhibits much lower resolution. **b**, Schematic of imaging GEMINI by fluorescence microscopy. High-resolution fluorescence patterns are retrievable regardless of the spatial orientation of GEMINI particles, allowing fast signal retrieval from a large population of cells. **c**, A linear recorder tilted at an angle in culture.

Scale bar: 50 μm. **d,e**, 3D cell culture expressing GEMINI (**d**) and iPAK4 linear recorder (**e**). Scale bars: 20 μm. **f**, The decoded time and error of iPAK4 linear recorder as a function of the tilted angle. Timestamps were applied at 0 and 8 h, while the artificial signal was introduced at 4 h. **g**, Comparison of the decoded time from in-plane iPak4, out-of-plane iPak4, and GEMINI. **h**, Average number of z-planes needed to obtain complete fluorescent profiles from individual recorders. **g,h**, n = 13/15/31 fibers or particles for iPAK4 in-plane/iPAK4 out-of-plane/GEMINI. Box bounds: the 25th/75th percentiles; whiskers: min/max; squares: mean; and center lines: median.

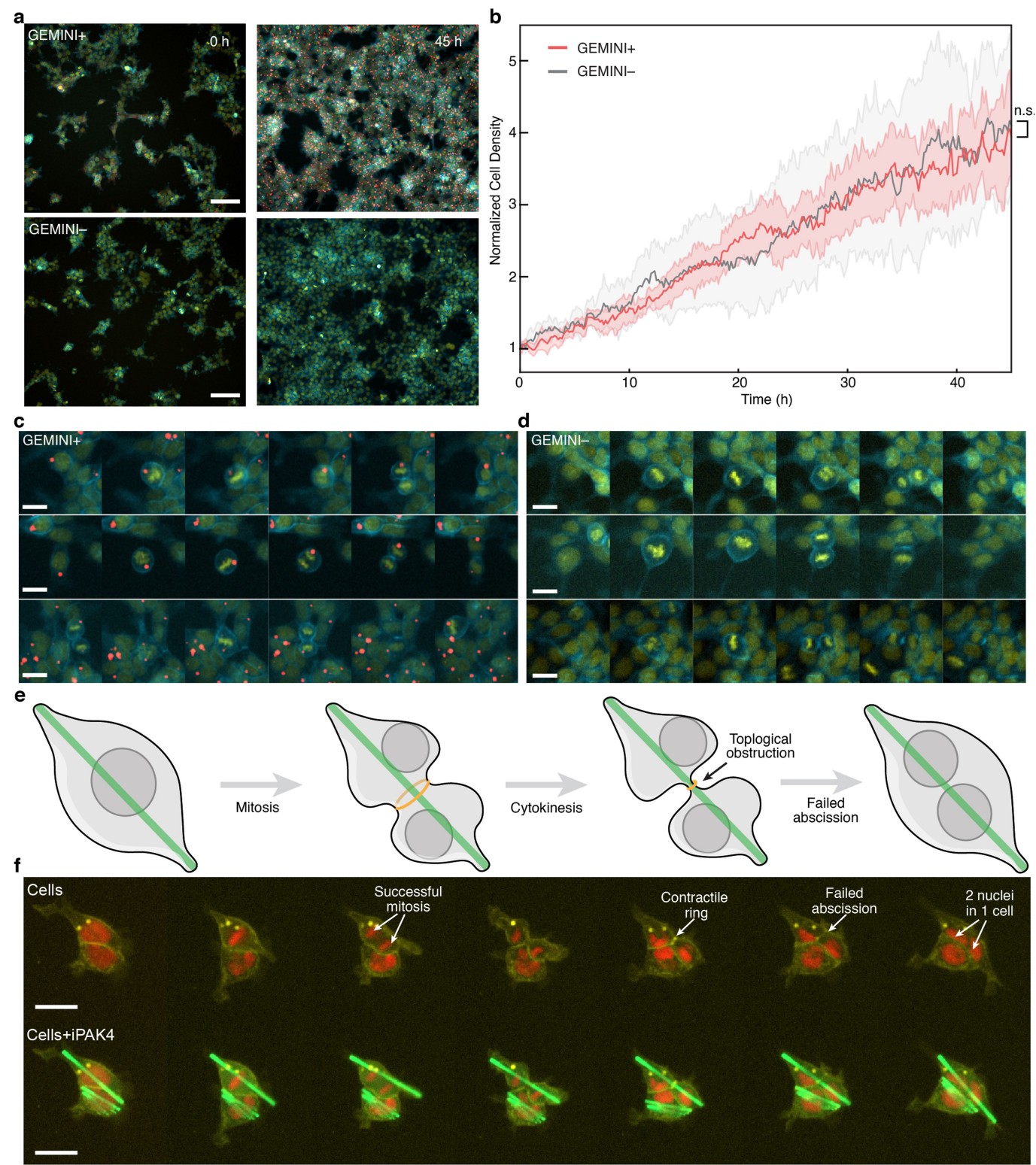

**Extended Data Fig. 3 | Impact of GEMINI and iPAK4 linear recorder on cell proliferation and division. a**, Images comparing the cell density with (top) and without (bottom) GEMINI growth at t = 0 hour (left) and 45 h (right). Scale bars: 100 μm. **b**, Comparison of the cell proliferation rate between the GEMINI+ and − groups (n = 8/5 timelapse videos for GEMINI+/− groups). Center lines: mean; shaded areas: s.d.; n.s., p > 0.05; two-sided Welch's t-test. **c,d**, Snapshots of the cell division processes of live cells with (**c**) and without (**d**) intracellular GEMINI nucleation, showing GEMINI nucleation has negligible impact on mitosis and cytokinesis. **e**, Schematic showing the topological constraint imposed by intracellular linear recorders that obstruct the abscission during cell division. **f**, Snapshots of time-lapse imaging showing HEK293T cells with an intracellular linear recorder (iPAK4) during the division processes. Though mitosis was successful, the contractile ring failed to split the cell into two daughter cells and, therefore, resulted in one cell with two nuclei after division. The cellular labels were introduced via transient transfection, together with the plasmids encoding iPAK4. Yellow: membrane (GFP-GPI), Red: nucleus (H2B-mCherry), Green: iPAK4. Scale bars: 20 μm.

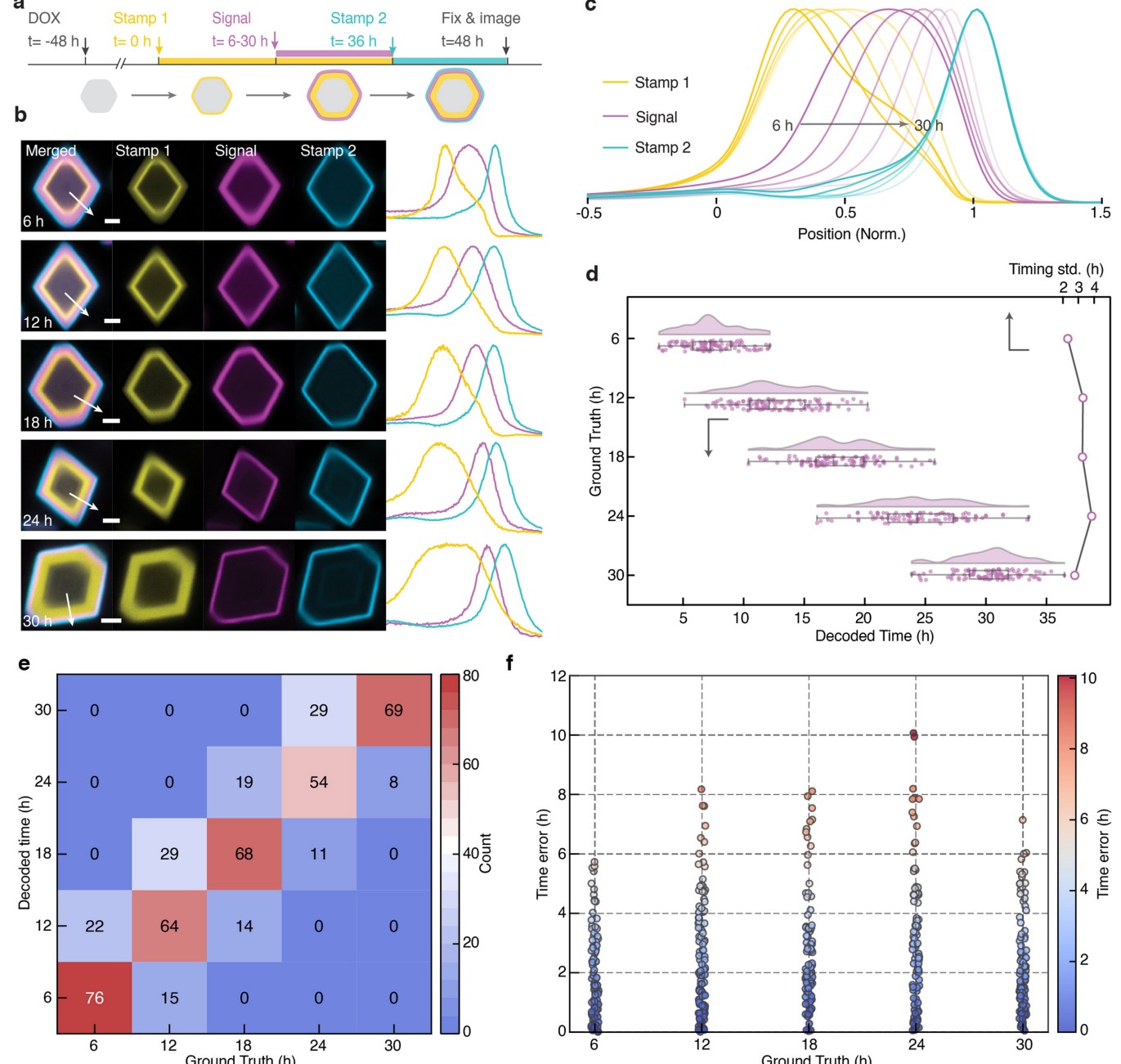

**Extended Data Fig. 4 | Temporal resolution of recording at the later stage of GEMINI growth. a**, Experimental procedure of recording at a later stage of GEMINI growth, where the recording did not start until 48 h after the induction of GEMINI expression (denoted as 0 h), when GEMINI particles are much larger in size. The signal (blue) was induced at t = 6/12/18/24/30 h within a 36 h window defined by stamps #1 (red) and #2 (yellow). The cells were fixed at 48 h. **b**, Images (left) and fluorescence profiles (right) of GEMINI particles with signals introduced at t = 6-30 h. Scale bars: 2 μm. **c**, Mean fluorescence profiles of the timestamps and signal channels. **d**, Decoded onsets from individual GEMINI particles

(bottom-x, n = 100/108/101/96/99 particles for t = 6/12/18/24/30 h) and s.d. (top-x) for each group. **e**, Confusion matrix comparing the decoded time (y-axis) to the ground truth (x-axis) for GEMINI particles in **d**. The color scale indicates the number of particles decoded to each time, highlighting accurate versus misassigned time points. **f**, Time error distribution for individual GEMINI particles in **d** plotted against their ground truths (x-axis). Each dot represents a single particle, with color indicating the magnitude of time decoding error (color bar, right). Box bounds: the 25th/75th percentiles; whiskers: min/max; and center lines: median.

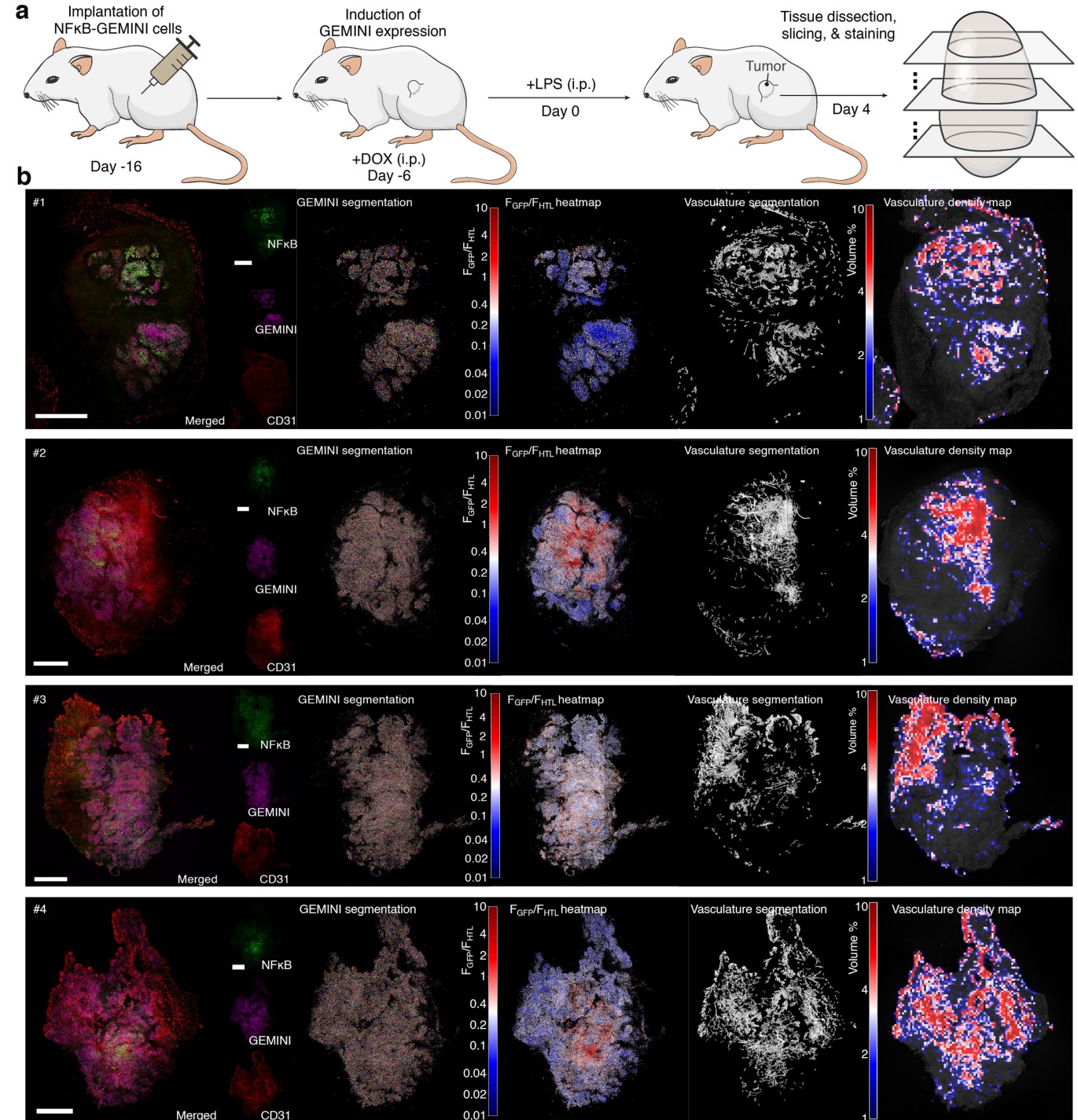

**Extended Data Fig. 5 | GEMINI expression across a tumor xenograft.**
**a**, Experimental procedure for recording inflammation-induced NFκB activity in HEK293T xenografts using GEMINI. Schematic of the mouse adapted from ref. 49, Springer Nature. **b**, Tissue slices from a xenograft following i.p. injection of 3 mg kg$^{-1}$ LPS. GEMINI particles were segmented, and NFκB signal intensity (GFP), normalized to HTL staining, was visualized as heatmaps across the tissue. Vasculature was labeled via anti-CD31 staining, segmented, and displayed as a vascular density heatmap. Scale bars: 1 mm.

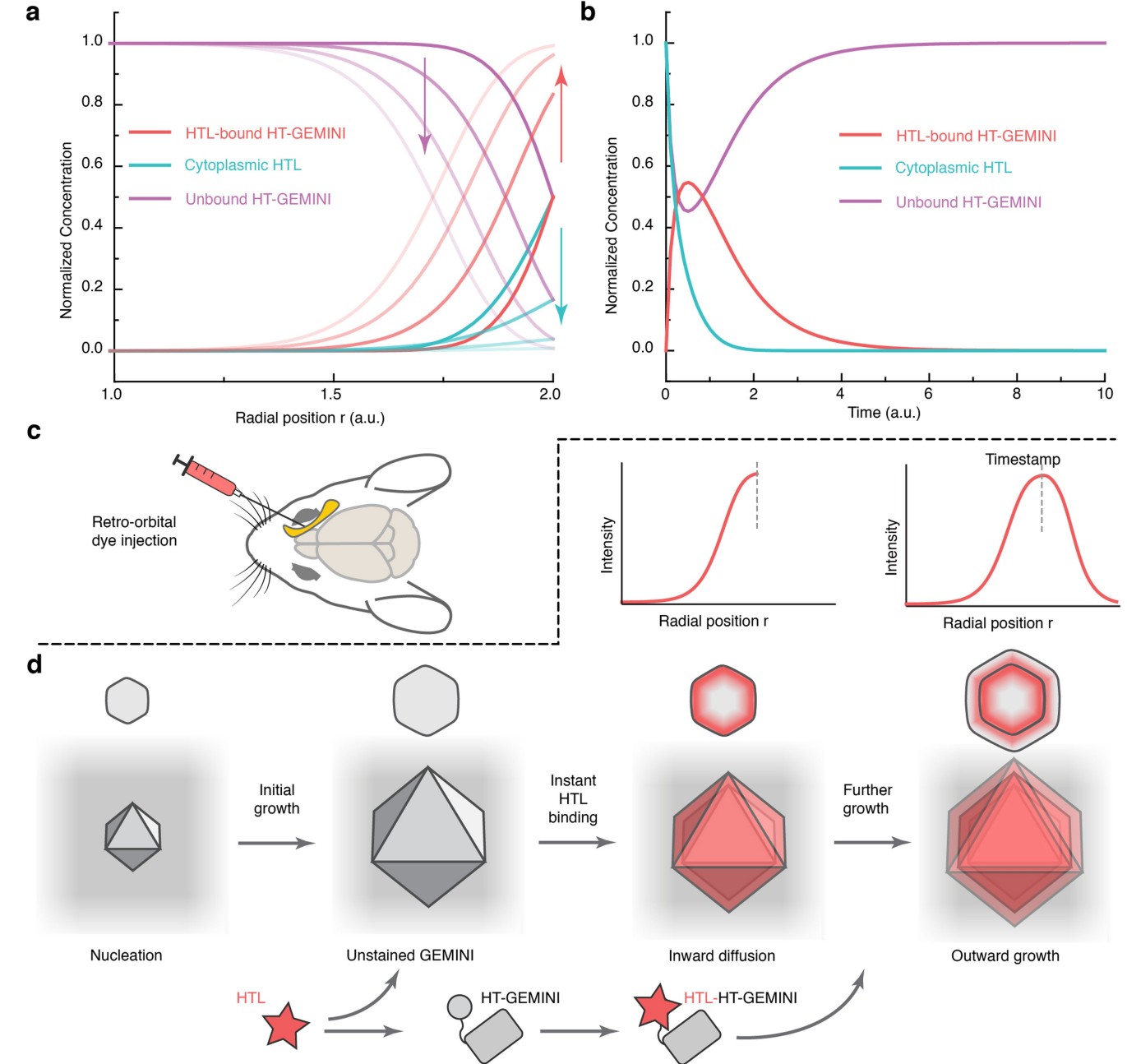

**Extended Data Fig. 6 | In vivo timestamping. a,b** Numerical simulation of the HTL diffusion into existing GEMINI and binding to HT (**a**), and binding with cytoplasmic HT that will further grow onto GEMINI (**b**). **c**, Schematic illustration of systemic injection of HTL dyes into the mouse retro-orbital sinus.

**d**, Schematic illustration of the HT labeling via systemic delivery, where we reasoned that the peak of the fluorescence band is a good proxy for the time of HTL injection.

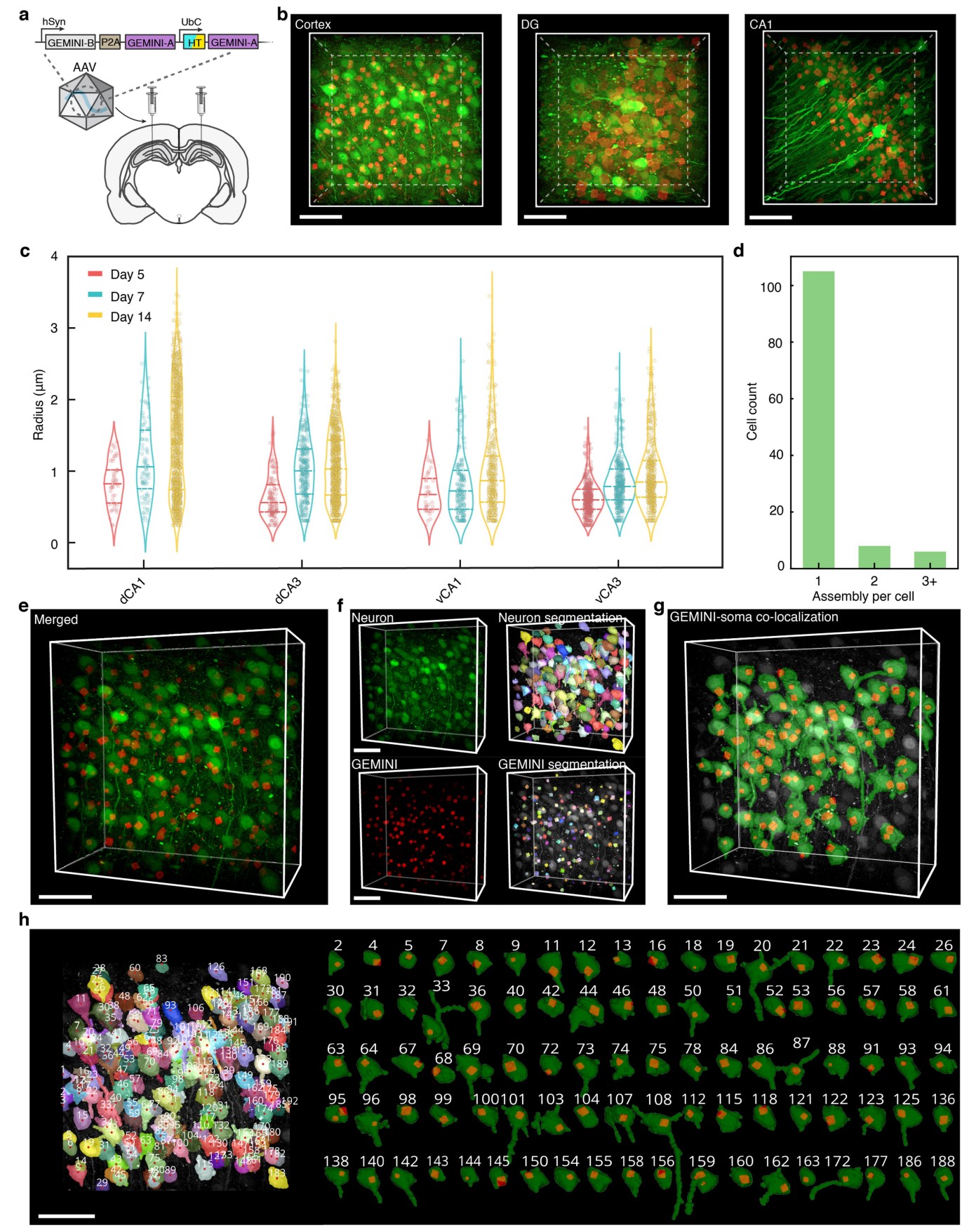

**Extended Data Fig. 7** | See next page for caption.

**Extended Data Fig. 7 | Implementation of GEMINI to the brain and cellular registration. a**, Schematic showing the DNA construct for AAV packaging and the intracranial injection. **b**, 3D images showing GEMINI expression in various brain regions. Scale bar: 50 µm. **c**, Statistical analysis of the GEMINI particle size in various hippocampal regions at days 5, 7, and 14 (n = 2/3/5 mice for days 5/7/14). **d**, Number of GEMINI particles found per cell. Most cells only nucleate one GEMINI particle. **e**, 3D image of brain tissue expressing GEMINI. **f**, Segmentation of neurons (top) and GEMINI particles (bottom). **g**, Spatial colocalization of neurons and GEMINI within the tissue. **h**, Spatial labeling of individual neurons within the tissue (*left*) and the registration of each GEMINI particle to the soma of corresponding neurons (*right*). Each neuron was labeled with a unique number. Neurons with GEMINI were isolated and displayed on the right. Scale bars: 50 µm.

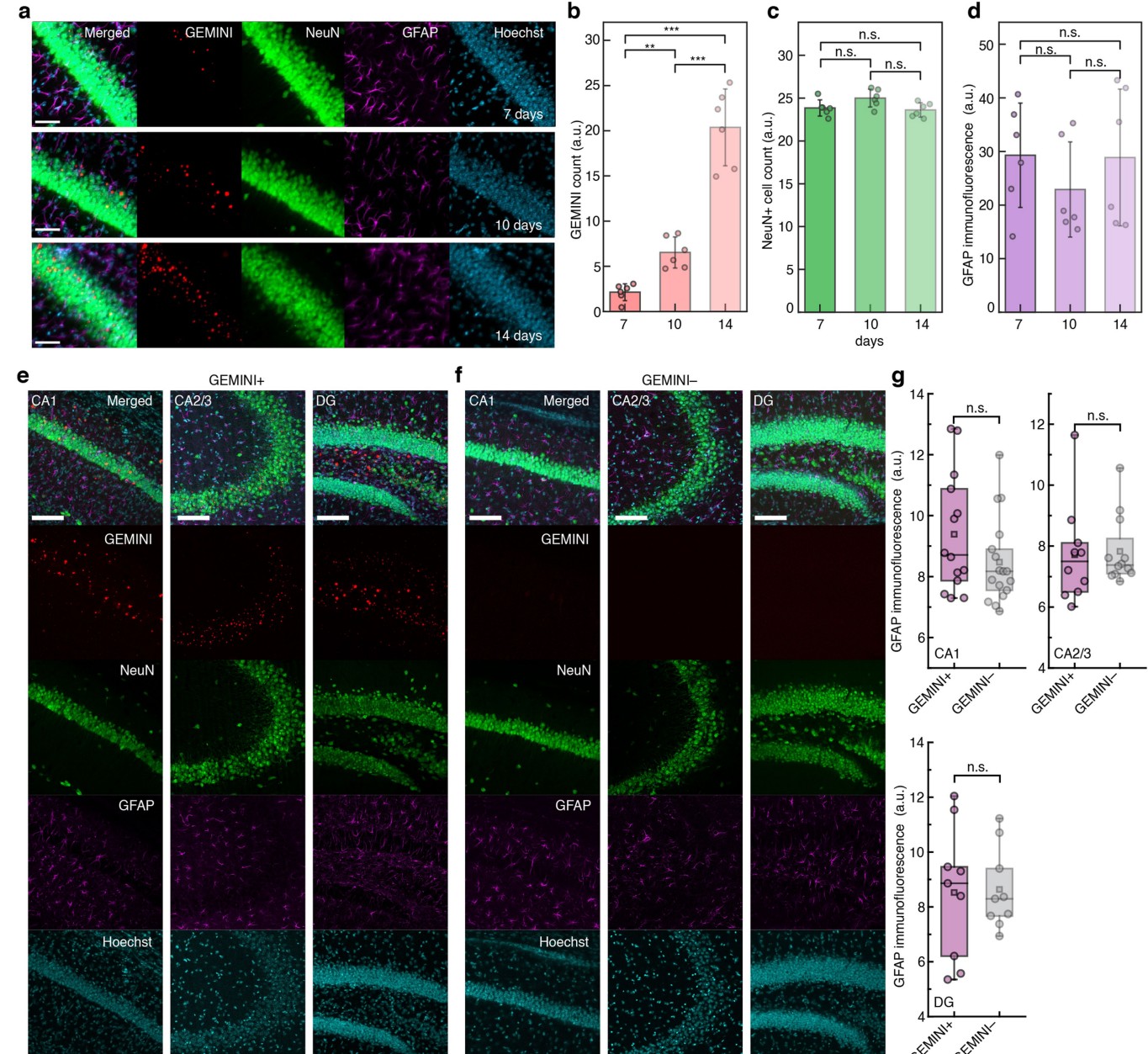

**Extended Data Fig. 8 | Immunohistochemical characterization of the brain tissue. a**, Images showing the CA1 of the mouse hippocampus at 7-14 days after the injection of AAV encoding GEMINI. Scale bars: 50 μm. **b**, Comparison of GEMINI density across 7-14 days, where significant increase in GEMINI count was observed (n = 6 images/group). **c,d**, Comparison of the neuronal density (NeuN+ cells, **c**) and immune response (GFAP immunofluorescence, **d**) across 7-14 days after the injection of AAV encoding GEMINI (n = 6 images/group). **e,f**, Images showing the CA1 (left), CA2/3 (middle), and DG (right) of the mouse hippocampus with (**e**) and without (**f**) intracellular GEMINI nucleation in neurons. The GEMINI+ group received intracranial injection of AAV encoding GEMINI, while the GEMINI− group received equivolume saline. GEMINI particles (JF$_{669}$, red), neurons (anti-NeuN, green), astrocytes (anti-GFAP, magenta), and Nuclei (Hoechst, cyan), were stained and imaged. **g**, Comparison of the immune response (GFAP immunofluorescence) between GEMINI +/− groups (CA1: n = 14/17; CA2/3: n = 10/12; DG: n = 9/9 images for GEMINI +/− groups, from 5 mice/group). Scale bars: 100 μm. **b-d**, bars: mean; whiskers: s.d. **g**, Box bounds: the 25$^{th}$/75$^{th}$ percentiles; whiskers: min/max; squares: mean; and center lines: median. **b-d,g**, two-sided Welch's t-test.

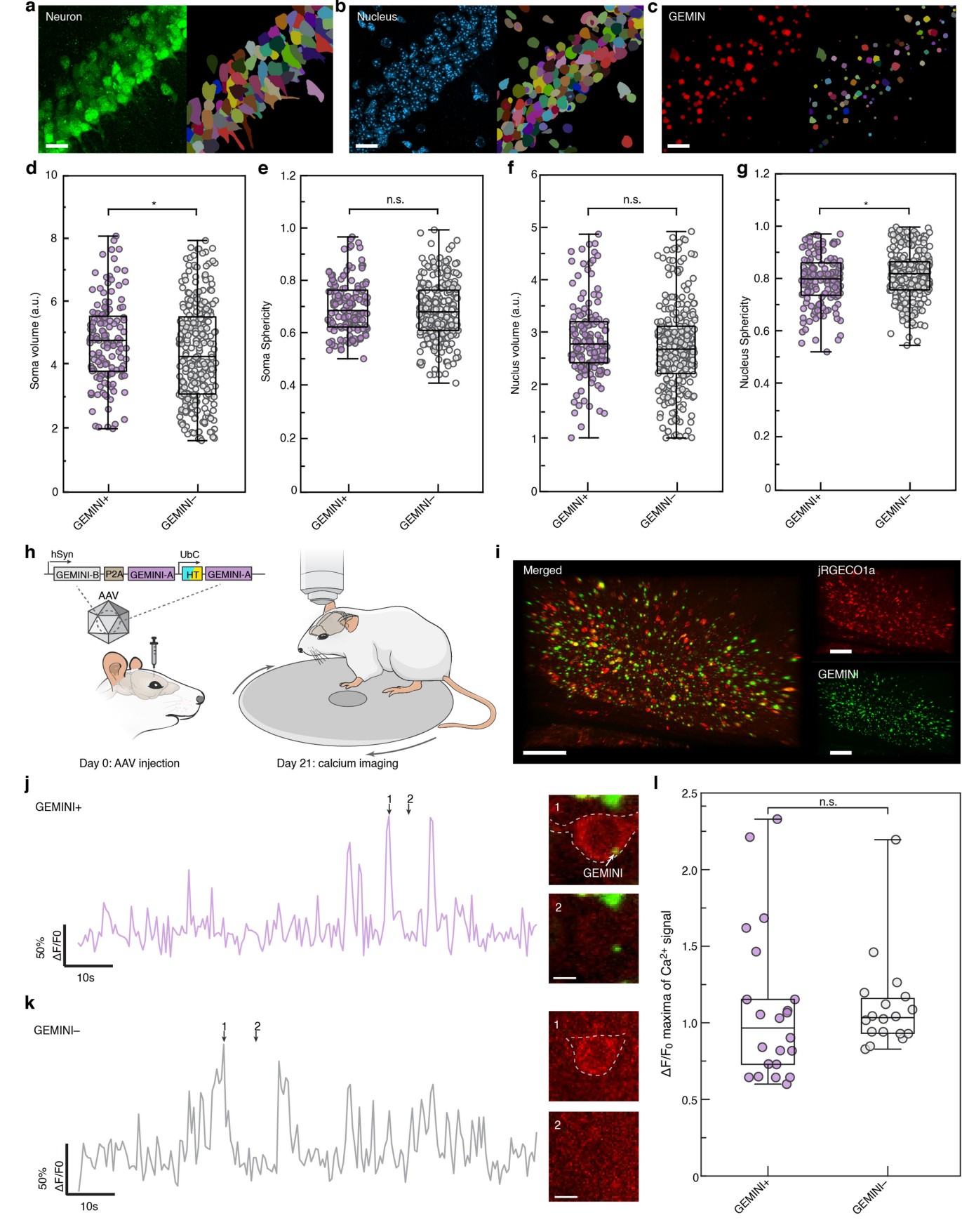

**Extended Data Fig. 9 |** See next page for caption.

**Extended Data Fig. 9 | Impact of GEMINI growth on neuronal morphology and firing. a-c**, Images (left) and their segmentations (right) of neurons (**a**), their nuclei (**b**), and GEMINI (**c**) within the mouse hippocampus CA1. The tissue was harvested 14 days after AAV injection. Scale bars: 20 μm. **d,e**, Comparison of the volume (**d**) and sphericity (**e**) of soma between GEMINI+ and GEMINI− neurons (n = 146/278 cells for GEMINI+/− groups, from 2 mice). **f,g**, Comparison of the volume (**f**) and sphericity (**g**) of nuclei between GEMINI+ and GEMINI− neurons (n = 146/278 cells for GEMINI+/− groups, from 2 mice). **h**, Schematic of the experimental design for in vivo characterization of neuronal activity. AAV encoding GEMINI was injected into the primary visual cortex (V1) of Thy1-jRGECO1a-WPRE transgenic mice on day 0, and two-photon calcium imaging was performed on day 21 with mice head-fixed on a running wheel. Schematic of the mouse adapted from ref. 49, Springer Nature. **i**, Image of the mouse's cortical area showing the high-level co-expression of jRGECO1a and GEMINI. Scale bars: 50 μm. **j,k**, Calcium traces (left) from GEMINI+ (top) and GEMINI− (bottom) neurons in V1, with corresponding cellular images at indicated time points shown (right). Scale bars: 5 μm. **l**, Comparison of $\Delta F/F_0$ of $Ca^{2+}$ peak maxima between GEMINI+ and − groups. (n = 22/18 neurons for GEMINI+/− groups, from 2 mice). **d-g,l**, Box bounds: the 25th/75th percentiles; whiskers: min/max; and center lines: median; n.s., p > 0.05; *, p < 0.05; two-sided Welch's t-test.

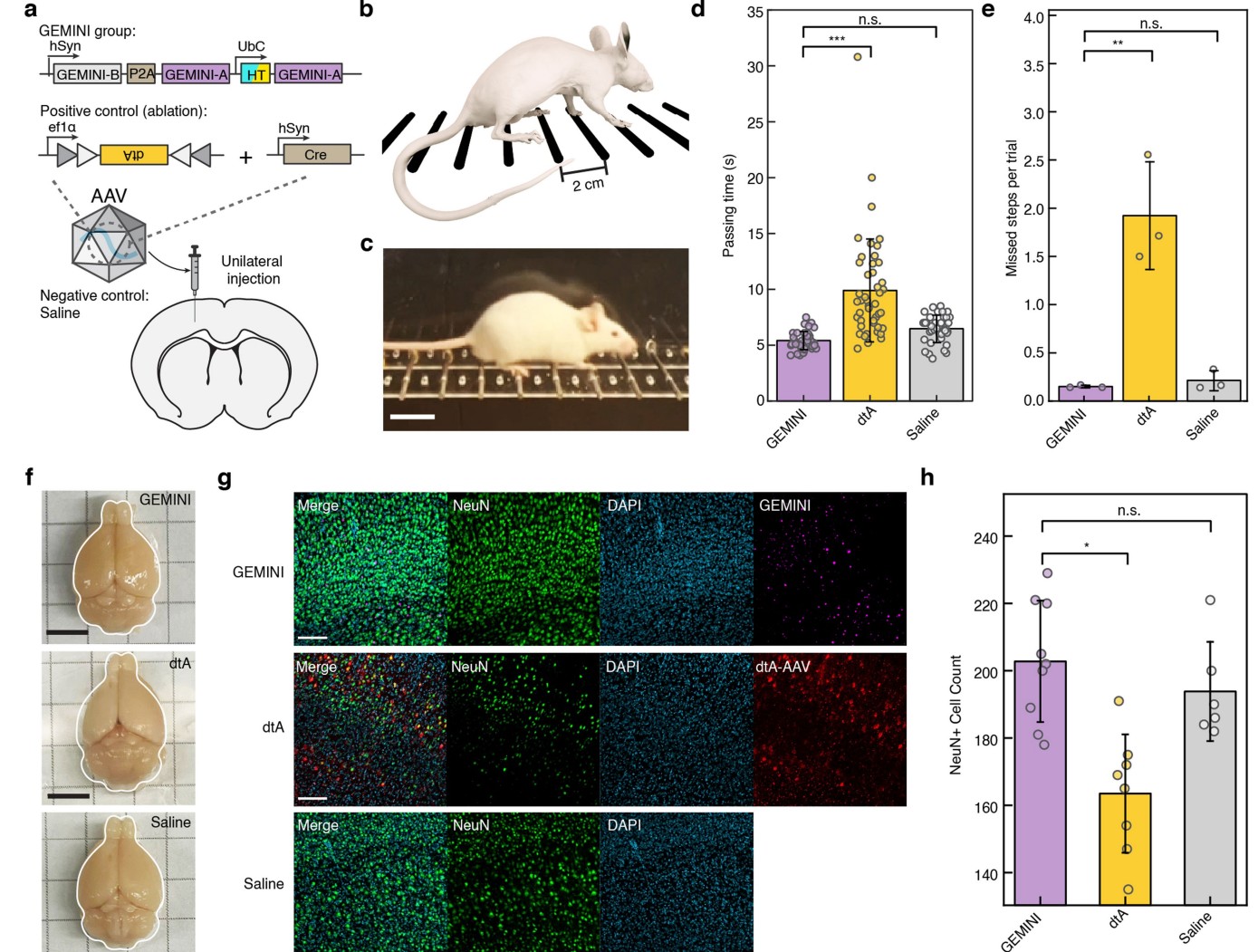

**Extended Data Fig. 10 | Impact of GEMINI expression on fine motor coordination. a**, Schematic of experimental design. AAV encoding GEMINI was unilaterally injected into the primary motor cortex (M1) of mice (GEMINI group). A positive control group received AAVs expressing diphtheria toxin subunit A (dtA) at a comparable dose (dtA group), while a negative control group was injected with equivolume saline (saline group). **b,c**, Schematic (**b**) and snapshot (**c**) of the horizontal ladder (HL) rung walking test. The test was conducted 14 days after the AAV injection. Scale bar: 2 cm. Panel **b** adapted from ref. 49, Springer Nature. **d,e**, Comparison of ladder crossing time (**d**) and mean missed steps per trial (**e**) across the GEMINI, dtA and saline groups (n = 40/46/38 trials for GEMINI/dtA/saline groups). **f**, Images of the brains post-fixation. GEMINI and saline groups have comparable size of both hemispheres, while the dtA group exhibited visible shrinkage on the dtA-injected (left) hemisphere. Scale bars: 5 mm. **g**, Images of the M1 regions from GEMINI, dtA, and saline. Neurons (anti-NeuN, green), Nuclei (DAPI, cyan), GEMINI particles (JF$_{669}$, violet), and cells transduced by the dtA AAV (mCherry, red) were stained and imaged. The mice were sacrificed immediately after the HL tests. Scale bars: 50 µm. **h**, Comparison of the neuronal density (NeuN+ cells) among the GEMINI, dtA, and saline groups (n = 9/8/6 images). **d,e,h**, bars: mean; whiskers: s.d.; n.s., p > 0.05; *, p < 0.05; **, p < 0.01; ***, p < 0.001; two-sided Welch's t-test.

# Reporting Summary

## Statistics

For all statistical analyses, confirm that the following items are present in the figure legend, table legend, main text, or Methods section.

| n/a | Confirmed | |
|---|---|---|
| ☐ | ☒ | The exact sample size (*n*) for each experimental group/condition, given as a discrete number and unit of measurement |
| ☐ | ☒ | A statement on whether measurements were taken from distinct samples or whether the same sample was measured repeatedly |
| ☐ | ☒ | The statistical test(s) used AND whether they are one- or two-sided<br>*Only common tests should be described solely by name; describe more complex techniques in the Methods section.* |
| ☒ | ☐ | A description of all covariates tested |
| ☒ | ☐ | A description of any assumptions or corrections, such as tests of normality and adjustment for multiple comparisons |
| ☐ | ☒ | A full description of the statistical parameters including central tendency (e.g. means) or other basic estimates (e.g. regression coefficient) AND variation (e.g. standard deviation) or associated estimates of uncertainty (e.g. confidence intervals) |
| ☐ | ☒ | For null hypothesis testing, the test statistic (e.g. $F$, $t$, $r$) with confidence intervals, effect sizes, degrees of freedom and $P$ value noted<br>*Give P values as exact values whenever suitable.* |
| ☒ | ☐ | For Bayesian analysis, information on the choice of priors and Markov chain Monte Carlo settings |
| ☒ | ☐ | For hierarchical and complex designs, identification of the appropriate level for tests and full reporting of outcomes |
| ☒ | ☐ | Estimates of effect sizes (e.g. Cohen's *d*, Pearson's *r*), indicating how they were calculated |

*Our web collection on statistics for biologists contains articles on many of the points above.*

## Software and code

Policy information about availability of computer code

| Data collection | We used Zeiss LSM780, LSM 800, LSM980 or Observer7 for data acquisition. The associated Zen Blue or Zen Black softwares (Zeiss) were used for data collection. |
|---|---|
| Data analysis | Linear spectral unmixing was performed using Zen Blue (Zeiss). Matlab (R2023a), Python (3.11), Pymol (3.1.0) and/or ImageJ were used for other data analyses. The code involved custom scripts comprising standard image processing steps, written for the specifics of the datasets, without any novel algorithms involved. Details of data analyses can be found in the manuscripts. The code for data analysis can be found in https://github.com/DCLinLab/GEMINI |

For manuscripts utilizing custom algorithms or software that are central to the research but not yet described in published literature, software must be made available to editors and reviewers. We strongly encourage code deposition in a community repository (e.g. GitHub). See the Nature Portfolio guidelines for submitting code & software for further information.

# Data

Policy information about availability of data

All manuscripts must include a data availability statement. This statement should provide the following information, where applicable:

- Accession codes, unique identifiers, or web links for publicly available datasets
- A description of any restrictions on data availability
- For clinical datasets or third party data, please ensure that the statement adheres to our policy

Source datasets for in vivo experiments are provided with the paper. Complete datasets reported in this article are available at GitHub (https://github.com/LinLab/GEMINI). The plasmids constructed and used in this work are deposited to Addgene (Plasmid IDs: 228881-228886, 228888, 228889).

# Research involving human participants, their data, or biological material

Policy information about studies with human participants or human data. See also policy information about sex, gender (identity/presentation), and sexual orientation and race, ethnicity and racism.

| | |
|---|---|
| Reporting on sex and gender | N/A |
| Reporting on race, ethnicity, or other socially relevant groupings | N/A |
| Population characteristics | N/A |
| Recruitment | N/A |
| Ethics oversight | N/A |

Note that full information on the approval of the study protocol must also be provided in the manuscript.

# Field-specific reporting

Please select the one below that is the best fit for your research. If you are not sure, read the appropriate sections before making your selection.

☒ Life sciences      ☐ Behavioural & social sciences      ☐ Ecological, evolutionary & environmental sciences

For a reference copy of the document with all sections, see nature.com/documents/nr-reporting-summary-flat.pdf

# Life sciences study design

All studies must disclose on these points even when the disclosure is negative.

| | |
|---|---|
| Sample size | For each experiment, sample size was as large as could be practically achieved by our microscopic imaging, given the number of cells available in the images taken. Sample sizes were not pre-defined, but proper statistical analyses were performed to account for technical and biological variability. |
| Data exclusions | GEMINI particles that nucleated later than the first timestamp (without showing complete timestamps) were excluded. In seizure model, mice with a seizure score <5 were excluded. No other GEMINI particles or animals were purposely excluded from analysis. |
| Replication | At least three replicates are performed or include for each experiment. The reproducibility of all experiments are ensured. |
| Randomization | Animals were randomly assigned to experimental groups prior to data collection. For cell-based experiments, culture dishes from the same preparation batch were randomly allocated to conditions. |
| Blinding | Investigators were blinded to group allocations. |

# Reporting for specific materials, systems and methods

We require information from authors about some types of materials, experimental systems and methods used in many studies. Here, indicate whether each material, system or method listed is relevant to your study. If you are not sure if a list item applies to your research, read the appropriate section before selecting a response.

## Materials & experimental systems

| n/a | Involved in the study |
|---|---|
| ☐ | ☒ Antibodies |
| ☐ | ☒ Eukaryotic cell lines |
| ☒ | ☐ Palaeontology and archaeology |
| ☐ | ☒ Animals and other organisms |
| ☒ | ☐ Clinical data |
| ☒ | ☐ Dual use research of concern |
| ☒ | ☐ Plants |

## Methods

| n/a | Involved in the study |
|---|---|
| ☒ | ☐ ChIP-seq |
| ☒ | ☐ Flow cytometry |
| ☒ | ☐ MRI-based neuroimaging |

# Antibodies

| | |
|---|---|
| Antibodies used | anti-NeuN (GeneTex, GTX132974), anti-glial fibrillary acidic protein (anti-GFAP, Cell Signaling, #3670), anti-Phospho-IκBα antibody (Cell Signaling, 9246S), anti-IκBα antibody (Cell Signaling, 4812S), anti-Vinculin (Cell Signaling, 13901), HRP-conjugated goat anti-rabbit IgG(H+L) (ProteinTech, SA00001-2), goat anti-mouse IgG (H + L)-HRP Conjugate (Bio-Rad, 1706516), anti-CD31 (BD Biosciences, 550274), Donkey anti-Rat AF647 antibody (Invitrogen, A48272), Donkey anti-Mouse IgG (H+L) Alexa Fluor™ 488 (Invitrogen, A-21202), Donkey anti-Rabbit IgG (H+L) Alexa Fluor™ 546 (Invitrogen, A10040) |
| Validation | The vendors of the antibodies have provided validation, relevant test results, and/or certificates through their websites:<br>Anti-neuronal nuclear (NeuN, GeneTex, GTX132974) https://www.genetex.com/Product/Detail/NeuN-antibody/GTX132974<br>Anti-glial fibrillary acidic protein (GFAP, Cell Signaling, #3670):https://www.cellsignal.com/products/primary-antibodies/gfap-ga5-mouse-monoclonal-antibody/3670<br>Anti-Phospho-IκBα antibody (Cell Signaling, 9246S): https://www.cellsignal.com/products/primary-antibodies/phospho-ikappab-alpha-ser32-36-5a5-mouse-monoclonal-antibody/9246<br>Anti-IκBα antibody (Cell Signaling, 4812S): https://www.cellsignal.com/products/primary-antibodies/ikappab-alpha-44d4-rabbit-monoclonal-antibody/4812<br>Anti-Vinculin (Cell Signaling, 13901): https://www.cellsignal.com/products/primary-antibodies/vinculin-e1e9v-rabbit-monoclonal-antibody/13901<br>HRP-conjugated goat anti-rabbit IgG(H+L) (ProteinTech, SA00001-2): https://www.ptglab.com/products/HRP-conjugated-Goat-Anti-Rabbit-IgG-H-L-SA00001-2.htm<br>Goat anti-mouse IgG (H + L)-HRP Conjugate (Bio-Rad, 1706516): https://www.bio-rad.com/en-us/sku/1706516-goat-anti-mouse-igg-h-l-hrp-conjugate?ID=1706516<br>Anti-CD31 (BD Biosciences, 550274): https://www.bdbiosciences.com/en-us/products/reagents/flow-cytometry-reagents/research-reagents/single-color-antibodies-ruo/purified-rat-anti-mouse-cd31.550274?tab=product_details<br>Donkey anti-Rat AF647 antibody (Invitrogen, A48272): https://www.thermofisher.com/antibody/product/Donkey-anti-Rat-IgG-H-L-Highly-Cross-Adsorbed-Secondary-Antibody-Polyclonal/A48272TR<br>Donkey anti-Mouse IgG (H+L) Alexa Fluor™ 488 (Invitrogen, A-21202): https://www.thermofisher.com/antibody/product/Donkey-anti-Mouse-IgG-H-L-Highly-Cross-Adsorbed-Secondary-Antibody-Polyclonal/A-21202<br>Donkey anti-Rabbit IgG (H+L) Alexa Fluor™ 546 (Invitrogen, A10040): https://www.thermofisher.com/antibody/product/Donkey-anti-Rabbit-IgG-H-L-Highly-Cross-Adsorbed-Secondary-Antibody-Polyclonal/A10040 |

# Eukaryotic cell lines

Policy information about cell lines and Sex and Gender in Research

| | |
|---|---|
| Cell line source(s) | HEK293T cells (ATCC, CRL-3216); COS-7 (ATCC, CRL-1651); MDCK (ATCC, CCL-34); U-2 OS (ATCC, HTB-96); B16-F10 (ATCC, CRL-6475); 4T1 (ATCC, CRL-2539); MDA-MB-231 (ATCC, HTB-26); U-87 MG (ATCC, HTB-14) |
| Authentication | None of the cell lines used were further authenticated. |
| Mycoplasma contamination | All cells were tested negative for mycoplasma. |
| Commonly misidentified lines (See ICLAC register) | No commonly misidentified cell lines were used in the study |

# Animals and other research organisms

Policy information about studies involving animals; ARRIVE guidelines recommended for reporting animal research, and Sex and Gender in Research

| | |
|---|---|
| Laboratory animals | 5-10 weeks CD-1 (Charles River Laboratories, 022), C57BL/6 mice (Jackson Laboratory, 000664), and 4-8 weeks J:Nu (Jackson Laboratory, 007850) were used in this study |
| Wild animals | No wild animals were used in this study |
| Reporting on sex | In the xenograft model, female J:Nu immunodeficient mice (Homozygous for Foxn1<nu>) were used in the study. Otherwise, sex was not considered in this study. Mixed male and female mice were used in the remaining study. Mice of different genders were assigned |

randomly to different groups.

| | |
|---|---|
| Field-collected samples | No field-collected samples were used in this study |
| Ethics oversight | The Animal Care and Use Committee (ACUC) of Johns Hopkins University approved the animal protocol (MO21E409 and MO24E325) for this study. |

Note that full information on the approval of the study protocol must also be provided in the manuscript.

## Plants

| | |
|---|---|
| Seed stocks | N/A |
| Novel plant genotypes | N/A |
| Authentication | N/A |

