## [Peer Review File · Nature]

Genetically encoded assembly recorder temporally resolves cellular history

Corresponding Author: Dr Dingchang Lin

Version 0:

Reviewer comments:

Referee #1

(Remarks to the Author)

In this manuscript, the authors have developed an in cellulo recording platform termed Granularly Expanding Memory for Intracellular Narrative Integration (GEMINI), which utilizes a computer-designed two-component self-assembly complex to record cellular activity history. This self-assembly complex is relatively large, with bright and easily observable signals, enabling high temporal resolution that surpasses benchmark fluorescent transcriptional reporters by two orders of magnitude. The authors first expressed GEMINI in cultured cells, demonstrating the system's stability and accuracy, then successfully applied it in vivo to record transcriptional changes and cellular activity during PTZ-induced seizure behavior in the mouse brain.

Overall, I found this to be a solid body of work that reports the development and application of a useful new tool. The manuscript is well-written, logically explained. However, it appears that the authors may have prepared the manuscript in haste, as the arrangement of the figures is suboptimal. Substantial improvements and careful proofreading are needed to ensure accuracy and clarity. Additionally, the authors need to further explore and demonstrate how this tool can address and uncover new biological questions. I would recommend accepting this manuscript for publication following revisions to address the comments below.

Major critiques:

1. This paper presents the use of two-component protein cages, GEMINI-A and GEMINI-B, to form an assembly complex, with the two components linked via the P2A sequence to ensure equal expression levels. This makes me wonder why the authors did not consider designing a single-component self-assembling protein for the GEMINI system. A single component could potentially reduce the number of exogenous genes introduced into the cells, allow for greater packaging capacity in the AAV vector, and simplify AAV packaging. It would be beneficial for the authors to discuss whether they tried that or is that possible in principle.
2. In the main figures, the GEMINI strategy constructs are not depicted, and they only appear in the extended figures. This makes it inconvenient for readers to fully understand the design and how the complex constructs are applied in different experiments. It would be helpful to include at least some of the GEMINI strategy constructs (possibly accompanied with cartoons) in the main figures to improve the clarity and context of the experiments.
3. In this paper, the authors do not fully demonstrate the advantages of GEMINI compared with previously published linear recording tools through a side-by-side comparison. For example, it would be helpful to show whether the 3D structure provided by GEMINI is significantly more accurate than the 1D approach used under the same conditions. Comparisons in other aspects are also feasible. These comparisons would help clarify the advantages and unique contributions of GEMINI.
4. In Figure 4b, the authors use a destabilization method to improve temporal resolution, but the specific strategy is not clearly outlined in the main text or figure, making it difficult to understand how it was implemented. If the strategy is the same as that presented in the Fig. S10 figure legend (where P1N4 was used), this should be referenced in the main text for clarity.
5. In Figure 6, the authors use GEMINI to record cellular activity during PTZ-induced seizure behavior in the mouse brain, which involves a strong, non-physiological stimulus. This raises the question of whether this tool could also be used to record activity under physiological conditions, such as visual stimuli, feeding, or motor behavior. What is the sensitivity of this tool for detecting activity under more naturalistic, physiological conditions?
6. In this paper, the authors have validated the usage of GEMINI in vivo, but I feel the practicality of this tool is not fully demonstrated. I suggest they need to further explore some new biological questions or discoveries that this tool could address.
7. Based on Extended Fig. 6c and Fig. S4, as time progresses, the radius of the assembled particles increases, reaching up

to 3 μm . I wonder if this could impact the physiology or excitability of neurons. The authors may consider addressing this by recording and comparing the calcium signaling or electrophysiological properties between GEMINI-expressing neurons and non-GEMINI-expressing neurons. This could provide valuable insights into potential effects on neuronal activity. Of course, the existing cell viability and behavioral tests can help readers understand the impact of GEMINI on cells or mice from different perspectives.

8. Similar to the analysis of decoded time in Figure 2 and 3, have the authors conducted similar experiments and analysis in vivo? What temporal resolution can GEMINI achieve in vivo? It would be helpful if the authors could present some results or elaborate on this in the discussion section.

Minor critiques:

1. In Extended Fig. 1c, the assembly images of variants #3-v1 and #3-v3 appear unclear and difficult to interpret. Based on the fluorescent images in Extended Fig. 1d, #3-v1 and #3-v3 show a similar size to #3-v2, so I expected to see similar assembly patterns, but this is not evident in the images. Could the authors provide an explanation for these results or present more representative data?
2. In Figure 2j and Extended Fig. 4d, the meaning of the arrows in the figure is unclear. Could the authors provide an explanation for what these arrows indicate in the figure legend? Clarifying this would improve the figure's interpretability.
3. In Figures 3b and 3e, the sequence in which the different stamps are exhibited appears a bit unusual, as stamp 3 is added last but exhibited before stamp 2. If possible, could the authors consider rearranging the presentation for improved readability? One potential suggestion would be to reorder the stamps to align with the chronological appearance of the signals, which might help clarify the progression.
4. In Figure 3g, two different JF dyes are used, but they are not labeled in the schematic above. Additionally, the presentation style is inconsistent with Figures 3b and 3e.
5. In Figure 4c, the concentration of TNF- α used in the bottom row of fluorescence images appears to be missing. Additionally, the meaning of "12x" and "10x" in the figure should be clarified. In Figure 4d, the concentration labels at the top and the fluorescence intensity measurements recorded over time at the bottom are not properly aligned, which makes it challenging to interpret the data.
6. In the Extended Fig. 4 figure legend, the numbering for panels 4c and 4d is incorrectly written as "i" and "j".
7. In Extended Fig. 6b, is the expression time the same across the cortex, DG, and CA1 regions? It appears that the GEMINI particles are notably larger in the DG group, even under the same scale bar. Could the authors explain the reasons?
8. In Line 403, you mention Extended Fig. 7c, but I cannot find this figure.

Referee #2

(Remarks to the Author)

Yan and colleagues described an interesting way of recording temporal events in living cells (termed GEMINI) by growing a three-dimensional structure using a computationally designed protein as the basic unit. The unit can be fluorescently labeled by HALO dyes or expressed as an FP-fusion protein from promoters that can be temporally controlled. This 3D recorder has several advantages over previously described linear recorder in reduced toxicity and ease of decoding (by imaging from all angles), etc. While the design of GEMINI is innovative and interesting, and some of the in vitro characterization impressive, the application is quite limited (see specific critiques below), particularly in vivo. For a new method to be published in Nature, we should expect that it can solve a problem that cannot be solved by other methods. At this point, GEMINI is quite far from that. We hope that our critiques below are nevertheless useful for the authors to improve the method and/or its presentation.

A. The "low cytotoxicity" is not well characterized.

1. In cultured cells, from the main text to the extended data and methods, the authors did not clarify how and when they assayed the live/dead ratio. Effects from a "particle" close to ~25% of the size of the nucleus should be assayed at different time points to identify the optimal working time window for GEMINI.
2. If possible, co-staining GEMINI with different organelle markers (ER, Golgi, mitochondria, lysosomes, endosomes) at different time points would help assess cellular health at the subcellular level.
3. For in vivo validations, conditions for cytotoxicity assays are omitted, assay timing is not shown, "normal morphology" is described without data, and GEMINI particles are not co-stained.
4. The behavioral assays are not sensitive to perturbation in those specific brain regions. A more sensitive assay would be unilateral motor cortex injection; it is known that asymmetric function of motor cortex may cause animals to turn in a specific direction. A good control for these experiments would be to inject an AAV expressing a transgene that silences or kills the transduced cells—would that yield quantifiable difference in the behavioral assay?

B. The claim "recording at the single-cell level" is not well demonstrated or does not apply in vivo:

1. It should include identifiable cell segmentation and the registration of each GEMINI particle to a nucleus.
2. To decode the "record," the authors need to pool many heterogeneous single cells/particles together to narrow the distribution. Individual cells/particles do not yield effective decoding.

C. Other comments:

1. High temporal resolution is observed only in the early stages of cultured cells.
2. For in vivo applications, the authors have not demonstrated the ability to decode stimulus timing or dose as they do in vitro. In fact, we are quite confused with Fig 6d. It seems that even when DOX is applied at 0h (same as timestamp #1?), the

peak occurs after timestamp#2, which was applied 1 day later. Is that due to delay in DOX-induced transcription?
3. The GitHub link does not work.

Referee #3

(Remarks to the Author)

The manuscript by Yan and co-workers describes a new cellular recording system that utilizes growing protein crystals to store information about order of events in-cellulo and in-vivo. The method generates large protein crystals intracellularly and pulses of fluorescent materials can be generated to store information of timing events that generates optically readable patterns like growth rings in trees. The work is innovative and novel. It appears that it however lacks a demonstration that clearly leverages the advantages claimed in the introduction. The authors are correct when they write that many systems that record cellular events have been demonstrated before by recording edits to DNA that can be read by sequencing. The authors are also correct that these other methods are i) typically read out over a population of cells, and ii) since they are not optical, would require disruption of the tissue as well as iii) would not be able to answer the absolute chronological order of events (only order of events). For all of these it is not clear to this reviewer how they are leveraged in the in-vivo demonstration provided, i.e. couldn't for example the transcription and seizure events in-vivo be measured with genetic recording? Especially since in all the experiments where the authors are resolving time-stamps for events, they appear to use averaged traces making this more of a bulk method (providing info of effects for a group of cells). In my opinion, the work is better suited for Nature Biotechnology or Nature Methods.

Specific remarks:

1. Growth rate. Fig 2d shows the "volume index" (a measure of volume based on 2D imaging). This is clearly not linear at the later phases of the growth (>35h). Why not focus on the linear part of the growth range? In addition, the spikes in the individual growth profiles seem to indicate that individual particles are either non-stable or not growing linearly. This is an issue with the claimed advantage i) above. Population averages appears quite important for this method.

2. Also, in fig 2b it appears that the outer layer is affected more by longer post-treatment incubation. Doesn't this mean that the outer layer is not that "locked"? (I.e. contradicting "minimal exchange between BBs in GEMINI particles and those in the cytoplasm"?)

3. The authors write "The results demonstrate the high accuracy of the model in describing individual GEMINI growth, which also corroborates the assumption of steady protein expression, an important indicator of normal cell metabolism." I think there are several problems with this. First of all it seems to assume that all synthesized BBs get either degraded or incorporated, why wouldn't you have a stable non-incorporated population? Secondly, based on the curves (see comments about Fig. 2 above), the growth does not seem to be continuous for individual particles. Also these assumptions are only true to cell culture conditions, I don't think they can be assumed in physiological conditions.

4. Line 208 "We also analyzed the influence of GEMINI on cell morphology and division. Nucleation did not alter cell shapes or interrupt mitosis and cytokinesis. After division, GEMINI particles randomly entered one of the daughter cells (Extended Data Fig. 3e,f)." I don't think this is enough data to make this kind of broad statement, soften the wording.

5. Line 230 "The results showed close agreement with the ground truths, where the means of the decoded time exhibited errors of less than 30 minutes." The mean did, on an individual level it would be hard to interpret. The authors should scramble the data and try to predict the timestamp to get a real accuracy value (e.g. take segmented images of individual particles and try to predict the time).

6. Line 239: "a temporal resolution of 2-4 hours was still achievable, demonstrating GEMINI's broad recording window." One cannot equate temporal resolution to the spread (std) of a population measurement. If one wants to use the std then the safely resolvable "distance" of events would be around 2x std.

7. Line 253: "Comparable basal level and response to tumor necrosis factor-alpha (TNF- α) were found between the groups with and without GEMINI growth, indicating negligible impact of GEMINI growth on the pathway." They do have a higher baseline I κ Ba phosphorylation with GEMINI, which might suggest a cellular stress response. Please acknowledge this in the text.

8. Fig 5 and: "In the Y-maze test, mice from the two groups spent a comparable amount of time in the novel arm versus the alternative arm, indicating negligible impairment of short-term memory by GEMINI expression." As the plots show, the average performance of mice with GEMINI seems similar to the control, however the spread of the metrics seems different which might be due to the particles?

9. Inheritance of particles could have an influence on the interpretation of a lot of the data. Meaning that only one daughter cell inherits the particle and the other one has to grow a new one with "erased" memory. How is this handled?

10. Bulk vs. individual cells. As mentioned in the beginning of the review, in all the experiments where the authors are resolving time-stamps for events they appear to use averaged traces making this more of a bulk method (providing info of effects for a group of cells). This could be a clear limitation as in their artificial systems where they induce events with

injections, this is fine as all cells are going through the same events at the same time, but even there it is hard to say where differences come from as differences in diffusion of the dyes they use for timestamping and compounds they use for inducing transcriptional changes can have an effect on their results. It would be nice if the authors could address these concerns with experiments, but in any case it is my opinion that the language about the method being able to record events on a single cell level should be toned down a lot.

Referee #4

(Remarks to the Author)

Nature is committed to facilitate training in peer-review and to ensure that everyone involved in our peer review process is appropriately recognised. This reviewer co-reviewed one of the listed reports.

Version 1:

Reviewer comments:

Referee #1

(Remarks to the Author)

This manuscript has been thoroughly revised, and the authors have adequately addressed the concerns I raised previously. In the present version, I still have some minor concerns and critiques that I believe should be considered to further address. Overall, I recommend acceptance of this manuscript pending minor revisions.

1. In the newly added Fig. S7, the authors examined the impact of GEMINI expression on cell morphology and subcellular structures. I noticed that the authors analyzed all signal (Hoechst, ConA, Syto14, MitoTracker) except for Phalloidin+WGA. Could the authors clarify why this analysis was excluded—was there any significant difference? In addition, since one of the GEMINI's important application is to record neuron activity history, have the authors tested its potential impact on neuronal subcellular structures? In Fig. 5a, GEMINI expression in neurons sometimes appears relatively large in size. Could GEMINI expression affect organelle transport (e.g., mitochondria or vesicles) from the soma to dendrites and axons?
2. In Extended Data Fig. 3, the authors present a clear side-by-side comparison of GEMINI and iPAK4 behavior during cell division. However, the methods for this experiment are not described in the manuscript. It would be helpful to provide details on how the experiment was performed, including what each signal represents and how the nucleus or membrane was labeled.
3. There are some inconsistencies between figures, legends, and the main text that should be addressed. In Fig. 2, the authors optimized the color coding of each timestamp, but the figure legend was not updated accordingly. In Extended Data Fig. 9h, the figure legend states that the virus was injected into M1, while the main text (line 527) states that it was injected into the visual cortex. In Extended Data Fig. 10, the figure legend indicates V1, but it should be M1.
4. In Fig. S6c, it is difficult to see GEMINI expression with the current contrast.

Referee #2

(Remarks to the Author)

Yan and colleagues made extensive efforts and provided additional evidence and analysis to address our critiques and those of other reviewers. These include detailing experimental methods for the live/dead ratio (Fig. S6), examining in vitro (Fig. S7) and in vivo (Extended Data Figs. 9, 10) cytotoxicity, and showing single-cell recording with high-quality segmentation and registration (Extended Data Fig. 7). The additional xenograft experiment supported the claimed "single-cell/particle decoding capabilities" better than the previous mouse brain experiments. Overall, we support its publication.

A few minor points need further discussion or clarification:

1. Regarding single-cell/particle decoding capabilities, reviewer #3 also raised similar questions. Although the authors emphasized the performance in vitro (Figs. 2b, g, i; Extended Data Fig. 4; Fig. S8), our previous comment B.2 was specifically about GEMINI's in vivo application. Considering the points the authors listed in their response (the heterogeneity of neurons, variability in timestamping and AAV delivery, etc. leading to the low confidence of "single-cell/particle" data points), can users learn which cell type or brain region responds to the induction earliest or most strongly?
2. Without cell membrane staining and segmentation, it's difficult to definitively link each particle to a specific cell (Fig 4b, e, g, i, and j). The authors demonstrated single-cell recording with high-quality segmentation and registration (Extended Data Fig. 7). However, ~10% of cells contained more than one assembly/particle (Extended Data Fig. 7d), and membrane staining/segmentation is not a standard step when decoding or tracing events in most analyses. As a result, each dot represents one particle/assembly, but not necessarily a unique cell. We suggest using the description "single-particle/assembly" and reserving "single-cell" only if you can definitively link a particle to a cell.
3. Fig. 4c suggested level-dependent signal, and the authors reasoned that cytokine diffusion may contribute to the peak delay. In the revision experiments (Figs. 4i–l), do GEMINI particles closer to vascularization also show higher response levels?
4. In Figs. S7a and S7f, Syto14 staining showed more puncta in the nucleus under GEMINI+ conditions. If Hoechst is used for segmentation and Syto14 staining is compared only in the nucleus between GEMINI+ and GEMINI- groups, would you see any differences?
5. What does the green color represent in the heatmap in the Fig. 4b inset (right)?

Referee #3

(Remarks to the Author)

The revised manuscript by Yan and co-workers includes new experiments and data and is an improved version of their previous manuscript. Nevertheless it appears that the revisions leave some of the main criticisms.

The authors have added a new study where they implanted GEMINI-expressing cells into immunodeficient mice and triggered inflammation by injection of LPS. I recommend the authors for this effort and I think it strengthens their manuscript. In particular, it shows a possibility of measuring dosage response, which I believe is a feature not commonly found in other cellular recording methods. The authors write that "This model is less strong and artificial than the PTZ-induced seizure." Maybe that is true for the inflammation aspect of the model, but I would argue that subcutaneous xenograft into immunodeficient mice is artificial. So still, it is not clear to this reviewer how this significantly changes the manuscript in terms of showing a clear case for previously inaccessible discoveries.

The authors strongly argue that the method is single-cell and that averages are only used for statistical purposes. I agree with the authors that the readout in itself is single-cell, but that, in contrast to scRNA-seq, the method is not used to differentiate between cell types (or in this case, cell reactions to events), but rather to measure the population as a whole. I still think that the way the data is used, not to look for differences, but to look for average responses, makes it harder to argue the case that this is significantly disparate from bulk methods. Again, I agree that the readout is single-cell, and that this could potentially be leveraged to look for cell-type or cell-position specific responses in the future, but it is not leveraged in this study. This needs to be concretely stated in the manuscript in the form that yes, in the future this method can potentially differentiate between different responses in different populations, but with the current state of the method and data, this remains hypothetical.

Regarding my specific critique 1, the authors provide answers that are mostly satisfactory except for how fig. 1h (previously 2d) is not reliant on population averages. The authors do not in my opinion explain how one can do away with the population averaging in 1h. In fact, I would say that their reply highlight why this is a valid critique "While decoding variability exists, as is common with any biological sensing platform, it arises primarily from factors including: (1) intrinsic cellular heterogeneity that affects signal response kinetics, (2) variability in protein expression levels and thus the growth dynamics of GEMINI particles, and (3) error associated with timestamping and the temporal decoding model." Again, the wording in the manuscript needs to acknowledge this.

Overall, I think this is very nice work, but some main criticisms from the initial review remains.

Referee #4

(Remarks to the Author)

I co-reviewed this manuscript with one of the reviewers who provided the listed reports.

Version 2:

Reviewer comments:

Referee #1

(Remarks to the Author)

Thank you for addressing all of my concerns, I have no further revisions.

Referee #2

(Remarks to the Author)

The authors have addressed our comments comprehensively. We look forward to future in vivo applications with optimized AAV delivery or transgenic lines. Regarding our question on cytokine diffusion, our intent was not to request a mechanistic investigation or more detailed 3D characterization; rather, the presented data implies a decent in vivo dose sensitivity within a local region (even within $\sim 150 \mu\text{m}$). Overall, this work now represents a valuable recording method that merits publication in Nature.

Referee #3

(Remarks to the Author)

Given the discussion outlined in the reply to reviewers I think this sentence in the introduction (line 43-44) should be removed:

"However, many recorders rely on populational analyses to obtain faithful decoding, losing crucial single-cell information."

I don't think it is fair to contrast with previous methods like this given that the present work also largely rely on population analyses to obtain useful information (at this stage).

No further comments.

Referee #4

(Remarks to the Author)

I co-reviewed this manuscript with one of the reviewers who provided the listed reports.

Referee #1 (Remarks to the Author):

In this manuscript, the authors have developed an *in cellulo* recording platform termed Granularly Expanding Memory for Intracellular Narrative Integration (GEMINI), which utilizes a computer-designed two-component self-assembly complex to record cellular activity history. This self-assembly complex is relatively large, with bright and easily observable signals, enabling high temporal resolution that surpasses benchmark fluorescent transcriptional reporters by two orders of magnitude. The authors first expressed GEMINI in cultured cells, demonstrating the system's stability and accuracy, then successfully applied it *in vivo* to record transcriptional changes and cellular activity during PTZ-induced seizure behavior in the mouse brain.

Overall, I found this to be a solid body of work that reports the development and application of a useful new tool. The manuscript is well-written, logically explained. However, it appears that the authors may have prepared the manuscript in haste, as the arrangement of the figures is suboptimal. Substantial improvements and careful proofreading are needed to ensure accuracy and clarity. Additionally, the authors need to further explore and demonstrate how this tool can address and uncover new biological questions. I would recommend accepting this manuscript for publication following revisions to address the comments below.

R: We thank the Reviewer for the positive feedback. We are encouraged that the Reviewer found the GEMINI platform to be a solid and innovative tool. We also appreciate the Reviewer's note regarding figure organization. In response, we carefully reviewed the overall structure and made significant revisions to improve clarity, logical flow, and readability.

Specifically, we made the following changes:

- (1) The characterization of GEMINI growth behaviors and properties (**original Fig. 2a–f**) has been integrated into the **new Fig. 1 (e–j)** alongside the conceptual overview (**Fig. 1a–c**). The revised figure is now titled “Concept and characterization of GEMINI.”
- (2) The *in vitro* recording of artificial signals (via JF dye addition) and NFκB signaling has been consolidated into the **new Fig. 2**, which emphasizes GEMINI's core recording capabilities in live cells. The figure title has been updated to “*In cellulo* recording using GEMINI.”
- (3) Multi-event recording (**original Fig. 3g–i**), fast dynamics, and signal amplitude measurements (**original Fig. 4**) have been combined into the **new Fig. 3**, all of which pertain to temporal dynamics of cellular signaling. This unified presentation improves thematic coherence. The figure is now titled “Recording multiple events and dynamics in live cells.”

We hope these revisions enhance the manuscript's clarity and readability. Regarding other critiques, please find our point-by-point responses to the critiques below.

Major critiques:

1. This paper presents the use of two-component protein cages, GEMINI-A and GEMINI-B, to form an assembly complex, with the two components linked via the P2A sequence to ensure equal expression levels. This makes me wonder why the authors did not consider designing a single-component self-assembling protein for the GEMINI system. A single component could potentially reduce the number of exogenous genes introduced into the cells, allow for greater packaging capacity in the AAV vector, and simplify AAV packaging. It would be beneficial for the authors to discuss whether they tried that or is that possible in principle.

R: This is an excellent point. We agree that single-component assemblies could offer advantages, such as reducing the number of exogenous genes, increasing AAV packaging efficiency, and simplifying the overall delivery strategy. However, our initial focus on two-component assemblies was driven by practical considerations during development and screening. Specifically, single-component variants are often difficult to purify due to their tendency to spontaneously precipitate in bacterial cultures. In contrast, the two subunits of our two-component system can be produced separately and only assemble upon mixing, greatly facilitating *in vitro* screening and identification of suitable candidates for the initial demonstration of this platform.

[REDACTION]

It is worth noting that, despite being a two-component system, GEMINI's subunits are relatively small (GEMINI-A: 19.0 kDa; GEMINI-B: 13.4 kDa) and fall well within the size range of widely used genetically encoded reporters such as FLARE (99 kDa, Addgene #92213) and CaMPARI (47.1 kDa, Addgene #60421). In this study, we successfully packaged all components into 2–3 AAV vectors and demonstrated robust *in vivo* recording of physiologically relevant cellular events.

In all, while we recognize the importance of simplifying recorder design, we believe GEMINI, at its current form, has already offered a powerful and practical platform for cellular activity recording. Its successful demonstration establishes a strong foundation for future iterations, including more compact, single-component variants.

To clarify this point, a brief discussion of the GEMINI size and the comparison to other existing reporters of cellular history have been included (**Page 27, Line 632**):

“The obtained scaffold is relatively compact, with its subunits (GEMINI-A/B: 19.0/13.4 kDa) comparable in size or even smaller than several genetically encodable reporters for cellular history^{6,7}.”

Furthermore, we listed the development of more compact, single-component assembly recorder as a potential future direction (**Page 29, Line 701**):

“In addition, it would be valuable to achieve (1) recorders with a smaller subunit, possibly using single-component scaffolds, (2) multiplexed recording of diverse signaling pathways...”

2. In the main figures, the GEMINI strategy constructs are not depicted, and they only appear in the extended figures. This makes it inconvenient for readers to fully understand the design and how the complex constructs are applied in different experiments. It would be helpful to include at least some of the GEMINI strategy constructs (possibly accompanied with cartoons) in the main figures to improve the clarity and context of the experiments.

R: We thank the Reviewer for the suggestion. We agree that including the GEMINI strategy constructs in the main figures will improve the readability of the manuscript.

Therefore, we have added the constructs for building the clonal GEMINI cell lines to **Fig. 1d**, and the constructs for building the clonal NF κ B-GEMINI cell lines to **Fig. 2e**.

3. In this paper, the authors do not fully demonstrate the advantages of GEMINI compared with previously published linear recording tools through a side-by-side comparison. For example, it would be helpful to show whether the 3D structure provided by GEMINI is significantly more accurate than the 1D approach used under the same conditions. Comparisons in other aspects are also feasible. These comparisons would help clarify the advantages and unique contributions of GEMINI.

R: We thank the Reviewer for the suggestion. Indeed, a side-by-side comparison with linear recording tools will be better to highlight the unique advantages of GEMINI. To achieve this, we performed further experiments to validate the effect and reorganize figures to highlight the advantages, which are summarized below.

One key distinction lies in the geometry of recording. Linear recorders encode information along a single axis, which poses major challenges for high-throughput fluorescence imaging. Only a small fraction of linear recorders happens to align horizontally with the imaging plane and can be captured with single-plane imaging (**Fig. R2a**). In most cases, recorders are tilted or oriented out-of-plane, requiring time-consuming volumetric imaging to reconstruct their signals (**Fig. R2a**). This greatly limits scalability and practical application.

In contrast, GEMINI assembles isotropically in 3D. Its granular geometry allows temporal information to be retrieved from any radial axis, regardless of orientation, enabling efficient and scalable single-plane imaging (**Fig. R2b**). While our original manuscript alluded to this distinction, it lacked a direct experimental comparison. To address this, we conducted additional experiments comparing GEMINI with the well-characterized linear recorder iPAK4 in 3D culture.

We cultured clonal HEK293 cells expressing GEMINI or iPAK4 in 3D. Similar to the experiment in **Fig. 2d**, two timestamps were applied at $t=0$ h and 8 h, respectively, while an artificial signal was introduced at $t=4$ h. As expected, both GEMINI particles and iPAK4 fibers adopted random orientations in 3D (**Fig. R2d,e**). While GEMINI signals were readily retrieved from a single mid-plane image, a substantial number of iPAK4 fibers required volumetric imaging, often exceeding 200 z-planes, to achieve comparable resolution (**Fig. R2f**). For in-plane iPAK4 ($<20^\circ$ tilt), and GEMINI particles, decoding could be done efficiently in 2D. However, for out-of-plane iPAK4 ($>45^\circ$ tilt), signal recovery became significantly more laborious.

We also assessed decoding accuracy across orientations. Despite the imaging burden, out-of-plane iPAK4 fibers showed similar decoding accuracy as in-plane fibers, indicating that volumetric imaging compensates for orientation (**Fig. R2g**). Finally, a direct comparison of decoding accuracy between GEMINI and iPAK4 (both in-plane and out-of-plane) showed no statistically significant difference (**Fig. R2h**), confirming that GEMINI's gain in imaging throughput does not compromise precision.

Fig. R2 Comparison of linear recorders and GEMINI in signal retrieval. **a**, Schematic of imaging linear recorders by fluorescence microscopy. When the recorder happens to align horizontally, they can be imaged at high resolution. However, for most recorders that are tilted out-of-plane, volumetric imaging is needed to retrieve the recorded patterns, which is slow and exhibits much lower resolution. **b**, Schematic of imaging GEMINI by fluorescence microscopy. High-resolution fluorescence patterns are retrievable regardless of the spatial orientation of GEMINI particles, allowing fast signal retrieval from a large population of cells. **c**, A linear recorder

tilted at an angle in culture. Scale bar: 50 μm . **d,e**, 3D cell culture expressing GEMINI (**d**) and iPAK4 linear recorder (**e**). Scale bars: 20 μm . **f**, The decoded time and error of iPAK4 linear recorder as a function of the tilted angle. Timestamps were applied at 0 and 8 hours, while the artificial signal was introduced at 4 hours. **g**, Comparison of the decoded time from in-plane iPAk4, out-of-plane iPAk4, and GEMINI. **h**, Average number of z-planes needed to obtain complete fluorescent profiles from individual recorders. Whiskers: s.d..

To highlight the results, **Fig. R2** was included in the revised manuscript as **Extended Data Fig. 2**. We also added the discussion to the manuscript accordingly (**Page 10, Line 223**):

“In contrast, linear recorders encoded information in a single direction, and a slight tilt from the focal plane significantly deteriorated imaging quality (**Extended Data Fig. 2a**). In 3D culture that mimics tissues, many linear recorders were oriented out-of-plane, posing challenges on scalable decoding (**Extended Data Fig. 2e**). Although volumetric imaging can improve the mapping of out-of-plane recorders with minimal impact on decoding precision, it substantially lowers throughput (**Extended Data Fig. 2f-h**).”

Another key advantage of GEMINI is its minimal perturbation to cell proliferation. In contrast, linear recorders can significantly deform cells by stretching the plasma membrane. In addition, their rigid, elongated structure imposes geometric constraints that interfere with cell division. In our original manuscript, we characterized cell division in both GEMINI- and iPAK4- expressing cells independently. While we noted the issues associated with linear recorders, a direct comparison was not included. To address this, we have reorganized the data and now present the side-by-side comparison in a single figure to clearly highlight the different behavior during cell division (**Fig. R3**). This figure has been included in the revised manuscript as **Extended Data Fig. 3**.

In our main text, we also rewrote the discussion accordingly (**Page 12, Line 266**):

“Moreover, we examined cell proliferation by tracking cell density over time. Comparable proliferation rates were observed between cells with and without GEMINI nucleation (**Extended Data Fig. 3a,b**), suggesting a negligible influence of GEMINI growth on cell proliferation. We then further investigated the influence of GEMINI on division at the cellular level, observing no apparent disruption of mitosis and cytokinesis following GEMINI nucleation. After division, GEMINI particles entered one of the daughter cells (**Extended Data Fig. 3c**). In contrast, cytoplasmic nucleation of linear recorders like iPAK4 was found to disrupt proliferation by preventing cytokinesis (**Extended Data Fig. 3e,f**).”

Fig. R3 a, Images comparing the cell density with (top) and without (bottom) GEMINI growth at $t = 0$ h (left) and 45 h (right). Scale bars: 100 μm . **b**, Comparison of the cell proliferation rate between the GEMINI+ and - groups. The two groups show comparable proliferation rate. **c,d**, Snapshots of the cell division processes of live cells with (c) and without (d) intracellular GEMINI nucleation, showing GEMINI nucleation has negligible impact on the mitosis and cytokinesis. **e**, Schematic showing the topological constraint imposed by intracellular linear recorders that obstruct the abscission during cell division. **f**, Snapshots of HEK293T cells with an intracellular

linear recorder during the division processes. Though mitosis was successful, the contractile ring failed to split the cell into two daughter cells and, therefore, resulted in one cell with two nuclei after division. Scale bars: 20 μm .

4. In Figure 4b, the authors use a destabilization method to improve temporal resolution, but the specific strategy is not clearly outlined in the main text or figure, making it difficult to understand how it was implemented. If the strategy is the same as that presented in the Fig. S10 figure legend (where P1N4 was used), this should be referenced in the main text for clarity.

R: We apologize for the ambiguity in our original manuscript. While we briefly introduced the concept of the destabilization strategy, we did not fully describe how it was implemented in the main text. To clarify, the destabilization method in **Fig. 4b** indeed exploited the P1N4 domain, as previously shown in the **original Fig. S10 (new Fig. S13)**.

We recognize that the underlying rationale and mechanism, particularly how this strategy improves temporal resolution, may not be clear to all readers. To address this, we have prepared a schematic illustration (**Fig. R4**) to improve clarity, which compares the behavior of soluble signal building blocks in systems with and without the P1N4 destabilization domain. In GEMINI constructs lacking a destabilization domain, residual signal building blocks remain in the cytoplasm after signal termination and are gradually incorporated into GEMINI particles, resulting in a slowly decaying fluorescent band. By contrast, the inclusion of the P1N4 domain accelerates degradation of the soluble signal BBs, enabling their rapid clearance from the cytoplasm. This leads to a sharper drop-off in the recorded signal band and effectively resets GEMINI for recording subsequent events in a shorter time window.

To better convey the implementation and benefits of the destabilization strategy, **Fig. R4** has been included in the revised manuscript as **Fig. S13**. Technical details of destabilizing strategy were also described accordingly in the main text (**Page 17, Line 393**):

“Although capturing the hour-level dynamics is adequate for mapping diverse cellular events, many signaling events exhibit faster dynamics, requiring further optimization of the recorder. **We hypothesized that this could be achieved by accelerating the turnover of cytoplasmic reporter subunit and its mRNA (Fig. S13a)**. This would produce a sharp decay in the signal band upon deactivation and reset the system more rapidly for capturing subsequent events. To achieve this, we incorporated the P1N4 domain, which was reported to enhance protein turnover by an order of magnitude³⁵, to destabilize both the reporter subunit and its mRNA. A clonal HEK293T cell line expressing the destabilized reporter subunit, termed NFκB-GEMINI-Boost, was developed with this modification (**Fig. S13b**). This modification enabled the resolution of even faster dynamics, where distinct signaling events separated by as short as 15 minutes could still be distinguished (**Fig. 3e**).”

Fig. R4. a, Schematic illustration of the impact of mRNA and protein stability on resolving the temporal dynamics of cellular events. Destabilization of both mRNA and protein of signal-BB accelerates their removal from the cytoplasm when the signal is turned off, resulting in a sharper decay of the "OFF" signal in GEMINI particles. **b**, DNA constructs used in the development of NFkB-GEMINI-Boost cell lines. NFkB-GEMINI-Boost contains a destabilization domain P1N4 that expedites the turnover of mRNA and protein. **c**, Images of GEMINI particles grown in the NFkB-GEMINI cells without (left) and with (right) incubation in 10 ng mL⁻¹ TNF- α . **d**, Images of GEMINI particles grown in the NFkB-GEMINI-Boost cells without (left) and with (right) incubation in 10 ng mL⁻¹ TNF- α . TNF- α was added 24 hours after DOX induction. Scale bars: 20 μ m.

5. In Figure 6, the authors use GEMINI to record cellular activity during PTZ-induced seizure behavior in the mouse brain, which involves a strong, non-physiological stimulus. This raises the question of whether this tool could also be used to record activity under physiological conditions, such as visual stimuli, feeding, or motor behavior. What is the sensitivity of this tool for detecting activity under more naturalistic, physiological conditions?

R: We agree with the Reviewer that the platform will be even more meaningful if we can test GEMINI's ability to record activity under more physiological conditions. However, we also note that natural behaviors such as vision, movement, and feeding generate continuous, spontaneous neural activity that is difficult to quantify and varies substantially across animals. Moreover, designing such experiments would require the development of new behavioral paradigms and ACUC-approved protocols, which are beyond the timeline of the current revision.

To address this important point within our current scope, we employed an alternative *in vivo* model based on HEK293T xenografts expressing NFκB-GEMINI. Specifically, we subcutaneously implanted GEMINI-expressing cells into immunodeficient mice and triggered acute inflammation using intraperitoneal (i.p.) injection of lipopolysaccharide (LPS), a well-established model for inflammation relevant to disorders such as sepsis (*Biomedicine & Pharmacotherapy* **141**, 111890 (2021)) and neuroinflammation (*Glia* **55(5)**, 453-462 (2007)). This model is much less strong and more physiologically relevant than the PTZ-induced seizure.

Following a 10-day engraftment period, we initiated GEMINI expression with doxycycline (DOX) and examined NFκB signaling in response to 3 mg kg⁻¹ LPS. Tumor-wide expression of GEMINI and cellular response to inflammation was observed (**Fig. R5a**). We further investigated the cellular responses to various doses of LPS, from 0.03 to 3 mg kg⁻¹. As shown in **Fig. R5b-c**, GEMINI recorded robust, dose-dependent NFκB activation, with sensitivity sufficient to detect inflammation induced by LPS doses as low as 1% of that commonly used in the literature for inflammation modeling (*Int. J. Biol. Macromol.* **279**, 135371 (2024)). These results highlight GEMINI's ability to resolve mild and graded physiological signals across a large tissue.

We then investigated if GEMINI could decode the timing of transient inflammatory events *in vivo*. Through retro-orbital injection of Janelia Fluor dyes, we applied two timestamps 48 hours apart, where clear timestamp bands were observed in most GEMINI particles (**Fig. R5d-e**). A dose of LPS administered at different times between the timestamps could be temporally decoded using the established model, and decoding results revealed consistent alignment between the timing of LPS administration and the peak NFκB activation (**Fig. R5g-h**). The temporal precision was on the order of one day, lower than

in vitro, which is likely due to heterogeneous cytokine diffusion, timestamping variability, and differential cellular responses.

Fig. R5. **a**, Experimental procedure for *in vivo* recording of LPS-induced inflammation. **b**, A section of a HEK293T xenograft showing GEMINI recording of NFκB activation following systemic injection of 3 mg kg⁻¹ LPS. Vasculature was labeled by anti-CD31 immunostaining (red). Insets: magnified 3D rendering of the central region (left) and a heatmap displaying the normalized GFP intensity ($F_{\text{GFP}}/F_{\text{HTL}}$) of individual particles (right). Scale bars: 1 mm, 100 μm (inset). **c**, GEMINI particles recording NFκB signaling at various LPS doses (left) and the dose-dependent signal intensity (right). Scale bar: 100 μm. **d**, Experimental procedure for *in vivo* timestamping via systemic injection of HTL dyes. **e**, Fluorescence images of GEMINI particles in xenografts labeled with two timestamps; inset: magnified image of a GEMINI particle. Scale bars: 20 μm, 2 μm (inset).

f, Experimental procedure for recording NFκB signal with LPS administered at various times. **g**, Representative images (left) and fluorescence profiles (right) of GEMINI particles recording NFκB activation in mice injected with LPS on days 0, 1, or 2. Scale bar: 2 μm. **h**, Decoded timing of NFκB activation peaks in groups receiving LPS on days 0–2 (n = 52/59/55 for t = 0/1/2 days; 3 mice per group). **i,j**, 3D images of xenograft tissue with high (**i**) and low (**j**) vasculature density after LPS administration on day 0. Scale bars: 100 μm. **k**, Mean fluorescence profiles of NFκB signals recorded in GEMINI particles located in regions with high (solid) and low (dashed) vasculature density. **l**, Decoded peak timing of NFκB activation for GEMINI particles from high- and low-vascularized regions (n = 42/47 for high/low; 3 mice). **c,h,i**, Box bounds: the 25th and 75th percentiles; whiskers: minimum and maximum; squares: mean; and center lines: median.

Given the known heterogeneity in vascularization within HEK293T-derived xenografts, we hypothesized that vascular density might influence cellular exposure to cytokine and thus affect signal timing. To test this, we labeled vasculature in xenograft using anti-CD31 staining and stratified the xenograft regions based on the vascular density. We performed temporal decoding separately for GEMINI particles in high (>8% volume occupation) and low (<2% volume occupation) vascularized regions. As expected, poorly perfused areas exhibited significantly delayed NFκB activation (**Fig. R5i–l**). These results showcase GEMINI’s power to capture spatial heterogeneity in signaling dynamics at the cellular level across intact tissue.

The temporal resolution achieved in the xenograft model is much higher than that was demonstrated in the mouse brain. However, it is still lower than that achieved *in vitro*. This difference can be attributed to: (1) intrinsic tissue heterogeneity, (2) reduced accuracy of timestamping, (3) lower imaging quality, and (4) lower accuracy of the decoding model *in vivo*. While the first factor is intrinsic to biological systems, the others can likely be addressed through future advances such as more bioavailable timestamping reagents, alternative timing strategies (e.g., using cell-cycle markers), improved imaging via tissue clearing or expansion, and machine learning–based decoding algorithms trained on *in vivo* datasets.

In summary, these new results address the Reviewer’s concerns regarding GEMINI’s sensitivity and ability to resolve more physiologically relevant events *in vivo*. Furthermore, they highlight GEMINI’s unique capacity to decode spatially heterogeneous cellular signaling history, providing valuable insight into processes such as cytokine diffusion and differential cellular responses.

To strengthen the manuscript, we incorporated this new study into the revision, with **Fig. R5** now included as **Fig. 4**. A new section titled “**GEMINI records inflammatory responses *in vivo***” has been added to the main text. A figure illustrating tissue slices from a representative xenograft with heterogeneous NFκB activation has been included as **Extended Data Fig. 5**. In addition, the abstract and introduction were revised to reflect the new results:

Abstract: “In a xenograft model, GEMINI recorded inflammation-induced signaling dynamics across tissue with cellular resolution, revealing spatial heterogeneity linked to vascular density.” (Page 2, Line 30)

Introduction: “Using an *in vivo* xenograft model, we demonstrated GEMINI’s ability to record both the amplitude and temporal information of inflammation-induced signaling and to correlate cellular responses with local vascular density.” (Page 5, Line 101)

6. In this paper, the authors have validated the usage of GEMINI *in vivo*, but I feel the practicality of this tool is not fully demonstrated. I suggest they need to further explore some new biological questions or discoveries that this tool could address.

R: We thank the Reviewer for the suggestion. We agree that applying the GEMINI platform to biologically relevant questions is essential for demonstrating its practical utility and helping researchers envision how it can be integrated into their own work. However, as the central focus of this study is the development of a novel recording tool, we aimed to avoid delving too deeply into specific biological questions that might shift the emphasis away from tool innovation.

We believe the *in vivo* xenograft model added in this revision, as discussed in our response to Critique #5, strikes an appropriate balance. It serves as a biologically meaningful demonstration of GEMINI's capabilities while remaining aligned with the manuscript's primary goal of technological development.

The inflammation model induced by LPS is widely used to mimic inflammatory conditions, such as sepsis (*Biomedicine & Pharmacotherapy* **141**, 111890 (2021)) and neuroinflammation (*Glia* **55(5)**, 453-462 (2007)). Likewise, HEK293T-derived xenografts are commonly used to model aspects of solid tissues and tumors. Although the systemic dynamics of LPS-induced cytokine release and inflammation have been characterized, the spatiotemporal heterogeneity of cellular response within intact tissues toward systemic inflammation remains underexplored due to the lack of suitable tools.

Using GEMINI, we found that NFκB signaling in individual cells within the xenograft peaked approximately one day after LPS administration, significantly later than peak cytokine levels reported in serum (*Agents and Actions* 39(S1) C52-C54 (1993)). This delay highlights the physical barrier posed by solid tissue to cytokine diffusion. In addition, we observed that vascular density strongly influences the spatial heterogeneity of inflammatory signaling, indicating that microvascular architecture is an important factor in tissue-level signaling dynamics and therapeutic response.

To clarify this point and further highlight the practicality of GEMINI, we included further discussion in the main text (**Page 28, Line 672**):

“Using a xenograft model, we showed GEMINI's potential for tissue-wide, cellular level recording of signaling history. This study revealed pronounced cellular heterogeneity in NFκB activation in response to systemic inflammation and highlighted heterogeneous vascularization as an important contributing factor. This approach, combining engineered cells and *in vivo* transplantation, offers a promising platform for preclinical research,

particularly in developing tumor models to map oncogenic signaling and evaluate therapeutic responses.”

7. Based on Extended Fig. 6c and Fig. S4, as time progresses, the radius of the assembled particles increases, reaching up to 3 μm . I wonder if this could impact the physiology or excitability of neurons. The authors may consider addressing this by recording and comparing the calcium signaling or electrophysiological properties between GEMINI-expressing neurons and non-GEMINI-expressing neurons. This could provide valuable insights into potential effects on neuronal activity. Of course, the existing cell viability and behavioral tests can help readers understand the impact of GEMINI on cells or mice from different perspectives.

R: We thank the Reviewer for raising this point. To directly assess whether GEMINI expression affects neuronal physiology, we performed *in vivo* two-photon calcium imaging to compare neuronal activity between GEMINI+ and GEMINI– neurons.

In this study, GEMINI was expressed in the primary visual cortex of *Thy1-jRGECO1a* transgenic mice (JAX strain#: 030526), via intracranial AAV injection (**Fig. R6a**), and calcium imaging was conducted on day 21 (**Fig. R6b**). During imaging, mice were head-fixed on a running wheel.

We recorded calcium transients from both GEMINI+ and GEMINI– neurons and observed similar overall firing behaviors (**Fig. R6c,d**). Quantitative analysis revealed that the maximal $\Delta F/F$ of calcium responses were also comparable between the two groups (**Fig. R6e**). These results indicate that GEMINI expression does not impair spontaneous neuronal activity under these experimental conditions.

To clarify this point, **Fig. R6** was added to the manuscript as **Extended Data Fig. 9h–i**. A further discussion of the data was added to the main text accordingly (**Page 22, Line 525**):

“To assess possible functional impact of GEMINI expression, we performed *in vivo* two-photon calcium imaging in *thy1-jRGECO1a* transgenic mice injected with btAAV in the primary visual cortex (**Extended Data Fig. 9h–i**). GEMINI+ neurons exhibited calcium transient patterns and $\Delta F/F$ of calcium responses comparable to GEMINI– neurons, suggesting minimal impact of GEMINI on spontaneous neuronal activities.”

Fig. R6 a, Schematic of the experimental procedure for *in vivo* calcium imaging. **b**, Low-magnification image of the primary visual cortex region of a Thy1-jRGECO1a transgenic mouse showing high-level expression of GEMINI (green). Scale bars: 100 μm . **c,d**, Representative traces of the calcium transient of GEMINI+ (**c**) and GEMINI- (**d**) neurons. The fluorescent images at 1 and 2 are shown on the right. **e**, Statistical comparison of the $\Delta F/F_0$ of Ca^{2+} peak maxima between GEMINI+ and GEMINI- neurons. **d-g,i**, Box bounds: 25th and 75th percentile; whiskers: minimum and maximum; squares: mean; and center lines: median.

8. Similar to the analysis of decoded time in Figure 2 and 3, have the authors conducted similar experiments and analysis *in vivo*? What temporal resolution can GEMINI achieve *in vivo*? It would be helpful if the authors could present some results or elaborate on this in the discussion section.

R: We thank the Reviewer for the suggestion. Indeed, we recognized that the *in vivo* temporal resolution of GEMINI recording in the brain is much lower than our *in vitro* study, which could be attributed to several factors: (1) intrinsic heterogeneity of neurons within brain tissue; (2) variability in GEMINI expression due to local AAV injection; (3) animal-to-animal variation; (4) reduced accuracy of *in vivo* timestamping; (5) lower imaging quality of GEMINI particles *in vivo*; and (6) lower accuracy of the decoding model *in vivo*. Therefore, we are yet to confidently decode robust absolute chronological information for individual particles. While we believe these challenges can be addressed with further optimization (e.g. systemic expression, new timestamping modality, tissue clearing, and modified decoding model), it necessitates years of effort and falls outside the current scope and timeline of this revision.

In contrast, the xenograft model yielded significantly better temporal resolution. As discussed in our response to Critique #5, we conducted *in vivo* temporal decoding analyses similar to those used *in vitro*. This model enabled GEMINI to report relative signal amplitudes in response to varying LPS doses (**Fig. R5c**) and to resolve the timing of NFκB activation with a temporal precision of approximately one day (**Fig. R5f-h**).

We believe the improved resolution in xenografts compared to that in the brain tissue stems from several factors: (1) higher cellular homogeneity due to the use of a clonal HEK293T line; (2) more facile diffusion of JF dyes into the xenograft tissue than brain, which supports enhance timestamping precision; and (3) the same cell background as our *in vitro* system, allowing better transferability of the decoding model.

To clarify this point, the results obtained from the *in vivo* xenograft study were added to the manuscript **as described in our response to Critique #5**. We further elaborated on the discussion of the existing difficulties in achieving high-resolution recording in the brain (**Page 29, Line 680**):

“It is noted that the temporal resolution of recording in the brain was lower than that observed in culture or xenografts, which could be attributed to factors such as the inherent heterogeneity of neurons, less accurate timestamping, variability in GEMINI expression, and less accurate predictive model for decoding. Disentangling these factors remains an ongoing challenge; nevertheless, GEMINI still holds great potential for further optimization in native tissues.”

Minor critiques:

1. In Extended Fig. 1c, the assembly images of variants #3-v1 and #3-v3 appear unclear and difficult to interpret. Based on the fluorescent images in Extended Fig. 1d, #3-v1 and #3-v3 show a similar size to #3-v2, so I expected to see similar assembly patterns, but this is not evident in the images. Could the authors provide an explanation for these results or present more representative data?

R: Sure! While we found that *in vitro* assembly under conditions mimicking the cytoplasmic ionic strength generally predicts *in cellulo* assembly behavior, the resulting morphologies can differ substantially. In our *in vitro* experiments, variants #3-v1 and #3-v3 produced assemblies that were significantly smaller than those of the other variants. To ensure a fair comparison across conditions, all images in **Extended Data Fig. 1c** were acquired at the same low magnification. As a result, the smaller assemblies of #3-v1 and #3-v3 are difficult to discern at that scale.

We acknowledge that the visualization of these smaller particles was not optimal in the original figure. To address this, we have now included **magnified insets** for all images, highlighting the morphology of each variant. These updated panels are presented in **Extended Data Fig. 1c** in the revised manuscript.

2. In Figure 2j and Extended Fig. 4d, the meaning of the arrows in the figure is unclear. Could the authors provide an explanation for what these arrows indicate in the figure legend? Clarifying this would improve the figure's interpretability.

R: We apologize for the confusion. As there are two distinct sets of the data, namely the distribution of the decoded time and the corresponding standard deviation, shown in the same plot, the arrows are intended to indicate which axis is used for each data. Specifically, the **bottom x-axis** should be used to interpret the box plots and distribution curves, while the **top x-axis** corresponds to the standard deviation values.

To clarify this in the manuscript, we have modified the figure caption describing the **Fig. 2d (original Fig. 2j)** and **Extended Data Fig. 4d** as:

Fig. 2: “d, Decoded onsets from individual GEMINI particles (**bottom-x**, $n = 258/275/279/261/225/219/275$ for $t = 2/3/4/5/6/7/8$ h) and the standard deviations (**top-x**) for each group.”

Extended Data Fig. 4: “d, Decoded onsets from individual GEMINI particles (**bottom-x**, $n = 100/108/101/96/99$ for $t = 6/12/18/24/30$ h) and the standard deviations (**top-x**) for each group.”

3. In Figures 3b and 3e, the sequence in which the different stamps are exhibited appears a bit unusual, as stamp 3 is added last but exhibited before stamp 2. If possible, could the authors consider rearranging the presentation for improved readability? One potential suggestion would be to reorder the stamps to align with the chronological appearance of the signals, which might help clarify the progression.

R: Our original intent was to place stamp 2 adjacent to the TNF- α -induced NF κ B activation channel, as it was applied concurrently and served as an “accompanying timestamp” for that specific signaling event. This arrangement was meant to facilitate direct comparison of the spatial alignment with the NF κ B signal. However, we understand that this layout may cause confusion. Therefore, we updated the channel order in the revised manuscript as Stamp 1→Stamp 2→NF κ B→Stamp 2+NF κ B→Stamp 3, following the temporal order of the experimental procedures. Since we have reorganized the figures, the update has been included in the revised version as **Fig. 2g and 2j**

4. In Figure 3g, two different JF dyes are used, but they are not labeled in the schematic above. Additionally, the presentation style is inconsistent with Figures 3b and 3e.

R: We agree that explicitly labeling the JF dyes used in the schematic enhances readability. This information has now been added to the revised figure. As we reorganized the figures substantially, the update is now included in **Fig. 3a**.

The presentation style of the data was also updated throughout the manuscript to ensure consistency. In the revised version of the figures, Green was utilized universally to illustrate the cell signal bands in GEMINI. When only two dyes were used, red and cyan were utilized to enhance the contrast among the channels. When three dyes were used, yellow, cyan, and violet were used. Specifically, in the **new Fig. 2a–c**, the colors of the timestamps now match those used in **new Fig. 2f–j**. Similarly, the colors in **Extended Data Fig. 4** were updated accordingly. We believe these revisions provide better stylistic consistency of the data.

5. In Figure 4c, the concentration of TNF- α used in the bottom row of fluorescence images appears to be missing. Additionally, the meaning of “12x” and “10x” in the figure should be clarified. In Figure 4d, the concentration labels at the top and the fluorescence intensity measurements recorded over time at the bottom are not properly aligned, which makes it challenging to interpret the data.

R: We apologize for the confusion. In the figure, the top and bottom rows of images are the same set of particles. The labels 1x means the original exposure level of the images,

while 12x means 12 times higher exposure of the particle images. For the low-dose conditions, the intensity of the bands was only visible at a much higher exposure level. However, if we used a high exposure for all conditions the high-dose conditions will be saturated. Therefore, we present two rows of images with distinct exposure level for clarity.

To clarify this point, instead of writing “1x” and “12 x” in the figure, we wrote “1x exposure” and “12x exposure”. We also added a note to the figure caption accordingly:

“Dose-dependent intensity of signals recorded by GEMINI. Inset: images of **single GEMINI particles capturing signal amplitudes at various doses. The images in the top row are the same particles as the bottom, but with a 12 times of exposure level.** Circles: mean; whiskers: s.d..”

6. In the Extended Fig. 4 figure legend, the numbering for panels 4c and 4d is incorrectly written as "i" and "j".

R: We apologize for the typo in the caption. They have been updated accordingly.

7. In Extended Fig. 6b, is the expression time the same across the cortex, DG, and CA1 regions? It appears that the GEMINI particles are notably larger in the DG group, even under the same scale bar. Could the authors explain the reasons?

R: The expression time is the same across different brain regions. We believe the larger GEMINI particles observed in the DG region in **Extended Data Fig. 6b** likely result from variation in local AAV delivery. Although we followed a consistent injection protocol across all animals and brain regions, minor differences in local viral spread or uptake may lead to region-specific variability in transduction efficiency.

A higher local dose of AAV could result in increased gene copy number per neuron, potentially accelerating GEMINI particle growth. However, this appears to be an isolated case. We did not observe similar phenomenon in other DG samples, including those shown in **Fig. 5** or the IHC images in **Extended Data Fig. 8e**.

8. In Line 403, you mention Extended Fig. 7c, but I cannot find this figure.

R: We apologize for the oversight. The correct reference was **Extended Data Fig. 6c** in the original manuscript. In the revised manuscript, **Extended Data Fig. 7c** is the correct citation. The manuscript underwent multiple rounds of revision and figure reorganization prior to submission, and we missed updating this particular citation to reflect the final figure structure. We have now corrected the reference in the revised manuscript to ensure consistency. Thank you for catching this!

Referee #2 (Remarks to the Author):

Yan and colleagues described an interesting way of recording temporal events in living cells (termed GEMINI) by growing a three-dimensional structure using a computationally designed protein as the basic unit. The unit can be fluorescently labeled by HALO dyes or expressed as an FP-fusion protein from promoters that can be temporally controlled. This 3D recorder has several advantages over previously described linear recorder in reduced toxicity and ease of decoding (by imaging from all angles), etc. While the design of GEMINI is innovative and interesting, and some of the *in vitro* characterization impressive, the application is quite limited (see specific critiques below), particularly *in vivo*. For a new method to be published in Nature, we should expect that it can solve a problem that cannot be solved by other methods. At this point, GEMINI is quite far from that. We hope that our critiques below are nevertheless useful for the authors to improve the method and/or its presentation.

R: We sincerely thank the Reviewer for positive comments and constructive feedback. We are encouraged that the Reviewer found the design of GEMINI innovative and our *in vitro* characterization impressive.

We acknowledge the Reviewer's concern regarding the limited scope of *in vivo* applications in the original version of the manuscript. As the primary focus of this work is on establishing a new intracellular recording platform, our initial emphasis was placed on demonstrating the technical feasibility, biocompatibility, and recording performance of GEMINI in both *in vitro* and *in vivo* contexts. Nonetheless, we fully agree that demonstrating the platform's ability to uncover biological insights that are otherwise difficult to access is essential to establishing its broader relevance and utility.

To address this important point, we explored an additional *in vivo* model based on HEK293T xenografts expressing NFκB-GEMINI. Specifically, we subcutaneously implanted GEMINI-expressing cells into immunodeficient mice and triggered acute inflammation using intraperitoneal (i.p.) injection of lipopolysaccharide (LPS), a well-established model for inflammation relevant to disorders such as sepsis (*Biomedicine & Pharmacotherapy* **141**, 111890 (2021)) and neuroinflammation (*Glia* **55(5)**, 453-462 (2007)). This model is less strong and artificial than the PTZ-induced seizure.

Following a 10-day engraftment period, we initiated GEMINI expression with doxycycline (DOX) and examined NFκB signaling in response to 3 mg kg⁻¹ LPS. Tumor-wide expression of GEMINI and cellular response to inflammation was observed (**Fig. R7a**). We further investigated the cellular responses to various doses of LPS, from 0.03 to 3 mg kg⁻¹. As shown in **Fig. R7b-c**, GEMINI recorded robust, dose-dependent NFκB activation, with sensitivity sufficient to detect inflammation induced by LPS doses as low as 1% of that commonly used in the literature for inflammation modeling (*Int. J. Biol. Macromol.*

279, 135371 (2024)). These results highlight GEMINI's ability to resolve mild and graded physiological signals across a large tissue.

We then investigated if GEMINI could decode the timing of transient inflammatory events *in vivo*. Through retro-orbital injection of Janelia Fluor dyes, we applied two timestamps 48 hours apart, where clear timestamp bands were observed in most GEMINI particles (**Fig. R7d-e**). A dose of LPS administered at different times between the timestamps could be temporally decoded using the established model, and decoding results revealed consistent alignment between the timing of LPS administration and the peak NFκB activation (**Fig. R7g-h**). The temporal resolution was on the order of one day, lower than *in vitro*, which is likely due to heterogeneous cytokine diffusion, timestamping variability, and differential cellular responses.

Given the known heterogeneity in vascularization within HEK293T-derived xenografts, we hypothesized that vascular density might influence cellular exposure to cytokine and thus affect signal timing. To test this, we labeled vasculature in xenograft using anti-CD31 staining and stratified the xenograft regions based on the vascular density. We performed temporal decoding separately for GEMINI particles in high (>8% volume occupation) and low (<2% volume occupation) vascularization regions. As expected, poorly perfused areas exhibited significantly delayed NFκB activation (**Fig. R7i-l**). These results showcase GEMINI's power to capture spatial heterogeneity in signaling dynamics at the cellular level across intact tissue.

This study exemplified capabilities enabled by GEMINI that are otherwise challenging. Here, the inflammation model induced by LPS is widely used to mimic inflammatory conditions, such as sepsis (*Biomedicine & Pharmacotherapy* **141**, 111890 (2021)) and neuroinflammation (*Glia* **55(5)**, 453-462 (2007)). Likewise, HEK293T-derived xenografts are commonly used to model aspects of solid tissues and tumors. Although the systemic dynamics of LPS-induced cytokine release and inflammation have been characterized (*Int. J. Biol. Macromol.* **279**, 135371 (2024)), the spatiotemporal heterogeneity of individual cells' response within intact tissues toward systemic inflammation remains underexplored due to the lack of suitable tools. Using GEMINI, we observed that vascular density influences the spatial heterogeneity of inflammation-induced NFκB signaling in tissues at the cellular level, indicating that microvascular architecture is an important factor in signaling dynamics and therapeutic response in tissues. Together, we believe it addresses the Reviewer's expectation that "it can solve a problem that cannot be solved by other methods".

In addition, this demonstration also showcased a generalizable method of building physiological models to investigate biological questions. For instance, one can follow the same strategy to develop clonal cancer cell lines stably expressing GEMINI for transplantation. Signaling dynamics of the pathways of interest during cancer

development could be resolved. In addition, a certain therapy can be given and GEMINI can capture their biochemical effects across the tumor at the cellular level. This would be extremely powerful for studying cancer biology and evaluating new anti-cancer therapies *in vivo*.

Fig. R7. **a**, Experimental procedure for *in vivo* recording of LPS-induced inflammation. **b**, A section of a HEK293T xenograft showing GEMINI recording of NFkB activation following systemic injection of 3 mg kg⁻¹ LPS. Vasculature was labeled by anti-CD31 immunostaining (red). Insets: magnified 3D rendering of the central region (left) and a heatmap displaying the normalized GFP intensity (F_{GFP}/F_{HTL}) of individual particles (right). Scale bars: 1 mm, 100 μ m (inset). **c**, GEMINI particles recording NFkB signaling at various LPS doses (left) and the dose-dependent signal

intensity (right). Scale bar: 100 μm . **d**, Experimental procedure for *in vivo* timestamping via systemic injection of HTL dyes. **e**, Fluorescence images of GEMINI particles in xenografts labeled with two timestamps; inset: magnified image of a GEMINI particle. Scale bars: 20 μm , 2 μm (inset). **f**, Experimental procedure for recording NF κ B signal with LPS administered at various times. **g**, Representative images (left) and fluorescence profiles (right) of GEMINI particles recording NF κ B activation in mice injected with LPS on days 0, 1, or 2. Scale bar: 2 μm . **h**, Decoded timing of NF κ B activation peaks in groups receiving LPS on days 0–2 ($n = 52/59/55$ for $t = 0/1/2$ days; 3 mice per group). **i,j**, 3D images of xenograft tissue with high (**i**) and low (**j**) vasculature density after LPS administration on day 0. Scale bars: 100 μm . **k**, Mean fluorescence profiles of NF κ B signals recorded in GEMINI particles located in regions with high (solid) and low (dashed) vasculature density. **l**, Decoded peak timing of NF κ B activation for GEMINI particles from high- and low-vascularized regions ($n = 42/47$ for high/low; 3 mice). **c,h,i**, Box bounds: the 25th and 75th percentiles; whiskers: minimum and maximum; squares: mean; and center lines: median.

To strengthen the manuscript, we incorporated this new study into the revision, with **Fig. R7** now included as **Fig. 4**. A new section titled “**GEMINI records inflammatory responses *in vivo***” has been added to the main text (**Page 18, Line 420**). A figure illustrating tissue slices from a representative xenograft with heterogeneous NF κ B activation has been included as **Extended Data Fig. 5**. In addition, the abstract and introduction were revised to reflect the new results:

Abstract: “**In a xenograft model, GEMINI recorded inflammation-induced signaling dynamics across tissue with cellular resolution, revealing spatial heterogeneity linked to vascular density.**” (**Page 2, Line 30**)

Introduction: “**Using an *in vivo* xenograft model, we demonstrated GEMINI’s ability to record both the amplitude and temporal information of inflammation-induced signaling and to correlate cellular responses with local vascular density.**” (**Page 5, Line 101**)

We would also like to emphasize that this study represents a major step toward the long-standing goal of spatiotemporal mapping of cell signaling dynamics across intact organs *in vivo*, a capability that holds transformative potential for cell biology, neuroscience, disease modeling, and drug discovery. While the present work may not focus on solving a specific biological question in depth (as its scope centers on technological innovation), we believe it indeed represents a significant leap toward spatiotemporal recording at the cellular level *in vivo* and lays the groundwork for next-generation platforms. We believe cell biologists and neuroscientists will be inspired by the capabilities shown in this work and explore new scientific questions that are otherwise difficult to approach.

We also recognize that GEMINI, as a platform in its early stages, is not yet perfect. There remains considerable room for improvement, including enhancing temporal precision, spatial coverage, and decoding accuracy, especially in the *in vivo* scenarios. Realizing the full potential of this platform will require dedicated, long-term effort from researchers across disciplines, from synthetic biology and protein design to advanced imaging and

data analysis. We believe this is the very nature of developing any tools: as with the building of Rome, new technologies evolve through iterative refinements and improvements over time.

We hope that the revisions and clarifications presented in the revised manuscript effectively address the Reviewer's concerns and convey both the current strengths and the future promise of GEMINI as a transformative tool for the study of dynamic cellular processes.

Regarding the specific concerns raised, we provide our detailed point-by-point responses below.

A. The “low cytotoxicity” is not well characterized.

1. In cultured cells, from the main text to the extended data and methods, the authors did not clarify how and when they assayed the live/dead ratio. Effects from a “particle” close to ~25% of the size of the nucleus should be assayed at different time points to identify the optimal working time window for GEMINI.

R: We thank the Reviewer for raising this point and apologize for the lack of certain details in our original manuscript.

In our initial submission, we performed the cell live/dead assay via calcein-AM (live) and ethidium homodimer-1 (EthD-1) staining at 48 hours after the induction of GEMINI expression (**Fig. R8a,b**). The live and dead cells were segmented separately to calculate the dead/live cell ratios.

In response to the Reviewer’s suggestion, we expanded this analysis to assess cell viability at multiple time points. We further compared the viability of the clonal GEMINI cell line with and without DOX treatment at 1, 2, 3, and 4 days post-induction (**Fig. R8c-e**). Beyond 96 hours, cell overgrowth and stacking introduced segmentation artifacts, limiting accurate quantification. Across all timepoints examined, we observed high viability with no significant differences between DOX+ and DOX- groups, indicating minimal cytotoxicity associated with GEMINI expression under our experimental conditions (**Fig. R8e**).

The results confirm that GEMINI growth has minimal impact on cell survival within the relevant recording window. They are also consistent with our *in vitro* temporal decoding experiments, which demonstrated accurate signal reconstruction even after 96 hours of GEMINI growth, despite a decrease in temporal resolution at later stages (**Extended Data Fig. 4**).

To clarify this point, the method describing the cell viability assay, as shown below, has been added to **Supplementary Methods** in the **Supplementary Information**, titled “**Live/dead cell viability assay**” (Page 7, Line 147 in SI).

The results of characterizing the cell viability at various time points (**Fig. R8**) were included in the revised manuscript as **Fig. S6**.

A brief description of the new viability test was also added to the Main Text (**Page 10, Line 233**):

“An ideal recorder should exhibit negligible interference with essential cell processes. **We first assessed the impact of GEMINI nucleation on survival of HEK cells using a live/dead assay (Fig. S6). After growing GEMINI for 4 days, no apparent increase in cell death was observed, indicating its low cytotoxicity (Fig. S6c-e).**”

Fig. R8 a, Images of cell live/dead assays on GEMINI HEK cell line with and without DOX, and wild-type (WT) HEK cells with and without DOX. The assays were performed at 48 hours post DOX induction. Green: live cells; Magenta: dead cells; Red: GEMINI. Scale bar: 200 μ m. **b**, Comparison of the dead/live cell ratio among the four groups. No significant difference was found between GEMINI growing cells with the other three control group. Error bars: standard deviation. **c,d**, Images of cell live/dead assays on GEMINI HEK cell line with (**c**) and without (**d**) DOX at 1-4 days after induction. Scale bars: 200 μ m. **e**, Comparison of the dead/live cell ratio between the GEMINI+ (DOX+) and GEMINI- (DOX-) groups on different days. No significant difference was found between the two group in each day.

2. If possible, co-staining GEMINI with different organelle markers (ER, Golgi, mitochondria, lysosomes, endosomes) at different time points would help assess cellular health at the subcellular level.

R: We thank the Reviewer for this suggestion. We agree that assessing cellular health at the subcellular level in the context of GEMINI expression is important, especially to rule out unintended perturbations to major organelles during GEMINI recording.

To address this, we employed the well-established **Cell Painting protocol** (*Nature Protocols* **18**, 1981–2013 (2023); Invitrogen Image-iT™ Cell Painting Kit) to co-stain GEMINI-expressing cells with a multiplexed panel of fluorescent markers targeting key subcellular compartments. These included: nucleus (Hoechst), endoplasmic reticulum (ConA), nucleoli and cytoplasmic RNA (Syto14), cytoskeleton (F-actin, Phalloidin), Golgi and plasma membrane (WGA), and mitochondria (MitoTracker). Imaging was performed at multiple time points from 0 to 48 hours post-GEMINI induction to monitor changes in organelle morphology and localization (**Fig. R9a**). No obvious changes were observed by visual inspection. To quantify potential effects, we segmented the stained regions and compared the areas across groups (**Fig. R9b-e**). While most markers showed no significant differences, we observed a slight decrease in nuclear area and a modest increase in mitochondrial area at 48 hours. We hypothesize these changes may be due to nuclear deformation from GEMINI growth and cell crowding at high confluency. In contrast, when we repeated the assay in U2OS cells, no significant change in any organelle markers was found (**Fig. R9f-j**). These results indicate that, though minor changes in cellular morphology was observed after extensive growth, GEMINI formation does not induce major subcellular stress.

To clarify this analysis, **Fig. R9** has been added as **Fig. S7** in the revised manuscript. Experimental details have also been included under the **Supplementary Methods** section in **Supplementary Information**, titled “**Cell Painting assay and segmentation.**” (**Page 7, Line 128 in SI**)

The discussion of the results was added to the main text accordingly (**Page 10, Line 236**):

“Next, we profiled the morphological features of subcellular structures during GEMINI growth via the Cell Painting assay³². While most compartments exhibited minimal structural changes, a decrease in the nuclear area and a subtle increase in mitochondrial area were observed at 48 hours after GEMINI expression (**Fig. S7a-e**), which could be due to potential physical interactions and changes in confluency. However, these changes were not observed in U2OS cells expressing GEMINI (**Fig. S7f-j**), suggesting that distinct cell types may respond differently to GEMINI.”

Fig. R9 a, Images of the Cell Painting assay on HEK293T cells with (top) and without (bottom) the induction of cytoplasmic GEMINI growth by DOX. **b-e**, Statistical comparison of the segmented area of Hoechst (**b**), ConA (**c**), Syto14 (**d**), and MitoTracker (**e**) at 0-48 hours after DOX induction. **f**, Images of the Cell Painting assay on U2OS cells with (top) and without (bottom) the induction of cytoplasmic GEMINI growth by DOX. Bright field images were shown to highlight the GEMINI particles. Scale bars: 20 μ m. **g-j**, Statistical comparison of the segmented area of Hoechst (**g**), ConA (**h**), Syto14 (**i**), and MitoTracker (**j**) at 0-48 hours after DOX induction. Hoechst: nucleus; Concanavalin A (ConA): endoplasmic reticulum; Syto14: nucleoli, cytoplasmic RNA; Phalloidin: F-actin; Wheat Germ Agglutinin (WGA): Golgi, plasma membrane; MitoTracker:

mitochondria. Phalloidin (Alexa Fluor 568) and WGA (Alexa Fluor 555) are combined in one channel as they are not separable on the microscope. GEMINI particles were highlighted in the bright-field images. Scale bars: 20 μm . n.s.: no significance; *: $p < 0.05$; ***: $p < 0.001$.

3. For in vivo validations, conditions for cytotoxicity assays are omitted, assay timing is not shown, “normal morphology” is described without data, and GEMINI particles are not co-stained.

R: We apologize for omitting several critical details of the experiments and data analyses, and we thank the Reviewer for pointing them out.

First, in our original manuscript, we mentioned that the assays were performed on day 14 post AAV injection, and we utilized immunohistochemistry to stain against NeuN (neuron marker) and GFAP (astrocyte marker) to compare the neuronal density and astrocyte activity. The GEMINI expressing group was compared with a negative control group injected with an equivolume saline.

We realized that our original description might be unclear, which resulted in confusion. Therefore, we revised the sentences describing the immunohistochemistry characterizations in the main text to improve clarity (**Page 22, Line 512**).

“By day 14, high-density GEMINI particles were consistently observed in brain regions receiving btAAV, such as the cortex and hippocampus (**Fig. 5a**). **At this timepoint, histological analyses of various subregions within the hippocampus revealed no significant differences from sham-injected controls in neuronal density or astrocyte reactivity (Fig. 5b,c; Extended Data Fig. 8e,f).**”

Second, the original “normal morphology” claim was merely supported by the immunohistochemistry images shown in **Extended Data Fig. 8**, where a quantitative analysis was absent. We recognized the importance of quantification and therefore, further performed imaging and segmentation for more detailed analysis of neuronal morphology. After 14 days of GEMINI expression, the soma and nuclear morphologies of the neurons were further quantified by measuring their sizes and sphericity and comparing results between the group with and without GEMINI expression (**Fig. R10a-c**). Our analysis revealed that the neurons in the GEMINI+ and - groups exhibited comparable soma sphericity, and nucleus volume. However, the GEMINI+ group exhibited slightly larger soma volume, which might be due to the occupation of the cytoplasmic space by some large GEMINI particles after extensive growth (**Fig. R10d**). We also observed modest decrease of nucleus sphericity, which might be attributed to the deformation of nuclei due to the large GEMINI particles (**Fig. R10g**). Overall, the impact of GEMINI on neuronal morphology is low, and we believe our original statement of “normal morphology” still holds. However, we would still like to tone down the claim by removing the statement of “normal morphology” and utilizing a more objective description of the data. To address this, **Fig. R10** was added to the revised manuscript as **Extended Data Fig. 9a-g**. Description and discussion of the results were included to the main text accordingly (**Page 22, Line 517**):

“Furthermore, we analyzed morphological features of GEMINI-expressing neurons, focusing on soma and nucleus (**Extended Data Fig. 9a-g**). While soma sphericity and nuclear volume remained unchanged, we observed a slight increase in soma volume and a modest decrease in nuclear sphericity (**Extended Data Fig. 9d-g**). These subtle changes may reflect the occupation of cytoplasmic space and nuclear deformation due to large GEMINI particles within some cells.”

Third, regarding the co-staining of GEMINI particles, it was actually included in our original manuscript. The second row of **new Extended Data Fig. 8e,f** (or the third row of **original Extended Data Fig. 7a,b**) showed the channel of GEMINI staining. For the GEMINI+ group, high-density GEMINI particles were visible in all the sub-regions in the hippocampus. In contrast, no GEMINI particles are present in the GEMINI- group as expected.

Fig. R10. **a-c**, Images (left) and their corresponding segmentations (right) of neurons (**a**), their nuclei (**b**), and GEMINI (**c**) within the mouse hippocampus CA1. The tissue was harvested 14 days after AAV injection. Scale bars: 20 μm. **d,e**, Comparison of the volume (**d**) and sphericity (**e**) of soma between GEMINI+ and GEMINI- neurons. **f,g**, Comparison of the volume (**f**) and sphericity (**g**) of soma between GEMINI+ and GEMINI- neurons. **d-g**, Box bounds: 25th and 75th percentile; whiskers: minimum and maximum; squares: mean; and center lines: median.

4. The behavioral assays are not sensitive to perturbation in those specific brain regions. A more sensitive assay would be unilateral motor cortex injection; it is known that asymmetric function of motor cortex may cause animals to turn in a specific direction. A good control for these experiments would be to inject an AAV expressing a transgene that silences or kills the transduced cells—would that yield quantifiable difference in the behavioral assay?

R: We thank the Reviewer for raising this point and for suggesting a better alternative behavioral assay. Following the Reviewer's suggestion, we designed and performed the experiment below:

We conducted an additional behavioral study comparing three groups: (1) mice unilaterally injected with AAV expressing GEMINI (GEMINI+ group), (2) a positive control group unilaterally injected with the 1:1 mixture of $ef1\alpha::FLEX$ -diphtheria toxin A (dtA) and $hsyn::Cre$ AAV at a same viral dose, and (3) a negative control group unilaterally injected with the same volume of saline (**Fig. R11a**). The positive control group serves as the functional impairment control suggested by the Reviewer, providing a direct comparison for evaluating whether GEMINI expression causes any similar behavioral deficits.

Behavioral tests were performed on day 14 post AAV injection. In the behavioral test, instead of using open-field test that evaluates gross motor function and exhibits low sensitivity, we decided to utilize the horizontal ladder rung walking test (**Fig. R11b,c**), a well-established method for detecting deficits of fine motor coordination that is more sensitive to subtle functional changes. Mice in the dtA group displayed an increased time passing the ladder and elevated count of missed steps per trial, confirming the assay's ability to detect motor dysfunction due to neuronal loss in M1 (**Fig. R11d,e**). In contrast, the GEMINI+ group performed comparably to the negative control group, with no statistically significant difference in either passing time or missed step. We further harvested the brain tissue from each group for histological analysis. The brain tissue from the dtA group exhibited slight shrinkage at the hemisphere with dtA expression, while no significant shrinkage was observed in the GEMINI and saline groups (**Fig. R11f**). Histological analysis revealed that the neuronal density of the GEMINI group is comparable to the saline group, coincided with the previous histological study (**Fig. 5b,c, and Extended Data Fig. 8**), while the dtA group showed a noticeable neuronal loss (**Fig. R11g,h**). These results indicate that GEMINI expression does not cause overt or measurable motor deficits within the investigated time window.

We greatly appreciate the Reviewer's input, which helped us further substantiate the biocompatibility of GEMINI in a functionally sensitive context. **Fig. R11** was added to the revised manuscript as **Extended Data Fig. 10**. Discussion of the results were added to the main text accordingly (**Page 23, Line 538**) as:

“To further evaluate fine motor coordination, we unilaterally injected btAAV into the M1 region. A positive control group expressing diphtheria toxin subunit A (dtA) and a negative control group receiving saline were also included (**Extended Data Fig. 10a**). Horizontal ladder rung walking tests were performed on day 14 (**Extended Data Fig. 10b,c**). GEMINI-expressing mice performed comparably to saline controls, while dtA-expressing mice exhibited impaired coordination and longer passing time (**Extended Data Fig. 10d,e**). Postmortem analysis revealed preserved brain symmetry in GEMINI mice, whereas dtA mice showed mild shrinkage in the injected hemisphere (**Extended Data Fig. 10f**). Histological analysis further confirmed minimal neuronal loss associated with GEMINI expression (**Extended Data Fig. 10g,h**).”

Fig. R11 a, Schematic of the experimental design. AAV encoding GEMINI was unilaterally injected into the primary motor cortex (M1) of mice (GEMINI group). A positive control group received AAVs expressing diphtheria toxin A subunit (dtA) at a comparable dose (dtA group), while a negative control group was injected with an equivolume of saline (saline group). **b,c**, Schematic (**b**) and snapshot (**c**) of the horizontal ladder (HL) rung walking test. The test was conducted 14 days after the AAV injection. Scale bar: 2 cm. **d,e**, Comparison of ladder crossing time (**d**) and

mean missed steps per trial (**e**) across the GEMINI, dtA and saline groups. **f**, Images of the brains post-fixation. GEMINI and saline groups have comparable size of the both hemispheres, while the dtA group exhibited visible shrinkage on the dtA-injected (*left*) hemisphere. Scale bars: 5 mm. **g**, Images of the M1 regions from GEMINI, dtA, and saline. Neurons (anti-NeuN, green), Nuclei (DAPI, cyan), GEMINI particles (JF₆₆₉, violet), and cells transduced by the dtAAAV (mCherry, red) were stained and imaged. The mice were sacrificed immediately after the HL tests. Scale bars: 50 μ m. **h**, Comparison of the neuronal density (NeuN+ cells) among the GEMINI, dtA, and saline groups (3 mice per group). **d,e,h**, bars: mean; whiskers: standard deviation.

B. The claim "recording at the single-cell level" is not well demonstrated or does not apply *in vivo*:

1. It should include identifiable cell segmentation and the registration of each GEMINI particle to a nucleus.

R: We appreciate the suggestion from the Reviewer. We do believe that it is important to showcase the capability to register each GEMINI particle to a designated cell as a proof of cellular level recording. During the revision, we made extensive effort to develop neuron and GEMINI particle segmentation and registration pipeline to show this capability.

Using a representative 3D brain tissue image (**Fig. R12a**), we showcased the segmentation and registration results using an in-house pipeline. In this pipeline, we first segmented neurons and GEMINI particles in the field of view (FOV) separately (**Fig. R12b**). The coordinates of the individual segments within the tissue were extracted and compared. The GEMINI-soma co-localization map was then obtained to show their spatial distribution (**Fig. R12c**). We then indexed all neurons within the FOV, the neurons with cytoplasmic GEMINI were then extracted and displayed individually (**Fig. R12d**). It is noteworthy that the segmentation and registration algorithm is yet to be optimized, as not 100% neurons and GEMINI particles are segmented and correlated. Despite the imperfection, this endeavor has demonstrated that the spatial information of the individual GEMINI and the neurons they belong to can be determined with confidence, showing GEMINI's capability to preserve the spatial information and pinpoint to single cells.

To clarify this point, we have added **Fig. R12** into the manuscript as **Extended Data Fig. 7e-h**. A brief description was added to the main text accordingly (**Page 22, Line 506**):

"Segmentation enabled accurate registration of each particle to its corresponding neuron, preserving their spatial information within the tissue (Extended Data Fig. 7e-h)."

Fig. R12 a, 3D image of brain tissue expressing GEMINI. **b**, Segmentation of neurons (*top*) and GEMINI particles (*bottom*) separately. **c**, Spatial colocalization of neurons and GEMINI in the tissue. **d**, Spatial labeling of individual neurons within the tissue (*left*) and the registration of each GEMINI particle to the soma of corresponding neurons (*right*). Scale bars: 50 μm .

2. To decode the “record,” the authors need to pool many heterogeneous single cells/particles together to narrow the distribution. Individual cells/particles do not yield effective decoding.

R: We thank the Reviewer for raising this point and appreciate the opportunity to clarify the key strength of the GEMINI platform. While the presentation of our data may have given the impression that decoding relies on pooling signals from multiple particles, this is not the case. Each GEMINI particle independently records the signaling activity of the individual cell in which it resides. The analysis of a single GEMINI particle reveals the signaling history of its corresponding cell, thereby enabling retrospective decoding with true cellular resolution. The quality of the band patterns recorded in single GEMINI particles (like those in **Fig. 2b,g,j**) is sufficient to provide a robust read.

We would like to highlight that GEMINI is a true single-cell technology. While some data presentations (**Fig. 2d, h, k**) include scatter plots summarizing a pool of particles, this reflects a common practice in single-cell technologies and data analysis rather than a limitation of the method itself. A useful parallel comparison is single-cell RNA sequencing (scRNA-seq), which captures transcriptional profiles at the level of individual cells. Despite this cellular resolution, the interpretation of scRNA-seq data rarely relies on single cells in isolation. Instead, researchers routinely analyze large populations of cells to identify transcriptional clusters, infer cellular heterogeneity, and uncover biological trends. Likewise, GEMINI records and decodes the signaling history of each cell independently, but pooling data from multiple GEMINI particles is essential for identifying spatiotemporal patterns, validating group differences, and interpreting biological relevance in the context of tissue heterogeneity. This strategy enhances statistical robustness while preserving the single-cell nature of the technology.

Upon revision, we recognized that our initial data presentation may have unintentionally contributed to confusion regarding this aspect of the GEMINI platform. To better illustrate the concept of single-cell decoding, we have performed an additional analysis that explicitly quantified the temporal decoding error of individual cells relative to the known ground truth. In this presentation, we “scrambled” the data in **Fig. 2d** and assigned them to a group based on the decoded timing. The confusion matrix comparing the decoded time to the ground truth showed highest frequency in the diagonal boxes, indicating high accuracy in decoding (**Fig. R13a**). We further calculated the time error of individual GEMINI particles, defined by the absolute value of difference between decoded time and ground truth (**Fig. R13b**). We found that, on average, **75.9%** of the single-cell decoding results fell within ± 1 hour of the ground truth, **98.2%** within ± 2 hours, and **100%** within ± 3 hours. These results demonstrate the high probability of decoding an event in a single cell with hour-level accuracy.

Fig. R13 a, Confusion matrix comparing the decoded time (y-axis) to the ground truth (x-axis) for GEMINI particles in **Fig. 2d**. The color scale indicates the number of particles decoded to each time, highlighting accurate versus misassigned time points. **b**, Time error distribution for individual GEMINI particles in **Fig. 2d** plotted against their ground truths (x-axis). Each dot represents a single particle, with color indicating the magnitude of time decoding error (color bar, right).

We further performed the same analysis for the GEMINI recording at the later stage of GEMINI growth (**Extended Data Fig. 4**). Similar results were obtained, while the temporal resolution is slightly lower ($92.9\% < \pm 6$ h error).

Fig. R14 a, Confusion matrix comparing the decoded time (y-axis) to the ground truth (x-axis) for GEMINI particles in **Extended Data Fig. 4d**. The color scale indicates the number of particles decoded to each time, highlighting accurate versus misassigned time points. **b**, Time error distribution for individual GEMINI particles plotted against their ground truths (x-axis).

As with all sensing modalities, some degree of error exists across cells. Not all errors found in such characterization arise from the tool, but could also come from the intrinsic

variation of cells and environments. In our system, decoding error arises from several sources: (1) intrinsic cellular heterogeneity that affects cellular response to stimuli, (2) variability in protein expression levels and thus the growth rate of GEMINI particles, and (3) error associated with the temporal decoding model. Despite these factors, our results consistently demonstrate that GEMINI enables decoding of signaling events at hour-level resolution for individual cells. This level of precision is sufficient for studying a broad range of cellular signaling events that occur over comparable timescales.

In the *in vivo* xenograft model showed earlier, we were able to demonstrate dose-dependent response and temporal decoding at the single particle level, despite the higher variation across particles. This could be attributed to (1) large cellular heterogeneity inherent to *in vivo* scenarios; (2) reduced accuracy of *in vivo* timestamping; (3) lower imaging quality of *in vivo* GEMINI particles; and (4) less accurate predictive model for signal decoding. Nevertheless, a temporal precision on the order of one day was still achievable. We do acknowledge that the cellular-level decoding of absolute chronological information is yet to be achieved in the central nervous system. This is due to several factors on the top of the ones listed above: (1) variability in GEMINI expression due to local AAV injection; (2) even higher cellular heterogeneity due to diverse neuronal subtypes. While the cellular heterogeneity reflects an intrinsic property of *in vivo* systems (and often time related to the scientific questions of interest), we believe that the latter challenges can be addressed through future optimization.

For example, lower variability in GEMINI expression in the brain can be achieved by systemic AAV delivery or the development of transgenic mice. Improved timestamping accuracy may be attainable through the development of labeling reagents with enhanced bioavailability or the implementation of alternative timestamping strategies, such as internal time references based on cellular processes like the cell cycle. Enhancements in imaging quality could be realized by employing tissue-clearing or expansion protocols. Furthermore, the accuracy of temporal decoding could be significantly improved using data-driven approaches, including deep learning-based models trained on *in vivo* datasets. These advancements, however, require years of sustained effort and multidisciplinary collaboration that extend beyond the scope of the present study. We believe our work already represents an outstanding innovation and anticipate that it will inspire further research in the field aimed at refining each of these components and ultimately advancing toward a better *in vivo* recording system.

We hope this clarification highlights the fundamental distinction between GEMINI and population-based techniques, and reinforces the platform's utility for cellular-level temporal mapping. To further clarify this point, we added **Fig. R13** and **Fig. R14** into the revised manuscript as **Fig. S8** and **Extended Data Fig. 4e,f**, respectively. Discussion was included in the main text accordingly:

“We further analyzed the time error between the decoded time and the ground truth, where **75.9%** of the single-particle decoding results fell within ± 1 hour of the ground truth, **98.2%** within ± 2 hours, and **100%** within ± 3 hours, indicating precise temporal decoding is possible at cellular level (**Fig. S8**).” (Page 13, Line 294)

and “Despite a much larger core and slower radial growth (**Extended Data Fig. 4b**), a timing standard deviation of 2-4 hours was still achievable (**Extended Data Fig. 4d**), and most cells (92.9%) exhibited a time error no more than 6 hours (**Extended Data Fig. 4f**), showing GEMENI’s capability to record at a broad window.” (Page 13, Line 301)

C. Other comments:

1. High temporal resolution is observed only in the early stages of cultured cells.

R: We thank the Reviewer for raising this concern. It is correct that the **highest temporal resolution is achieved during the early stages** of GEMINI growth. This is expected and consistent with the design of the platform, where the earliest timepoints correspond to the steepest phase of particle growth and therefore, offer a higher precision for temporal decoding.

However, we would like to emphasize that while the resolution is highest in the early stages (first day post-induction, **Fig. 2d**), **hour-level decoding remains robust throughout a longer time window**, as demonstrated in (4 days, **Extended Data Fig. 4**). In this experiment we applied the timestamps at 48 and 84 hours post-induction, while the signal was introduced at 54-78 hours and the cells were fixed at 96 hours. Still, a timing standard deviation of <4 hours (as opposed to ~1 hour in the first 24 hours) was maintained and 92.9% of GEMINI can report single-cell events with a time error no more than 6 hours (**Fig. R14**). This level of resolution is sufficient for resolving dynamic cellular events that occur over the course of days, while still offering meaningful information regarding the signaling dynamics of individual cells.

It is also noteworthy that the decoding accuracy reported here is based on the application of 2 timestamps, which is the minimum number needed for temporal decoding. Applying more timestamps will add additional reference points to boost the temporal resolution further.

To clarify this point, we added a brief discussion to the main text (**Page 13, Line 298**):

“As the radius of GEMINI particles grows nonlinearly, we expected a decreased temporal resolution at a later stage of recording. To quantify this, we initiated the recording at 48 hours post GEMINI expression and recorded for another 36 hours (**Extended Data Fig. 4a**). Despite a much larger core and slower radial growth (**Extended Data Fig. 4b**), a timing standard deviation of 2-4 hours was still achievable (**Extended Data Fig. 4d**), and most cells (92.9%) exhibited a time error no more than 6 hours (**Extended Data Fig. 4f**), showing GEMINI’s capability to record at a broad window.”

(**Page 28, Line 649**): “Even higher temporal resolution is achievable if more timestamps are employed as references for decoding.”

2. For *in vivo* applications, the authors have not demonstrated the ability to decode stimulus timing or dose as they do *in vitro*. In fact, we are quite confused with Fig 6d. It seems that even when DOX is applied at 0h (same as timestamp #1?), the peak occurs after timestamp#2, which was applied 1 day later. Is that due to delay in DOX-induced transcription?

R: These are good observations, and we are glad that the Reviewer raised these points. Here we would like to address the two concerns separately:

(1) Ability to decode stimulus timing or dose as we did *in vitro*.

In this revision, we explored *in vivo* xenograft model as described in our responses earlier. In this demonstration, we characterized dose-dependent responses and decoded the stimulus timing, similar to *in vitro* characterization. On one hand, differential responses to various levels of inflammation, induced by varying the dose of LPS administered, was demonstrated (**Fig. R7c**). On the other hand, the temporal information of the peak signal can be decoded reasonably well, achieving a temporal resolution on a one-day level (**Fig. R7h**). Though the accuracy is still not as high as the *in vitro* study due to factors such as heterogeneous cytokine diffusion within the tumor, lower precision of *in vivo* timestamping, and diverse cellular responses to inflammatory cues.

In the revision, we added discussion to account for the lower temporal resolution observed in the brain (**Page 29, Line 680**).

“It is noted that the temporal resolution of recording in the brain was lower than that observed in culture or xenografts, which could be attributed to factors such as the inherent heterogeneity of neurons, less accurate timestamping, variability in GEMINI expression, and less accurate predictive model for decoding. Disentangling these factors remains an ongoing challenge; nevertheless, GEMINI still holds great potential for further optimization in native tissues.”

The results from the *in vivo* xenograft model were included as **Fig. 4**. We have also added a new section entitled “**GEMINI records inflammatory responses *in vivo***” to present the new data. (**Page 18, Line 420**).

Indeed, we are not yet able to decode the absolute chronological information from GEMINI particles grown in the mouse brain via local AAV delivery, as we found that the variation is relatively large and the decoding could not provide a robust temporal readout as we saw *in vitro*. There are several factors that could contribute to the lower temporal resolution in the brain. For instance, we found the size distribution of GEMINI particles *in vivo* is much broader (**Extended Data Fig. 7c**), which is largely caused by the local AAV injection that creates a large variation in the copy number of the GEMINI DNA obtained by individual cells, thus resulting in asynchronous nucleation and varied growth rate. During recording, there are a considerable number of GEMINI particles that passed the

early stage of assembly, giving rise to a large error. In addition, the systemic administration of drugs might not reach different neurons at the same time, and even they reach at the same time, the response of individual neurons to the same drug might be different. Furthermore, we found it difficult at this stage to quantify the growth profile of GEMINI particles *in vivo* as we did *in vitro* through timelapse imaging. This is not only a technical challenge but also a consideration of animal welfare: when tracking GEMINI growth in the cortical area of live animals, we can only image for no more than 1-2 hours to avoid animal stress. Such a short period of tracking cannot afford meaningful profile of the GEMINI particle growth, especially considering that they grew slower *in vivo*. Therefore, the actual growth profile of GEMINI in neurons *in vivo* might be slightly different from that in HEK293T cells *in vitro* and requires a different decoding model. All these factors, among others, contribute to the failure of decoding reliable and verifiable chronological information in the mouse brain at this moment.

Overall, the below factors could all contribute to the inaccuracy: (1) the intrinsic heterogeneity of neurons within the brain tissue; (2) varied pharmacokinetics of systemically administered DOX and PTZ to different subregions of the brain, (3) the heterogeneity of GEMINI expression via local AAV injection; (4) the considerable animal-to-animal variation; (5) the lower accuracy of *in vivo* timestamping; (6) lower imaging quality of *in vivo* GEMINI particles; and (7) less accurate predictive model for signal decoding.

Despite the challenges, we believe there are approaches that could circumvent or partially address the above issues in the near future. For instance, instead of local AAV delivery, systemic administration could be employed to afford a more homogenous expression level across neurons. Alternatively, transgenic mice could be developed. Improved timestamping accuracy may be attainable through the development of labeling reagents with enhanced bioavailability or the implementation of alternative timestamping strategies, such as internal time references based on cellular processes like the cell cycle. Enhancements in imaging quality could be realized by employing tissue-clearing or expansion protocols. Furthermore, the accuracy of temporal decoding could be significantly improved using data-driven approaches, including deep learning-based models trained on *in vivo* datasets.

(2) Delay in the peak of the signals compared to the timestamps.

Yes, the delay in DOX-induced transcription is an important factor for the delay in GFP signals. Based on our experience, 1-2 hours are generally needed for transcription and translation *in vitro*, before any signals can be detected in GEMINI. Such delay will contribute to a later onset of the GFP signals with respect to the timestamps. After the onset, there is also a signal ramp-up phase that takes from hours to tens of hours to reach the peak, depending on factors such as the duration of the signals and the stability of the

mRNA. The *in vivo* administration of DOX was reported to exhibit a relatively slow removal from the central nervous system, with a $t_{1/2}$ ~4 hours (*J. Pharmacol. Exp. Ther.* **368.1** 32-40 (2019)). Therefore, after systemic administration of DOX, the signal is ON for an extended period of time, which resulted in the delay in reaching the peak of GFP signals.

To clarify this point, we add a discussion about the delay of peak position in the main text (**Page 26, Line 596**):

“Notably, the signal peak exhibited a considerable delay, as DOX persisted in the tissue and kept the signal ON for an extended period³⁹.”

3. The GitHub link does not work.

R: Our original plan was to make the GitHub link public upon the official publication of the manuscript. Similarly, all relevant plasmids have been deposited to Addgene and passed quality control, which will be released to the public immediately upon publication.

In response to the comment, we have now made the GitHub link publicly accessible (<https://github.com/DCLinLab/GEMINI>). After getting permission from the University Technology Transfer Office, we have also just posted the revised manuscript on **bioRxiv** (<https://doi.org/10.1101/2025.07.16.664392>).

Referee #3 (Remarks to the Author):

The manuscript by Yan and co-workers describes a new cellular recording system that utilizes growing protein crystals to store information about order of events in-cellulo and in-vivo. The method generates large protein crystals intracellularly and pulses of fluorescent materials can be generated to store information of timing events that generates optically readable patterns like growth rings in trees. The work is innovative and novel. It appears that it however lacks a demonstration that clearly leverages the advantages claimed in the introduction. The authors are correct when they write that many systems that record cellular events have been demonstrated before by recording edits to DNA that can be read by sequencing. The authors are also correct that these other methods are i) typically read out over a population of cells, and ii) since they are not optical, would require disruption of the tissue as well as iii) would not be able to answer the absolute chronological order of events (only order of events). For all of these it is not clear to this reviewer how they are leveraged in the in-vivo demonstration provided, i.e. couldn't for example the transcription and seizure events in-vivo be measured with genetic recording? Especially since in all the experiments where the authors are resolving time-stamps for events, they appear to use averaged traces making this more of a bulk method (providing info of effects for a group of cells). In my opinion, the work is better suited for Nature Biotechnology or Nature Methods.

R: We thank the Reviewer for the positive and constructive comments. We appreciate their recognition of the novelty and innovation of the GEMINI system, particularly in its use of intracellularly grown protein assemblies to encode temporally ordered events in living cells and tissues. We apologize for any ambiguity in our original manuscript and are grateful for the opportunity to clarify how the advantages we highlighted in the introduction, including single-cell resolution, optical readout in intact tissues, and absolute temporal decoding, are indeed realized and leveraged in the demonstrations.

Regarding these advantages of GEMINI, we would like to clarify/highlight them point-by-point as below:

1. **GEMINI is not a bulk or population-level method.**

We would like to highlight that GEMINI is a true single-cell technology and each data point corresponds to a decoding result from a single GEMINI particle within a single cell. While some data presentations (**Fig. 2d, h, k**) include scatter plots summarizing a pool of particles, this reflects a common practice in single-cell technologies and data analysis rather than a limitation of the method itself. A useful parallel comparison is single-cell RNA sequencing (scRNA-seq), which captures transcriptional profiles at the level of individual cells. Despite this cellular resolution, the interpretation of scRNA-seq data

rarely relies on single cells in isolation. Instead, researchers routinely analyze large populations of cells to identify transcriptional clusters, infer cellular heterogeneity, and uncover biological trends. Likewise, GEMINI records and decodes the signaling history of each cell independently, but pooling data from multiple GEMINI particles is essential for identifying spatiotemporal patterns, validating group differences, and interpreting biological relevance in the context of tissue heterogeneity. This strategy enhances statistical robustness while preserving the single-cell nature of the technology.

2. Absolute chronological timing is preserved and decoded optically.

Unlike DNA-editing-based recorders that provide only relative order (e.g., “event A occurred before B”), GEMINI stores real-time chronological information in the spatial structure of the growing protein lattice. By introducing timestamps at defined time points and decoding the relative position of the signal band, we can reconstruct the absolute timing of signaling events within individual cells, and with a temporal resolution on the order of hours *in vitro* (~2 hours for a 24-hour recording and ~6 hours for a 36-hour recording) and around one day *in vivo* (see the new results from *in vivo* xenograft model). This represents a unique capability that, to our knowledge, has not been achieved by any other modalities in intact tissues. More details can be found in our responses to Critique #5.

3. *In vivo* demonstrations highlight spatiotemporal decoding in intact tissues.

We acknowledge that the temporal resolution of GEMINI recording in the mouse brain was indeed much lower than our *in vitro* demonstration, which is mainly due to (1) the intrinsic heterogeneity of neurons within the brain tissue; (2) the large variation of GEMINI expression level via local AAV injection; (3) the considerable animal-to-animal variation; (4) the lower accuracy of *in vivo* timestamping; (5) lower imaging quality of *in vivo* GEMINI particles; and (6) less accurate predictive models for signal decoding. We believe the issues can be resolved or mitigated through engineering and system optimization in the near future, approaching a much higher temporal resolution close to or comparable with the *in vitro* scenario.

Despite the challenges, a much higher temporal resolution is achievable in other *in vivo* models at the current stage. To demonstrate this, we employed an alternative *in vivo* model based on HEK293T xenografts expressing NFκB-GEMINI. Specifically, we subcutaneously implanted GEMINI-expressing cells into immunodeficient mice and triggered acute inflammation using intraperitoneal (i.p.) injection of lipopolysaccharide (LPS), a well-established model for inflammation relevant to disorders such as sepsis (*Biomedicine & Pharmacotherapy* **141**, 111890 (2021)) and neuroinflammation (*Glia* **55(5)**, 453-462 (2007)). This model is less strong and artificial than the PTZ-induced seizure.

Fig. R15. **a**, Experimental procedure for *in vivo* recording of LPS-induced inflammation. **b**, A section of a HEK293T xenograft showing GEMINI recording of NFkB activation following systemic injection of 3 mg kg⁻¹ LPS. Vasculature was labeled by anti-CD31 immunostaining (red). Insets: magnified 3D rendering of the central region (left) and a heatmap displaying the normalized GFP intensity ($F_{\text{GFP}}/F_{\text{HTL}}$) of individual particles (right). Scale bars: 1 mm, 100 μm (inset). **c**, GEMINI particles recording NFkB signaling at various LPS doses (left) and the dose-dependent signal intensity (right). Scale bar: 100 μm . **d**, Experimental procedure for *in vivo* timestamping via systemic injection of HTL dyes. **e**, Fluorescence images of GEMINI particles in xenografts labeled with two timestamps; inset: magnified image of a GEMINI particle. Scale bars: 20 μm , 2 μm (inset). **f**, Experimental procedure for recording NFkB signal with LPS administered at various times. **g**, Representative images (left) and fluorescence profiles (right) of GEMINI particles recording NFkB activation in mice injected with LPS on days 0, 1, or 2. Scale bar: 2 μm . **h**, Decoded timing of

NFκB activation peaks in groups receiving LPS on days 0–2 ($n = 52/59/55$ for $t = 0/1/2$ days; 3 mice per group). **i,j**, 3D images of xenograft tissue with high (**i**) and low (**j**) vasculature density after LPS administration on day 0. Scale bars: 100 μm. **k**, Mean fluorescence profiles of NFκB signals recorded in GEMINI particles located in regions with high (solid) and low (dashed) vasculature density. **l**, Decoded peak timing of NFκB activation for GEMINI particles from high- and low-vascularized regions ($n = 42/47$ for high/low; 3 mice). **c,h,i**, Box bounds: the 25th and 75th percentiles; whiskers: minimum and maximum; squares: mean; and center lines: median.

Following a 10-day engraftment period, we initiated GEMINI expression with doxycycline (DOX) and examined NFκB signaling in response to 3 mg kg⁻¹ LPS. Tumor-wide expression of GEMINI and cellular response to inflammation was observed (**Fig. R15a**). We further investigated the cellular responses to various doses of LPS, from 0.03 to 3 mg kg⁻¹. As shown in **Fig. R15b-c**, GEMINI recorded robust, dose-dependent NFκB activation, with sensitivity sufficient to detect inflammation induced by LPS doses as low as 1% of that commonly used in the literature for inflammation modeling (*Int. J. Biol. Macromol.* **279**, 135371 (2024)). These results highlight GEMINI's ability to resolve mild and graded physiological signals across a large tissue.

We then investigated if GEMINI could decode the timing of transient inflammatory events *in vivo*. Through retro-orbital injection of Janelia Fluor dyes, we applied two timestamps 48 hours apart, where clear timestamp bands were observed in most GEMINI particles (**Fig. R15d-e**). A dose of LPS administered at different times between the timestamps could be temporally decoded using the established model, and decoding results revealed consistent alignment between the timing of LPS administration and the peak NFκB activation (**Fig. R15g-h**). The temporal resolution was on the order of one day, greater than *in vitro*, which is likely due to heterogeneous cytokine diffusion, timestamping variability, and differential cellular responses.

Given the known heterogeneity in vascularization within HEK293T-derived xenografts, we hypothesized that vascular density might influence cellular exposure to cytokine and thus affect signal timing. To test this, we labeled vasculature in xenograft using anti-CD31 staining and stratified the xenograft regions based on the vascular density. We performed temporal decoding separately for GEMINI particles in high (>8% volume occupation) and low (<2% volume occupation) vascularized regions. As expected, poorly perfused areas exhibited significantly delayed NFκB activation (**Fig. R15i-l**). These results showcase GEMINI's power to capture spatial heterogeneity in signaling dynamics at the cellular level across intact tissue.

To strengthen the manuscript, we incorporated this new study into the revision, with **Fig. R15** now included as **Fig. 4**. A new section titled “**GEMINI records inflammatory responses *in vivo***” has been added to the main text. A figure illustrating tissue slices from a representative xenograft with heterogeneous NFκB activation has been included as

Extended Data Fig. 5. In addition, the abstract and introduction were revised to reflect the new results:

Abstract: “In a xenograft model, GEMINI recorded inflammation-induced signaling dynamics across tissue with cellular resolution, revealing spatial heterogeneity linked to vascular density.” (Page 2, Line 30)

Introduction: “Using an *in vivo* xenograft model, we demonstrated GEMINI’s ability to record both the amplitude and temporal information of inflammation-induced signaling and to correlate cellular responses with local vascular density.” (Page 5, Line 101)

In addition, we believe our new results from *in vivo* xenograft can also well address the comment regarding the lack of a demonstration that clearly leverages the advantages claimed in the introduction.

We thank the Reviewer again for the valuable feedback. Please find our responses to the specific critiques in a point-by-point manner below.

Specific remarks:

1. Growth rate. Fig 2d shows the “volume index” (a measure of volume based on 2D imaging). This is clearly not linear at the later phases of the growth (>35h). Why not focus on the linear part of the growth range? In addition, the spikes in the individual growth profiles seem to indicate that individual particles are either non-stable or not growing linearly. This is an issue with the claimed advantage i) above. Population averages appears quite important for this method.

R: We are glad that the Reviewer raise these points and we apologize for did not present it clearly in our original manuscript. To address them clearly, we would like to respond individually to the three aspects of this comment: non-linearity at later stages, variability in individual growth curves, and the necessity of population averages in this method:

1. Non-linearity in later growth phases: We agree that the “volume index,” derived from 2D projections, begins to deviate slightly from linearity after ~35 hours. This non-linearity can be attributed to two main factors:

First, GEMINI particles are freely suspended in the cytoplasm and can undergo subtle z-axis movement. Our analysis assumes particles remain in the midplane (the plane of largest cross-sectional area); however, as they grow, they are less likely to remain precisely in that plane, causing 2D projections to slightly underestimate particle size over time.

Second, the GEMINI particles might indeed grow slightly slower at later stages. Rapid lattice assembly may accumulate structural defects that increase the energy barrier for continued growth, resulting in slight deceleration.

We have now clarified this in the main text (**Page 9, Line 206**):

“Minor deviations from linearity were observed near the end of the tracking period, likely due to reduced accuracy in capturing the true midplane as particles grew larger, as well as a possible increase in the energy barrier for assembly resulting from defect accumulation.”

Importantly, our decoding strategy does not rely exclusively on perfect linear growth of GEMINI particles. While linearity contributes to precision, GEMINI also uses timestamping as internal temporal references, allowing retrospective calibration of individual growth profiles. In this study, we used just two timestamps to simplify the experiment, yet achieved high accuracy. The system supports additional timestamps, which can further improve resolution, analogous to using a ruler with finer graduations for more precise measurement. In practical applications, end users can tailor the number of timestamps according to the temporal resolution required for their specific experimental design. As shown in **Fig. 1j**, up to 11 dye-switching events can be readily introduced, each can serve as a timestamp, offering an example of multi-timestamp implementation.

To highlight this point, we have added this to the Discussion accordingly (**Page 28, Line 649**):

“Even higher temporal resolution is achievable if more timestamps are employed as references for decoding.”

2. Spikes in individual growth profiles: The “spikes” observed in single-particle growth traces are primarily due to two technical factors: (1) tracking errors and (2) z-axis motion within the cytoplasm.

First, long-term single-particle tracking in live cells is inherently challenging, especially across hundreds of frames and in crowded fields. Minor segmentation fluctuations can occur as particles move and the cellular environment changes. As shown in Movie S2, our algorithm is robust, but small frame-to-frame variations persist. Second, since volume is estimated from 2D cross-sections, any vertical displacement of the particle due to Brownian motion causes deviation from the midplane and thus variation in apparent size. These fluctuations do not reflect actual instability or disassembly of GEMINI particles.

Unlike dynamic protein assemblies like the cytoskeleton, GEMINI particles are designed to be exceptionally stable, as confirmed in **Fig. 1f,g**. With prolonged incubation, the encoded bands exhibited negligible change in sharpness, indicating their intact structure and negligible subunit exchange. This characteristic is critical for preserving timestamp fidelity. This stability results from the design: ultra-stable protein cages assemble into a diamond-like lattice by binding to four neighbors, locking each unit in place.

To clarify this, we have added a brief discussion to the main text accordingly (**Page 9, Line 209**):

“Fluctuations in individual growth profiles were attributed to tracking inaccuracies and particles’ out-of-plane motion during imaging, rather than intrinsic structural instability.”

3. The necessity of population averages:

As clarified in our earlier response, **GEMINI is inherently a single-cell recorder**. While figures may appear to show population-level results, each data point corresponds to an individually decoded GEMINI particle. For instance, in **Fig. 2d**, each dot represents the decoded timing from a single cell. The same applies to **Figs. 3c** and **3f**, where ON and OFF transitions of NFκB signaling are recorded independently per cell.

While decoding variability exists, as is common with any biological sensing platform, it arises primarily from factors including: (1) intrinsic cellular heterogeneity that affects signal response kinetics, (2) variability in protein expression levels and thus the growth dynamics of GEMINI particles, and (3) error associated with timestamping and the temporal decoding model. Nevertheless, our platform consistently achieves high temporal precision, especially in the *in vitro* scenarios, to resolve dynamic events on the scale of hours for individual cells. More details regarding our single-particle decoding accuracy can be found in our response to **Critique #5**.

Finally, while GEMINI records at the single-cell level, pooled analyses remain valuable to capture statistically meaningful trends across heterogeneous cell populations. Thus, the pooled data analysis supports rather than contradicts the single-cell nature of GEMINI.

2. Also, in fig 2b it appears that the outer layer is affected more by longer post-treatment incubation. Doesn't this mean that the outer layer is not that "locked"? (I.e. contradicting "minimal exchange between BBs in GEMINI particles and those in the cytoplasm"?)

R: We appreciate the Reviewer's careful observation. In our original presentation, we showed only the fluorescence profiles of GEMINI particles over time (**Fig. R16a, right**) and a comparison of band positions (**Fig. R16b**), without accompanying images of the actual particles at different time points (**Fig. R16a, left**). We recognize that this may have made it unintuitive to interpret the evolution of particle stability.

The apparent rightward shift of the peak in the outer fluorescence layer does not indicate instability or subunit exchange but rather reflects the continued growth of GEMINI during the incubation. As the particles grow, the outer (blue) layer becomes thicker over time (**Fig. R16a, left**). In our analysis, the fluorescence profiles were normalized by setting the onset of the first band (red) and the peak of the second band (yellow) to positions 0 and 1, respectively. Because particle growth continues after these timestamps, the third band (blue) appears increasingly displaced outward in normalized space, even though the recorded signal remains stable.

Fig. R16 a, Representative fluorescent images (*left*) and mean fluorescence profiles (*right*) of GEMINI particles incubated in live cells for various durations before fixation, following identical dye-switch protocols ($n = 54/50/52/62$ for 6/12/25/50 h). Scale bars: 2 μm . **b**, Band positions of GEMINI particles in **a**.

To help clarify this, we have now included representative particle images from each time point to the revised main figure as **Fig. 1f**. We believe this visual context will reduce confusion and further support the conclusion that GEMINI particles exhibit minimal subunit exchange and maintain structural integrity during extended growth.

3. The authors write “The results demonstrate the high accuracy of the model in describing individual GEMINI growth, which also corroborates the assumption of steady protein expression, an important indicator of normal cell metabolism.” I think there are several problems with this. First of all it seems to assume that all synthesized BBs get either degraded or incorporated, why wouldn't you have a stable non-incorporated population? Secondly, based on the curves (see comments about Fig. 2 above), the growth does not seem to be continuous for individual particles. Also these assumptions are only true to cell culture conditions, I don't think they can be assumed in physiological conditions.

R: We thank the Reviewer for raising the concerns. Below, we address the three concerns raised separately:

1. Stable non-incorporated population of building blocks. A stable, non-incorporated pool of building blocks indeed exist and was considered in our model. In the main text, we stated:

“In the model, we hypothesized that the BB’s synthesis rate, cytoplasmic concentration, and degradation rate were all constant in the steady growth phase, affording a constant rate of BB addition onto GEMINI particles.”

This reflects the assumption of a constant cytoplasmic concentration of BBs after nucleation. In the Supplementary Notes (*Mathematical model of intracellular GEMINI growth*), this was listed as Assumption (3):

“The cytoplasmic concentration of the building blocks remains constant (saturation condition) after GEMINI nucleation.”

This equilibrium concentration represents the stable, non-incorporated pool of building blocks. This concentration is constant for fixed temperature and pressure. Since it remains constant, protein synthesis and incorporation into GEMINI can be directly linked, supporting the validity of the growth model.

It is worth noting that the equilibrium concentration of building blocks directly affects temporal resolution. Although not emphasized in the original manuscript, this principle was carefully considered in the design of the GEMINI system. We previously explored the optimal fusion position for HaloTag and fluorescent proteins to minimize cytoplasmic background, identifying the N-terminus of the A chain as the most effective (**Fig. R17a,b**). A higher cytoplasmic concentration slows the depletion of signal building blocks after signal termination, leading to blurred band transitions and reduced ability to resolve fast events (**Fig. R17c,d**). Similarly, elevated equilibrium concentrations can blur timestamp

boundaries and lower decoding accuracy. The GEMINI variant presented here maintains a low equilibrium concentration, resulting in sharp signal transitions and the ability to distinguish events separated by short time intervals (**Fig. 3d,e**).

Fig. R17 a, Images showing the growth of GEMINI in HEK293T cells with HT fused to various termini of A and B chains. **b**, Comparison of the GEMINI-to-cytoplasm fluorescent intensity ratio among the different fusions. Whiskers: standard deviation. **c,d**, Plots showing how the cytoplasmic concentration determines the band sharpness. A lower solubility results in a shorter transition time after switching to the second band (**c**), therefore affording a sharper transition boundary (**d**).

To highlight the importance of the low cytoplasmic concentration, we include further discussion in the main text accordingly (**Page 8, Line 165**):

“The timestamp subunit was constructed by fusing HaloTag (HT) to the N terminus of the A chain²⁹, which exhibited the lowest impact on their co-assembly, **as evidenced by its low equilibrium concentration in the cytoplasm (Fig. S3a,b)**. The same terminus was later used to construct the reporter subunit. This low equilibrium concentration of subunits is critical for achieving sharp transitions of timestamps and distinguishing closely spaced events (**Fig. S3c,d**).”

2. Discontinuous growth of individual particles. We acknowledge that fluctuations in individual growth curves may give the impression of discontinuous growth. However, as discussed in our response to Critique #1, these fluctuations are primarily due to tracking imperfections and vertical (z-axis) movement of particles during imaging, rather than actual pauses in growth. Time-lapse imaging (e.g., **Movie S2**) consistently shows GEMINI particle growth without regression, supporting the validity of a continuous growth model over longer timescales.

3. Applicability of the assumptions to physiological conditions. We recognize that the growth model developed and validated in cultured cells may not be directly applied to *in vivo* conditions, though many of its assumptions remain plausible. We attempted to track GEMINI particle growth *in vivo* but encountered significant technical challenges. Intravital imaging, necessary for live tracking, does not provide the same resolution as *in vitro* imaging on glass-bottom culture dishes, making accurate segmentation and tracking difficult. Moreover, particle growth *in vivo* is slower, while animal welfare considerations limit imaging duration to 1–2 hours, which is insufficient to capture meaningful growth profiles. As a result, building an *in vivo*-specific growth model based on accurate measurement of *in vivo* growth profile is currently impractical.

In our *in vivo* recording in the mouse brain, we did not attempt to decode the absolute temporal information of the signals using our model derived for culture. We noticed that the variation of the signal from particles from different cells is much larger than the *in vitro* tests, which could be attributed to the following factors: (1) the intrinsic heterogeneity of neurons within the brain tissue; (2) the large variation of GEMINI expression level via local AAV injection; (3) the considerable animal-to-animal variation; (4) the lower accuracy of *in vivo* timestamping; (5) lower imaging quality of *in vivo* GEMINI particles; and (6) less accurate model for signal decoding.

In contrast, we achieved higher temporal resolution than the mouse brain model using our *in vivo* HEK293T xenograft model. Here, the *in vitro*-derived growth model enabled reliable decoding of signal timing and dose, producing robust and reproducible results (**Fig. R15**). This compatibility is likely due to the use of the same clonal NFκB-GEMINI HEK293T cell line in both culture and xenograft contexts. While we were unable to validate *in vivo* growth profiles directly due to the imaging constraints described above, the consistency of decoding results supports the potential applicability of the model to xenograft studies.

To clarify the applicability of the model for the decoding of *in vivo* data, we included a discussion in the main text accordingly:

“Temporal decoding of individual particles using the model validated in culture showed that the NFκB signal peaked *ca.* one day after LPS injection in each group (**Fig. 4h**).”
(Page 21, Line 479)

“Third, decoding models must be modified for *in vivo* conditions, accounting for potential differences in the growth behavior within living tissues. However, real-time tracking remains challenging due to current limitations of long-term intravital imaging. As a result, alternative strategies, such as multi-timestamping or statistical analysis at discrete time points, may be employed as a proxy.” (Page 29, Line 693)

4. Line 208 “We also analyzed the influence of GEMINI on cell morphology and division. Nucleation did not alter cell shapes or interrupt mitosis and cytokinesis. After division, GEMINI particles randomly entered one of the daughter cells (Extended Data Fig. 3e,f).” I don’t think this is enough data to make this kind of broad statement, soften the wording.

R: We thank the Reviewer for this constructive suggestion. We indeed recognized that the original wording may have overstated the breadth of our observations based on the available data.

To address this, we further investigated the effects of GEMINI expression on subcellular morphology using the well-established **Cell Painting** assay (Bray et al., *Nat. Protoc.* 2016). GEMINI-expressing cells were co-stained with a multiplexed panel of fluorescent markers targeting key organelles: nucleus (Hoechst), endoplasmic reticulum (ConA), nucleoli and cytoplasmic RNA (Syto14), cytoskeleton (F-actin, Phalloidin), Golgi and plasma membrane (WGA), and mitochondria (MitoTracker). Imaging was performed at multiple time points from 12 to 48 hours post-induction to assess structural integrity during the recording period (**Fig. R18a**).

No major changes in the morphology of cellular and subcellular structures were observed by visual inspection. To quantify any subtle effects, we segmented and measured the area of key markers (Hoechst, ConA, Syto14, and MitoTracker) across timepoints (**Fig. R18b–e**). While no statistically significant changes were detected in most cases, we observed a slight decrease in nuclear area and a subtle increase in mitochondrial area at 48 hours. We hypothesize these changes may reflect mild nuclear deformation from GEMINI overgrowth or increased cell crowding due to confluency. We further performed the Cell Painting assay in U2OS cells, which showed no significant morphological changes (**Fig. R18f**). The study supports the conclusion that GEMINI does not induce substantial subcellular stress.

We also toned down the language to more accurately reflect the experimental observation and our analysis. In the revised manuscript, we discussed cell morphology and division separately:

Morphology (**Page 10, Line 236**): “Next, we profiled the morphological features of subcellular structures during GEMINI growth via the Cell Painting assay³². While most compartments exhibited minimal structural changes, a decrease in the nuclear area and a subtle increase in mitochondrial area were observed at 48 hours after GEMINI expression (**Fig. S7a–e**), which could be due to potential physical interactions and changes in confluency. However, these changes were not observed in U2OS cells expressing GEMINI (**Fig. S7f–j**), suggesting that distinct cell types may respond differently to GEMINI.”

Division (Page 12, Line 269) : “We then further investigated the influence of GEMINI on division at the cellular level, observing no apparent disruption of mitosis and cytokinesis following GEMINI nucleation. After division, GEMINI particles entered one of the daughter cells (Extended Data Fig. 3c). In contrast, cytoplasmic nucleation of linear recorders like iPAK4 was found to disrupt proliferation by preventing cytokinesis (Extended Data Fig. 3e,f).”

Fig. R18 a, Images of the Cell Painting assay on HEK293T cells with (top) and without (bottom) the induction of cytoplasmic GEMINI growth by DOX. **b-e**, Statistical comparison of the

segmented area of Hoechst (**b**), ConA (**c**), Syto14 (**d**), and MitoTracker (**e**) at 0-48 hours after DOX induction. **f**, Images of the Cell Painting assay on U2OS cells with (top) and without (bottom) the induction of cytoplasmic GEMINI growth by DOX. Bright field images were shown to highlight the GEMINI particles. Scale bars: 20 μm . **g-j**, Statistical comparison of the segmented area of Hoechst (**g**), ConA (**h**), Syto14 (**i**), and MitoTracker (**j**) at 0-48 hours after DOX induction. Hoechst: nucleus; Concanavalin A (ConA): endoplasmic reticulum; Syto14: nucleoli, cytoplasmic RNA; Phalloidin: F-actin; Wheat Germ Agglutinin (WGA): Golgi, plasma membrane; MitoTracker: mitochondria. Phalloidin (Alexa Fluor 568) and WGA (Alexa Fluor 555) are combined in one channel as they are not separable on the microscope. GEMINI particles were highlighted in the bright-field images. Scale bars: 20 μm . n.s.: no significance; *: $p < 0.05$; ***: $p < 0.001$.

5. Line 230 “The results showed close agreement with the ground truths, where the means of the decoded time exhibited errors of less than 30 minutes.” The mean did, on an individual level it would be hard to interpret. The authors should scramble the data and try to predict the timestamp to get a real accuracy value (e.g. take segmented images of individual particles and try to predict the time).

R: We thank the Reviewer for this great suggestion! Indeed, while the mean decoded time may closely match the ground truth, a broader distribution could obscure variability at the single-particle level. A more rigorous way to assess accuracy is to evaluate decoding on a per-particle basis and quantify the frequency of correct assignments.

Fig. R19 a, Confusion matrix comparing the decoded time (y-axis) to the ground truth (x-axis) for GEMINI particles in **Fig. 2d**. The color scale indicates the number of particles decoded to each time, highlighting accurate versus misassigned time points. **b**, Time error distribution for individual GEMINI particles in **Fig. 2d** plotted against their ground truths (x-axis). Each dot represents a single particle, with color indicating the magnitude of time decoding error (color bar, right).

As suggested by the Reviewer, we scrambled the data and binned the decoded time of each particle into discrete time groups. For example, particles decoded between 1.5–2.49 h were assigned to the 2 h group; 2.5–3.49 h to the 3 h group, and so on. We then constructed a confusion matrix comparing decoded versus actual time (**Fig. R19a**). As shown, the highest frequencies lie along the diagonal, indicating strong agreement between predictions and ground truth.

Next, we quantified the absolute time error for each GEMINI particle (**Fig. R19b**). This analysis revealed that 75.9% of decoded timepoints fell within ± 1 hour of the ground truth, 98.2% within ± 2 hours, and 100% within ± 3 hours, demonstrating that GEMINI enables hour-level temporal resolution at the single-cell level.

We further performed the same analysis for GEMINI particles in a later growth phase (**Extended Data Fig. 4d**). As expected, the resolution was slightly reduced, with 92.9% of particles showing errors under ± 6 hours (**Fig. R20**), consistent with our earlier observation of reduced decoding precision at later timepoints.

Fig. R20 a, Confusion matrix comparing the decoded time (y-axis) to the ground truth (x-axis) for GEMINI particles in **Extended Data Fig. 4d**. The color scale indicates the number of particles decoded to each time, highlighting accurate versus misassigned time points. **b**, Time error distribution for individual GEMINI particles plotted against their ground truths (x-axis).

These results validate the high single-particle decoding accuracy of GEMINI and reinforce our claim that this system provides true single-cell resolution. Accordingly, we have added these datasets to the revised manuscript:

“We further analyzed the time error between the decoded time and the ground truth, where **75.9%** of the single-cell decoding results fell within ± 1 hour of the ground truth, **98.2%** within ± 2 hours, and **100%** within ± 3 hours, indicating precise temporal decoding is possible from individual GEMINI particles (**Fig. S8**).” (**Page 13, Line 294**)

and “Despite a much larger core and slower radial growth (**Extended Data Fig. 4b**), a timing standard deviation of 2-4 hours was still achievable (**Extended Data Fig. 4d**), and most cells (**92.9%**) exhibited a time error no more than 6 hours (**Extended Data Fig. 4f**), showing GEMINI’s capability to record at a broad window.” (**Page 13, Line 301**)

6. Line 239: “a temporal resolution of 2-4 hours was still achievable, demonstrating GEMENI’s broad recording window.”

One cannot equate temporal resolution to the spread (std) of a population measurement. If one wants to use the std then the safely resolvable "distance" of events would be around 2x std.

R: We thank the Reviewer for this clarification regarding the interpretation of temporal resolution. We agree that the standard deviation reflects variability in decoded timepoints and should not be equated directly with resolution. As the Reviewer rightly notes, temporal resolution should be defined by the minimum reliably distinguishable interval between two events.

In our original manuscript, the estimate of “2-4 hours” was derived from the standard deviation of decoded timepoints. To avoid misinterpretation, we have revised the text to report this value explicitly as the standard deviation, rather than referring to it as “temporal resolution”.

Furthermore, to provide a more meaningful measure of decoding accuracy, we now report the ratio of single-particle decodings that fall within specific error margins (e.g., ± 1 hour, ± 2 hours), as described in our response to Critique #5. This revised analysis offers a clearer and more rigorous quantification of temporal precision.

We have updated the corresponding description and included the new analysis in the main text (**Page 13, Line 301**):

“Despite a much larger core and slower radial growth (**Extended Data Fig. 4b**), a **timing standard deviation of 2-4 hours was still achievable (Extended Data Fig. 4d)**, and most cells (92.9%) exhibited a time error no more than 6 hours (**Extended Data Fig. 4f**), showing GEMENI’s capability to record at a broad window.”

We have also toned down the statements on the temporal resolution in the manuscript. Previously we claimed that a sub-hour temporal resolution was achievable. In the revised manuscript, we changed it to “hour-level”:

Abstract: “Absolute chronological information of activity histories was attainable **with hour-level accuracy** through the integration of fiducial timestamps.” (**Page 2, Line 26**)

Discussion: “Using our model, the growth profile of GEMINI can be precisely resolved with as few as two timestamps. Exploiting this capability, artificial signals were temporally decoded with mean errors generally under **one hour** for individual particles. Even higher temporal resolution is achievable if more timestamps are employed as references for decoding.” (Page 28, Line 646)

7. Line 253: “Comparable basal level and response to tumor necrosis factor-alpha (TNF- α) were found between the groups with and without GEMINI growth, indicating negligible impact of GEMINI growth on the pathway.” They do have a higher baseline I κ B α phosphorylation with GEMINI, which might suggest a cellular stress response. Please acknowledge this in the text.

R: We thank the Reviewer for this insightful observation. Upon revisiting the data, we agree that the group expressing GEMINI exhibits a modest elevation in baseline I κ B α level and its phosphorylation compared to the control group. While the dynamic range and inducibility of the NF κ B pathway are largely preserved, we acknowledge that the elevated basal I κ B α level may reflect a mild cellular stress response associated with GEMINI particle growth.

We have revised the text accordingly to reflect this nuance. Specifically, we now state:

“Comparable response to tumor necrosis factor-alpha (TNF- α) were found between the groups with and without *in cellulo* GEMINI, indicating minimal impact of GEMINI on the pathway activation. **A modest increase in basal I κ B α level and its phosphorylation state was observed in GEMINI-expressing cells, which may indicate a low-level cellular stress associated with particle formation.**” (Page 14, Line 317)

8. Fig 5 and: “In the Y-maze test, mice from the two groups spent a comparable amount of time in the novel arm versus the alternative arm, indicating negligible impairment of short-term memory by GEMINI expression.” As the plots show, the average performance of mice with GEMINI seems similar to the control, however the spread of the metrics seems different which might be due to the particles?

R: We thank the Reviewer for raising the question regarding the variability difference in the Y-maze performance data.

While the mean time spent in the novel versus alternative arm was comparable between the GEMINI+ and GEMINI- groups, indicating no significant impairment of short-term memory, we acknowledge that the standard deviations differ between the two groups. Interestingly, the greater variability was observed in the GEMINI- group, not the GEMINI+ group. In fact, the poorest performing mouse in the entire cohort was from the GEMINI- group. This suggests that the observed variability is unlikely to stem from GEMINI expression and more likely reflects the inherent inter-animal variation typical for *in vivo* behavioral studies. Despite our efforts to minimize this by using littermate controls and random group assignment, such level of variation remains expected and common in behavioral studies.

Upon reviewing this point, we also recognized that the use of box plots may not have been ideal for a sample size of $N = 6$ mice per group, even though this is a commonly accepted group size in behavioral assays. Box plots can exaggerate apparent variability in small datasets. To improve clarity, **we have revised the data presentation using bar graphs that display individual data points alongside the mean and standard deviation, offering a transparent and more appropriate visualization of the results.**

9. Inheritance of particles could have an influence on the interpretation of a lot of the data. Meaning that only one daughter cell inherits the particle and the other one has to grow a new one with "erased" memory. How is this handled?

R: We appreciate the question raised by the Reviewer. We fully acknowledge that protein-assembly-based recorders like GEMINI do not possess inherent mechanisms for inheritance across cell divisions, in contrast to nucleic acid-based recorders, which leverage the faithful replication and transmission of genetic material to daughter cells. This key distinction means that, while GEMINI offers unique advantages, such as high spatial resolution and the ability to optically decode absolute temporal information at the single-cell level, it is intrinsically less suited for certain applications where continuous, intergenerational memory is essential, such as lineage-tracing. As a result, GEMINI and nucleic acid-based recorders are best viewed as complementary tools, each with strengths tailored to distinct biological questions.

Despite this limitation, we believe GEMINI remains highly valuable for applications in non-dividing cells, as well as in certain use cases involving dividing cells, where continuous lineage tracking is not a necessity. We discuss these two scenarios separately below:

1. **Non-dividing cells (e.g., neurons):** This inheritance limitation is naturally circumvented in these cells, therefore representing one of the major target application areas for GEMINI. Once nucleated, GEMINI particles remain within the same cell over extended periods, enabling uninterrupted, cell-specific recording of signaling dynamics. In our study, we demonstrated that GEMINI expression in neurons is stable and well tolerated, underscoring its suitability for long-term in vivo applications in the nervous system and other non-proliferative tissues.

2. **Dividing cells:** In dividing cells, GEMINI particles are typically inherited asymmetrically during mitosis, with one daughter cell retaining the original particle and the other initiating fresh nucleation. Therefore, one lineage maintains a continuous temporal record, while the other effectively starts a new "timeline" with a reset recorder.

Despite the loss of historical records in one lineage, this does not compromise the interpretability of the dataset from the other, as newly nucleated particles can be readily identified and distinguished from pre-existing ones. These new particles lack the earlier timestamp layers applied prior to division and can be excluded from analyses requiring uninterrupted temporal continuity, or analyzed independently for post-division dynamics.

To mitigate the limitations of asymmetric inheritance, we envision future strategies such as engineering multi-nucleation assemblies, which promote the formation of multiple GEMINI particles per cell. This would increase the likelihood that both daughter cells

retain at least one pre-existing recorder, helping preserve recording continuity across divisions. However, we recognize that this approach may not fully eliminate the loss of records over multiple cell generations, especially in highly proliferative systems.

Alternatively, the GEMINI platform could be integrated with lineage tracing methods. In cases where a daughter cell lacks earlier records, its lineage relationship could be used to trace back to its sibling cell, allowing retrieval of the missing temporal information from the shared ancestry.

We have now incorporated this discussion into the revised manuscript to acknowledge current limitations and outline potential solutions (**Page 30, Line 706**):

“It is important to acknowledge that GEMINI, as a protein-based recorder, does not inherently support inheritance across cell divisions. In future applications, especially when intergenerational continuity is critical, integrating GEMINI with lineage tracing methods could allow reconstruction of lost historical information through clonal relationships.”

10. Bulk vs. individual cells. As mentioned in the beginning of the review, in all the experiments where the authors are resolving time-stamps for events they appear to use averaged traces making this more of a bulk method (providing info of effects for a group of cells). This could be a clear limitation as in their artificial systems where they induce events with injections, this is fine as all cells are going through the same events at the same time, but even there it is hard to say where differences come from as differences in diffusion of the dyes they use for timestamping and compounds they use for inducing transcriptional changes can have an effect on their results. It would be nice if the authors could address these concerns with experiments, but in any case it is my opinion that the language about the method being able to record events on a single cell level should be toned down a lot.

R: We thank the Reviewer for their thoughtful comments regarding the distinction between bulk versus single-cell analyses in our study. We again apologize for leaving the impression that the use of average traces is necessary for temporal decoding.

In our opinion, **GEMINI recorder satisfies the criteria that define single-cell technology**. For this platform, each GEMINI particle forms and grows autonomously within a single cell and records that cell's unique signaling history. The temporal decoding is performed independently for each particle. Population-level averaging is not required to derive the decoded events.

Though pool analyses of many particles were employed in several experiments, this was utilized for statistical comparison between groups or experimental conditions. This approach is standard in single-cell analysis workflows (including single-cell RNA-seq and imaging-based reporters), where cell-level data are visualized collectively to capture trends, distributions, and between-group differences. However, this aggregation is analytical. It does not imply that the recording is population-based or lacks cellular resolution.

We also appreciate the Reviewer's point regarding the use of externally induced and synchronized stimuli. This design was intentional and reflects practical consideration in validating a new recording technology. Specifically, there is a dilemma in validating a temporal recording system: if one uses completely physiological and asynchronous events, such as spontaneous signaling patterns, it becomes impossible to validate the decoding results due to the lack of ground truths. To address this, we used well-defined, temporally controlled stimuli (e.g., TNF- α injection, timestamp dye delivery) to place all cells under synchronized conditions, enabling us to quantitatively evaluate whether GEMINI can correctly reconstruct the timing of known events. This controlled setting allows for systematic validation of decoding precision, error margins, and reproducibility.

We do agree that in more complex biological systems, especially under *in vivo* scenarios, the readouts from individual GEMINI particles reflect a composite of diverse variables, including but not limited to heterogeneity in stimulus propagation, signal transduction, and timestamp dye diffusion. Some of these factors are biologically meaningful (e.g., stimulus propagation), while others introduce systemic noise (e.g., variability in timestamp labeling). Disentangling these factors remains a challenge, not only for GEMINI but for most *in vivo* sensing platforms.

Our current strategy focuses on minimizing sources of systemic noise. Notably, our *in vivo* xenograft model exhibited reduced variability compared to the brain model, demonstrating that lower systemic noise is achievable under more controlled conditions or after systemic optimization. At this stage, we do acknowledge that GEMINI is not yet capable of resolving spontaneous, asynchronous signals in native tissues like the brain, such as neuronal activity linked to spontaneous behavior. This remains a future goal for further development.

In our revised manuscript we have added some discussion related to this point (**Page 29, Line 680**):

“It is noted that the temporal resolution of recording in the brain was lower than that observed in culture or xenografts, which could be attributed to factors such as the inherent heterogeneity of neurons, less accurate timestamping, variability in GEMINI expression, and less accurate predictive model for decoding. Disentangling these factors remains an ongoing challenge; nevertheless, GEMINI still holds great potential for further optimization in native tissues.”

Referee #4 (Remarks to the Author):

Nature is committed to facilitate training in peer-review and to ensure that everyone involved in our peer review process is appropriately recognised. This reviewer co-reviewed one of the listed reports.

R: We thank Reviewer #4 for the valuable input to our manuscript.

Referee #1

This manuscript has been thoroughly revised, and the authors have adequately addressed the concerns I raised previously. In the present version, I still have some minor concerns and critiques that I believe should be considered to further address. Overall, I recommend acceptance of this manuscript pending minor revisions.

R: We are glad to learn that the Reviewer found our revision thorough and satisfactory. We are also thankful for the Reviewer's constructive comments that helped us improve the work, as well as the recommendation of acceptance for publication after minor revisions. Regarding the follow-up comments, please find our responses below in a point-by-point fashion.

1. In the newly added Fig. S7, the authors examined the impact of GEMINI expression on cell morphology and subcellular structures. I noticed that the authors analyzed all signal (Hoechst, ConA, Syto14, MitoTracker) except for Phalloidin+WGA. Could the authors clarify why this analysis was excluded—was there any significant difference? In addition, since one of the GEMINI's important application is to record neuron activity history, have the authors tested its potential impact on neuronal subcellular structures? In Fig. 5a, GEMINI expression in neurons sometimes appears relatively large in size. Could GEMINI expression affect organelle transport (e.g., mitochondria or vesicles) from the soma to dendrites and axons?

R: We appreciate the Reviewer's follow-up questions. We did not quantitatively analyze the Phalloidin+WGA channel as the method was not provided in the open-source Cell Painting analysis pipeline used here. We utilized the recently published SPACe pipeline to analyze the Cell Painting data (Stossi et al. *Nature Commun.* **15**, 10170 (2024)). Under this analysis framework, the Phalloidin+WGA channel was not segmented and quantified, probably due to the challenges resolving the fine structures accurately. Therefore, we did not further explore the segmentation and analysis of Phalloidin+WGA. Instead, only the representative images were shown for visual comparison.

To further investigate this, we tried to segment the Phalloidin+WGA channel using the CellPose-SAM model and obtained reasonable segmentation results (**Fig. R1a**). No significant difference was found in the segmented area of Phalloidin+WGA for HEK293T and U2OS cells over 48 h of GEMINI growth (**Fig. R1b,c**). However, as this method is not part of the SPACe analysis framework, we prefer not to add the results to our manuscript.

It is also a good question whether GEMINI affects the transport behavior in neurons. To examine this, we performed an *in vivo* experiment in which we co-injected AAVs

encoding GEMINI and CD63-mScarlet (a vesicle marker) into the mouse brain. The mouse was sacrificed 10 days post-injection for imaging. We then segmented intrasomatic CD63 signal and quantified the number of CD63 puncta per neuron (**Fig. R2a**). Qualitative inspection of the imaging results did not reveal any obvious changes in puncta distribution within the soma. The average number of CD63 puncta per cell is comparable between the GEMINI+ and GEMINI- populations (**Fig. R2b**). These results suggest that GEMINI has a negligible impact on vesicle formation and/or somatic distribution. The dendrites and axons, however, can hardly be traced and segmented here with confidence due to their high aspect ratio and fine structure. As GEMINI particles are mostly located in soma where the vesicle number and distribution are negligibly affected by GEMINI, we believe GEMINI is most likely to exhibit minimal influence on vesicle transport to dendrites and axon.

Fig. R2 Impact of GEMINI growth on vesicle distribution in the mouse brain. **a**, Images (*top*) and segmentation results (*bottom*) of CD63 (vesicle) distribution within the mouse brain tissue. The intrasomatic vesicles were segmented using the soma segmentation results as masks. Scale bar: 100 μm . **b**, Statistical comparison of the number of vesicle puncta per soma in GEMINI+/- groups (n=77/272 neurons for GEMINI+/- groups, from 1 mouse). Box bounds: the 25th/75th percentiles; whiskers: min/max; squares: mean; and center lines: median.

2. In Extended Data Fig. 3, the authors present a clear side-by-side comparison of GEMINI and iPAK4 behavior during cell division. However, the methods for this experiment are not described in the manuscript. It would be helpful to provide details on how the experiment was performed, including what each signal represents and how the nucleus or membrane was labeled.

R: We apologize for the oversight of methods description for this experiment. The nucleus and membrane were labeled by transfecting cells with plasmids encoding GFP-GPI (membrane labeling) and H2B-mCherry (nucleus labeling). The transfected cells were characterized using time-lapse imaging following what was described in our “Time-lapse imaging” section in Methods.

As this method is specific to this single experiment, we included a brief experimental description in the figure legend. The revised version is shown below and can be found in **Page 19, Line 629**.

f, Snapshots of time-lapse imaging showing HEK293T cells with an intracellular linear recorder (iPAK4) during the division processes. Though mitosis was successful, the contractile ring failed to split the cell into two daughter cells and, therefore, resulted in one cell with two nuclei after division. The cellular labels were introduced via transient transfection, together with the plasmids encoding iPAK4. Yellow: membrane (GFP-GPI), Red: nucleus (H2B-mCherry), Green: iPAK4. Scale bars: 20 μm .”

In addition, we updated the “Cell culture and transfection” by adding a sentence describing the transient transfection strategy (Page 23, Line 788):

“For expressing *in cellulo* assemblies (including GEMINI and iPAK4) through transient transfection, plasmids encode untagged subunits and the FP/HT-tagged subunits were co-transfected at a molar ratio of ~10:1.”

3. There are some inconsistencies between figures, legends, and the main text that should be addressed. In Fig. 2, the authors optimized the color coding of each timestamp, but the figure legend was not updated accordingly. In Extended Data Fig. 9h, the figure legend states that the virus was injected into M1, while the main text (line 527) states that it was injected into the visual cortex. In Extended Data Fig. 10, the figure legend indicates V1, but it should be M1.

R: We apologize for the typos and are thankful that the reviewer pointed them out! We have updated the figure legends accordingly, as below:

Figure 2: “a, Experimental procedure for testing the precision of temporal decoding. The onset of signal-mimetic dye-addition events (violet) is decoded using two timestamps: yellow (t = 0) and blue (t = 11 h).” (Page 17, Line 524)

Extended Data Figure 9: “h, Schematic of the experimental design for *in vivo* characterization of neuronal activity. AAV encoding GEMINI was injected into the primary visual cortex (V1) of Thy1-jRGECO1a-WPRE transgenic mice on day 0, and two-photon calcium imaging was performed on day 21 with mice head-fixed on a running wheel.” (Page 21, Line 688)

Extended Data Figure 10: “a, Schematic of experimental design. AAV encoding GEMINI was unilaterally injected into the primary motor cortex (M1) of mice (GEMINI group). A positive control group received AAVs expressing diphtheria toxin subunit A (dtA) at a comparable dose (dtA group), while a negative control group was injected with equivolume saline (saline group).” (Page 21, Line 698)

4. In Fig. S6c, it is difficult to see GEMINI expression with the current contrast.

R: We appreciate the suggestion. The image exposure has been adjusted accordingly for better visualization of the GEMINI particles.

Referee #2

Yan and colleagues made extensive efforts and provided additional evidence and analysis to address our critiques and those of other reviewers. These include detailing experimental methods for the live/dead ratio (Fig. S6), examining *in vitro* (Fig. S7) and *in vivo* (Extended Data Figs. 9, 10) cytotoxicity, and showing single-cell recording with high-quality segmentation and registration (Extended Data Fig. 7). The additional xenograft experiment supported the claimed “single-cell/particle decoding capabilities” better than the previous mouse brain experiments. Overall, we support its publication.

R: We thank the reviewer for the constructive comments that helped us improve the manuscript and for supporting the publication of our work. Regarding the follow-up minor comments, please find our point-by-point response below.

A few minor points need further discussion or clarification:

1. Regarding single-cell/particle decoding capabilities, reviewer #3 also raised similar questions. Although the authors emphasized the performance *in vitro* (Figs. 2b, g, i; Extended Data Fig. 4; Fig. S8), our previous comment B.2 was specifically about GEMINI’s *in vivo* application. Considering the points the authors listed in their response (the heterogeneity of neurons, variability in timestamping and AAV delivery, etc. leading to the low confidence of “single-cell/particle” data points), can users learn which cell type or brain region responds to the induction earliest or most strongly?

R: This is an excellent question, and it is a point we are actively exploring in ongoing work.

At the current stage of our *in vivo* implementation, we do not yet have sufficient evidence to make a definitive and generalizable claim about which specific cell type or which brain region responds earliest or most strongly. A major limitation is the substantial variability introduced by local AAV delivery: even within the same brain region, individual neurons can receive drastically different AAV copy numbers, resulting in large cell-to-cell variations in expression level and GEMINI growth kinetics that can confound comparisons of the nucleation onset and growth rate.

[REDACTION]

2. Without cell membrane staining and segmentation, it's difficult to definitively link each particle to a specific cell (Fig 4b, e, g, i, and j). The authors demonstrated single-cell recording with high-quality segmentation and registration (Extended Data Fig. 7). However, ~10% of cells contained more than one assembly/particle (Extended Data Fig. 7d), and membrane staining/segmentation is not a standard step when decoding or tracing events in most analyses. As a result, each dot represents one particle/assembly, but not necessarily a unique cell. We suggest using the description “single-particle/assembly” and reserving “single-cell” only if you can definitively link a particle to a cell.

R: We completely understand the reviewer’s concern. We agree that although most cells nucleate only one particle and most of the particles can be assigned to a unique cell confidently (**Extended Data Fig. 7f**), our current analyses are primarily focused on individual particles rather than the cells they reside in, and therefore “single-particle” is a better term to describe what was achieved in this work. In the current version, the term of “single-cell” is not used in the discussion of our results, while the discussion was focused more on the “single-particle” level. Relevant discussions include:

“We further analyzed time error between decoded onsets and ground truths, where **75.9%** of the **single-particle** decoding results fell within ± 1 hour of the ground truths, **98.2%** within ± 2 hours, and **100%** within ± 3 hours, showing precise temporal decoding at **single-particle level** (**Fig. S8**). It is noteworthy that we only deployed two timestamps here. Even higher temporal resolution can be expected with more timestamps.” (**Page 7, Line 184**)

“Despite a much larger core and slower radial growth (**Extended Data Fig. 4b**), a timing s.d. of 2-4 hours was still achieved (**Extended Data Fig. 4d**), and most **particles** (92.9%) still exhibited a time error of no more than 6 hours (**Extended Data Fig. 4f**), showing GEMINI’s capability to record at an extended window.” (**Page 7, Line 190**)

Discussion acknowledging the current limitation of *in vivo* GEMINI implementation was also included as:

“It is noteworthy that these analyses utilize single-particle readouts to quantify population-level trends. A true atlas-like map of neuronal dynamics at the single-cell level is still not attainable at the current stage, largely due to variations in decoding.” (**Page 13, Line 368**)

“The temporal resolution in the brain, however, was lower than that in culture or xenografts, owing to factors such as inherent heterogeneity of neurons, less accurate timestamping, variability in GEMINI expression, and less accurate decoding model. **Disentangling these factors and resolving spatiotemporal maps of single-cell activities without populational analysis requires further efforts on homogenous expression and precise signal decoding.**” (Page 14, Line 407)

We are thankful for the reviewer’s suggestion that help us improve the rigor of the work.

3. Fig. 4c suggested level-dependent signal, and the authors reasoned that cytokine diffusion may contribute to the peak delay. In the revision experiments (Figs. 4i–l), do GEMINI particles closer to vascularization also show higher response levels?

R: This is a great question! To examine it, we performed a quantitative analysis relating response level to proximity to vasculature by correlating the normalized GFP intensity of individual GEMINI particles with their distance to the nearest blood vessel. [REDACTION]

[REDACTION]

[REDACTION]

Distance to nearest vasculature

It is noteworthy that such analysis has a limitation: the xenograft sections here were only 150 μm thick, and our distance calculation can only account for vasculature within the same section. As a result, some particles classified as “far” or “ultra-far” may in fact be closer to vessels located in adjacent sections that are not captured in our imaging dataset. We believe that a more rigorous assessment of this question would require 3D imaging of optically cleared thick sections or entire xenograft volumes, together with an analysis workflow capable of determining proximity in 3D. However, we currently do not have such xenograft specimens as well as an optimized sample processing/analysis pipeline for their high-quality 3D imaging and analyses. We feel our analysis in 2D, though indeed showed differences, is not sufficient to support

quantification and mechanistic interpretation. Considering that this characterization is not central to the focus of this work, we decide not to discuss these preliminary results in the current manuscript. Nonetheless, we agree this is an important question that is worth dedicated follow-up study.

4. In Figs. S7a and S7f, Syto14 staining showed more puncta in the nucleus under GEMINI+ conditions. If Hoechst is used for segmentation and Syto14 staining is compared only in the nucleus between GEMINI+ and GEMINI- groups, would you see any differences?

R: This is a good point and in fact Hoechst segmentation has been used as masks in our Syto14 analysis. To analyze the Cell Painting results, we followed the open-source SPACe pipeline that was published recently (Stossi et al. *Nature Commun.* **15**, 10170 (2024)). In this analysis, the nuclear (Hoechst) was first segmented, while nucleolar masks are further generated by adaptive Otsu & MaxEntropy thresholding, using nuclear masks for nucleolar segmentation exactly as the Reviewer recommended. In this case, the cytosolic Syto14 staining was effectively excluded during the identification of the nucleolar region. To show this, a representative segmentation mask from our previous analysis is presented as a reference (**Fig. R5**).

To clarify this point, we added more details to the Supplementary Methods accordingly:

“For image analysis, the Hoechst channel was segmented using the generalized CellPose-SAM model¹ on the Hoechst channel; these masks served as seeds for watershed-based cell segmentation on the ConA-488 channel. Syto14 and MitoTracker channels were segmented using adaptive Otsu and MaxEntropy thresholding. **Syto14 was segmented using cell nuclei as masks. The data analysis pipeline followed the open-source SPACe pipeline².**” (**Page 7, Line 140 in SI**)

Fig. R5 Representative segmentation results of Hoechst and Syto14. Images (*left*) and their corresponding segmentation results (*right*) of Hoechst (*top*) and Syto14 (*bottom*) channels. The Hoechst segmentation was used as the mask for Syto14 segmentation. Scale bar: 20 μm .

5. What does the green color represent in the heatmap in the Fig. 4b inset (right)?

R: We apologize for the oversight. The green structure represents the segmentation of vasculature. It has been added to the figure caption accordingly, as shown below.

“**b**, Section of a HEK293T xenograft showing GEMINI recording of NF κ B activation following systemic injection of 3mg kg⁻¹ LPS. GEMINI: violet, NF κ B: green, vasculature (anti-CD31): red. Insets: magnified 3D image (left) and the corresponding heatmap

displaying the normalized GFP intensity ($F_{\text{GFP}}/F_{\text{HTL}}$) of individual particles (right). **Green in the heatmap represents vasculature segmentation.** Scale bars: 1 mm, 100 μm (insets).” (Page 17, Line 561)

Referee #3

The revised manuscript by Yan and co-workers includes new experiments and data and is an improved version of their previous manuscript. Nevertheless it appears that the revisions leave some of the main criticisms.

R: We are glad that the reviewer finds our manuscript improved in the last round of revision. To address the remaining criticisms, we have included our responses to the comments below point by point.

The authors have added a new study where they implanted GEMINI-expressing cells into immunodeficient mice and triggered inflammation by injection of LPS. I recommend the authors for this effort and I think it strengthens their manuscript. In particular, it shows a possibility of measuring dosage response, which I believe is a feature not commonly found in other cellular recording methods. The authors write that “This model is less strong and artificial than the PTZ-induced seizure.” Maybe that is true for the inflammation aspect of the model, but I would argue that subcutaneous xenograft into immunodeficient mice is artificial. So still, it is not clear to this reviewer how this significantly changes the manuscript in terms of showing a clear case for previously inaccessible discoveries.

R: We are grateful for the Reviewer’s recommendation in the previous round that help us strengthen the manuscript! We also thank the Reviewer for acknowledging the uniqueness of our technology among the existing cellular recording methods.

We agree with the Reviewer that subcutaneous xenograft transplantation into immunodeficient mice is an artificial model. Our intent in the prior discussion was to contrast the mode of induction (systemic inflammatory stimulation by LPS versus chemically induced seizures by PTZ), rather than to suggest that the xenograft setting is broadly native. We apologize for any wording that may have implied otherwise. **We have carefully reviewed the current version of the manuscript and ensured that potentially misleading phrasing like this is not included in the text.**

We would also like to clarify the purpose of introducing the xenograft/LPS model in this revision. This experiment was designed to demonstrate GEMINI’s feasibility and recording properties *in vivo*, rather than to make a new biological discovery in the present study. The primary goal of this work is to establish GEMINI as a recording platform by (i) defining its design principles, (ii) characterizing key performance metrics (e.g., sensitivity, dynamic range, and dose dependence), and (iii) demonstrating feasibility across multiple *in vivo* contexts. Discovery-oriented biological applications were therefore not included within our scope of this tool-development work. In fact, because the manuscript’s central focus is technology innovation, we intentionally avoided pursuing deep, model-specific biological questions that could shift the emphasis away from establishing the platform. As a team with primary expertise in

biotechnology and bioengineering, our goal here is to rigorously characterize GEMINI and present representative use-case scenarios that inspire and motivate researchers in diverse fields to deploy the tool and pursue fundamental questions that were previously difficult to access.

Finally, we fully agree with the Reviewer that discovery-focused studies using GEMINI are exciting and important, and we are actively moving in this direction through collaborations with experts in diverse fields, especially neuroscientists and cancer biologists. For example, we have adapted GEMINI with collaborators to track YAP signaling dynamics in tumor mechanobiology both *in vitro* and *in vivo*. We anticipate that the capabilities demonstrated here will support even broader applications and facilitate more discovery-driven research in the near future.

The authors strongly argue that the method is single-cell and that averages are only used for statistical purposes. I agree with the authors that the readout in itself is single-cell, but that, in contrast to scRNA-seq, the method is not used to differentiate between cell types (or in this case, cell reactions to events), but rather to measure the population as a whole. I still think that the way the data is used, not to look for differences, but to look for average responses, makes it harder to argue the case that this is significantly disparate from bulk methods. Again, I agree that the readout is single-cell, and that this could potentially be leveraged to look for cell-type or cell-position specific responses in the future, but it is not leveraged in this study. This needs to be concretely stated in the manuscript in the form that yes, in the future this method can potentially differentiate between different responses in different populations, but with the current state of the method and data, this remains hypothetical.

R: We thank the Reviewer for this clarification, and we agree with the Reviewer's viewpoint. GEMINI's measurement unit is single-particle (i.e., each GEMINI particle provides an individual, spatially resolved record), but in the current manuscript, especially in the *in vivo* models, we primarily use these single-particle readouts to quantify population-level trends (e.g., mean/median response and dose dependence) rather than to explicitly identify divergent responses at the single-cell level. We acknowledge that, as presented, this makes the manuscript's data analysis closer in spirit to a population-response measurement, albeit derived from many single-cell observations, than to a "single-cell atlas".

In this version of manuscript, we explicitly acknowledge the limitation of the GEMINI platform at its current stage:

"Two timestamps were applied with a 24-hour interval and seizure was induced at either 0 or 24 hours (**Fig. 5o**). The individual particles (**Fig. 5p**) and the mean normalized fluorescence profiles (**Fig. S14d**) showed expected band positions corresponding to the timing of induction. Statistical analysis revealed significance between the two groups (**Fig. 5q**). **It is noteworthy that these analyses utilize single-particle readouts to quantify population-level trends. A true atlas-like map of neuronal dynamics at the single-cell level is still not attainable at the current stage, largely due to variations in decoding.**" (**Page 13, Line 368**)

“Disentangling these factors and resolving spatiotemporal maps of single-cell activities without populational analysis requires further efforts on homogenous expression and precise signal decoding. Continued optimization would help transform GEMINI into a platform broadly applicable to spatiotemporally resolve cellular histories in animals, enabling deeper insight into the mechanisms underlying health and disease.” (Page 14, Line 410)

Regarding my specific critique 1, the authors provide answers that are mostly satisfactory except for how fig. 1h (previously 2d) is not reliant on population averages. The authors do not in my opinion explain how one can do away with the population averaging in 1h. In fact, I would say that their reply highlight why this is a valid critique “While decoding variability exists, as is common with any biological sensing platform, it arises primarily from factors including: (1) intrinsic cellular heterogeneity that affects signal response kinetics, (2) variability in protein expression levels and thus the growth dynamics of GEMINI particles, and (3) error associated with timestamping and the temporal decoding model.” Again, the wording in the manuscript needs to acknowledge this.

R: We thank the Reviewer for reiterating this point, and we apologize for our earlier misunderstanding of the critique. We now understand that the Reviewer’s comment was not that population averaging is intrinsic to the GEMINI recording mechanism, but rather that population averaging was used in **Fig. 1h** to characterize/illustrate the growth profile, which is absolutely correct.

To make this point clearer to the audience, we add the wording below to explicitly acknowledge that population average was indeed utilized in this experiment:

“An equation was derived that linked the radius (R) of particles with time (t): $R = (Kt + A)^{1/3}$, where K and A are constants. The model was examined by fitting the mean linearized growth profile (defined as volume index) **from population average of representative particles**, showing an R^2 of 0.995 (**Fig. 1h**), while the fitting of growth profiles from individual particles yielded a mean R^2 of 0.982 (**Fig. 1i**).” (Page 6, Line 130)

Overall, I think this is very nice work, but some main criticisms from the initial review remains.

R: We really appreciate the positive comments from the Reviewer! We hope our responses and further revisions above have addressed the concerns.

Referee #4

I co-reviewed this manuscript with one of the reviewers who provided the listed reports.

R: We thank the Reviewer for the valuable input to our manuscript.

Referees' comments:

Referee #1 (Remarks to the Author):

Thank you for addressing all of my concerns, I have no further revisions.

R: We thank the Reviewer for the time and constructive input that helped us improve our work.

Referee #2 (Remarks to the Author):

The authors have addressed our comments comprehensively. We look forward to future in vivo applications with optimized AAV delivery or transgenic lines. Regarding our question on cytokine diffusion, our intent was not to request a mechanistic investigation or more detailed 3D characterization; rather, the presented data implies a decent in vivo dose sensitivity within a local region (even within $\sim 150 \mu\text{m}$). Overall, this work now represents a valuable recording method that merits publication in Nature.

R: We thank the Reviewer for the time and constructive input that helped us improve our work.

Referee #3 (Remarks to the Author):

Given the discussion outlined in the reply to reviewers I think this sentence in the introduction (line 43-44) should be removed:

"However, many recorders rely on populational analyses to obtain faithful decoding, losing crucial single-cell information."

I don't think it is fair to contrast with previous methods like this given that the present work also largely rely on population anaynses to obtain useful information (at this stage).

No further comments.

R: We thank the Reviewer for the comment. The sentence has now been removed as suggested. We appreciate the constructive input from the Reviewer that helped us improve our work.

Referee #4 (Remarks to the Author):

I co-reviewed this manuscript with one of the reviewers who provided the listed reports.

R: We thank the Reviewer for co-reviewing our work!